# Robust Offline Reinforcement Learning with Linearly Structured $f$-Divergence Regularization

**Cheng Tang** [* 1]   **Zhishuai Liu** [* 2]   **Pan Xu** [2]

## Abstract

The Robust Regularized Markov Decision Process (RRMDP) is proposed to learn policies robust to dynamics shifts by adding regularization to the transition dynamics in the value function. Existing methods mostly use unstructured regularization, potentially leading to conservative policies under unrealistic transitions. To address this limitation, we propose a novel framework, the $d$-rectangular linear RRMDP ($d$-RRMDP), which introduces latent structures into both transition kernels and regularization. We focus on offline reinforcement learning, where an agent learns policies from a precollected dataset in the nominal environment. We develop the Robust Regularized Pessimistic Value Iteration (R2PVI) algorithm that employs linear function approximation for robust policy learning in $d$-RRMDPs with $f$-divergence based regularization terms on transition kernels. We provide instance-dependent upper bounds on the suboptimality gap of R2PVI policies, demonstrating that these bounds are influenced by how well the dataset covers state-action spaces visited by the optimal robust policy under robustly admissible transitions. We establish information-theoretic lower bounds to verify that our algorithm is near-optimal. Finally, numerical experiments validate that R2PVI learns robust policies and exhibits superior computational efficiency compared to baseline methods.

## 1. Introduction

Offline reinforcement learning (RL) (Levine et al., 2020) facilitates policy learning from fixed datasets, eliminating the need for direct interaction with the environment. When the policy deployment environment differs from the one where the dataset was collected, robust policies that remain effective under the environment shift are required (García & Fernández, 2015; Packer et al., 2018; Zhang et al., 2020; Wang et al., 2024b; Guo et al., 2024). A widely adopted framework for learning such policies is the distributionally robust Markov decision process (DRMDP) (Iyengar, 2005; Nilim & El Ghaoui, 2005), which models dynamics changes as an uncertainty set around the nominal transition kernel. In this setup, an agent seeks policies performing well even in the worst-case environment within the uncertainty set. The most common design of uncertainty sets is the $(s, a)$-rectangularity (Iyengar, 2005; Nilim & El Ghaoui, 2005), which independently models uncertainty for each state-action pair. Although mathematically elegant, the $(s, a)$-rectangularity can result in overly conservative policies, especially when the state and action spaces are large. To address this issue, Goyal & Grand-Clement (2023) introduce the $r$-rectangular uncertainty set, which parameterizes transition kernels using latent factors. This concept has since been incorporated into $d$-rectangular linear DRMDPs ($d$-DRMDPs, Ma et al. (2022)), extending its applicability to robust decision-making with linear function approximation. Building on $d$-DRMDPs, recent works (Blanchet et al., 2024; Wang et al., 2024a; Liu & Xu, 2024b) propose provably efficient algorithms that leverage function approximation for robust policy learning.

However, the $d$-DRMDP framework has several problems that remain unaddressed, which we summarize as follows. *Theoretical Gaps:* Current understanding of $d$-DRMDPs is largely restricted to uncertainty sets defined by the Total Variation (TV) divergence (Liu & Xu, 2024a;b). For uncertainty sets defined by the Kullback-Leibler (KL) divergence, prior works (Ma et al., 2022; Blanchet et al., 2024) rely on additional regularity assumptions regarding the KL dual variable, which is hard to validate in practice. Moreover, the $\chi^2$-divergence defined uncertainty set has demonstrated effectiveness in certain empirical applications (Panaganti & Kalathil, 2022; Xu et al., 2023) and has also been analyzed under the $(s, a)$-rectangularity (Shi et al., 2024). Yet there are no theoretical results or efficient algorithms for $d$-DRMDPs. *Practical challenges:* Existing practical algorithms (Ma et al., 2022; Liu & Xu, 2024b; Wang et al.,

---

[*]Equal contribution  [1] University of Illinois Urbana-Champaign, work was done when Cheng was in Tsinghua University  [2]Duke University. Correspondence to: Pan Xu <pan.xu@duke.edu>.

*Proceedings of the $42^{st}$ International Conference on Machine Learning*, Vancouver, Canada. PMLR 267, 2025. Copyright 2025 by the author(s).

2024a) depend on a dual optimization oracle (see Remark 4.2 in Liu & Xu (2024a)) to estimate the robust value function. The computation complexity of these methods is proportional to the feature dimension $d$ and the planning horizon $H$. While heuristic methods like the Nelder-Mead algorithm (Nelder & Mead, 1965) can approximate the oracle, they become computationally expensive when dealing with high-dimensional features (large $d$) and extended planning horizons (large $H$), which are common in real-world applications. These limitations raise an important question:

> *Can we design efficient offline robust RL algorithms using general $f$-divergence[1] uncertainty models with linearly structured transitions?*

In this work, we provide a positive answer to this question. Inspired by the robust regularized MDP (RRMDP) framework with the $(s, a)$-rectangularity condition (Yang et al., 2023; Zhang et al., 2020; Panaganti et al., 2024a), where the uncertainty set constraint in DRMDP is replaced by a regularization penalty term measuring the divergence between the nominal and perturbed dynamics, we propose the $d$-rectangular linear RRMDP ($d$-RRMDP) framework. Specifically, $d$-RRMDP replaces the $d$-rectangular uncertainty set in $d$-DRMDPs with a carefully designed penalty term that preserves the linear structure. The motivations are two folds: (1) it has been shown by Yang et al. (2023) that the robust value function under the RRMDP is equivalent to that under the DRMDP with $(s, a)$-rectangularity as long as the regularizer is properly chosen; (2) removing the uncertainty set constraint simplifies the dual problem for certain divergences (Zhang et al., 2024), potentially improving computational efficiency and facilitating theoretical analysis. We summarize our contributions as follows:

- We establish that key dynamic programming principles, including the robust Bellman equation and the existence of deterministic optimal robust policies, hold under the $d$-RRMDP framework. Additionally, we derive dual formulations of robust Q-functions with TV, KL and $\chi^2$ divergences defined regularization, highlighting their linear structures.

- We propose a computationally tractable meta-algorithm, Robust Regularized Pessimistic Value Iteration (R2PVI), for offline $d$-RRMDPs with general $f$-divergence regularization. For TV, KL, and $\chi^2$ divergences, we provide instance-dependent upper bounds on the suboptimality gap of policies learned by R2PVI, in a general form of $\beta \sup_{P \in \mathcal{U}^\lambda(P^0)} \sum_{h=1}^H \mathbb{E}^{\pi^\star, P} \left[ \sum_{i=1}^d \|\phi_i(s,a)\mathbf{1}_i\|_{\Lambda_h^{-1}} \mid s_1 = s \right]$, where $d$ is the feature dimension, $H$ is the horizon length, $\phi(s, a)$ is the feature mapping, $\lambda$ is the

regularization parameter, and $\beta$ is a problem-dependent parameter whose specific form depends on the choice of the divergence (see Section 5.1 for details). The set $\mathcal{U}^\lambda(P^0)$ is derived from our theoretical analysis, and it does not represent an uncertainty set in the conventional DRMDP framework. We further construct an information-theoretic lower bound, demonstrating that this instance-dependent uncertainty function is intrinsic.

- We conduct experiments in simulated environments, including a linear MDP setting (Liu & Xu, 2024a) and the American Put Option environment (Tamar et al., 2014). Our findings show that: 1. The $d$-RRMDP framework yields equivalent robust policies as $d$-DRMDP with appropriately chosen regularization parameters. 2. R2PVI significantly improves algorithms designed for $d$-DRMDPs in terms of the computation complexity, and is comparable to algorithms designed for standard linear MDPs.

**Notations.** In this paper, we denote $\Delta(\mathcal{S})$ as the probability distribution in the state space $\mathcal{S}$. For any $H \in \mathbb{N}$, $[H]$ represents the set $\{1, 2, 3, \cdots, H\}$. For a vector $\boldsymbol{v} \in \mathbb{R}^d$, we denote $v_i$ as the $i$-th element. For any function $V : \mathcal{S} \to [0, H]$, we denote $V_{\min} = \min_{s \in \mathcal{S}} V(s)$ and $V_{\max} = \max_{s \in \mathcal{S}} V(s)$. For any distribution $\mu \in \Delta(\mathcal{S})$, we denote $\mathrm{Var}_{s \sim \mu} V(s)$ as the variance of the random variable $V(s)$ under $\mu$. For any two probability measures $P$ and $Q$ satisfying that $P$ is absolute continuous with respect to $Q$, the $f$-divergence is defined as $D_f(P\|Q) = \int_{\mathcal{S}} f(P(s)/Q(s))Q(s)\mathrm{d}s$, where $f$ is a convex function on $\mathbb{R}$ and differentiable on $\mathbb{R}_+$ satisfying $f(1) = 0$ and $f(t) = +\infty, \forall t < 0$. The Total Variation (TV) divergence, Kullback-Leibler (KL) divergence and Chi-Square ($\chi^2$) divergence between $P$ and $Q$ are defined by $f(x) = |x - 1|/2, f(x) = x \log x, f(x) = (x - 1)^2$, respectively. Given a scalar $\alpha$, we denote $[V(s)]_\alpha = \min\{V(s), \alpha\}$. Given an interval $I$, we define $[V(s)]_I$ as the result of clipping $V(s)$ to lie within the interval $I$. We denote $\mathbf{I}$ as the identity matrix and $\mathbf{1}_i \in \mathbb{R}^d$ as the one-hot vector with the $i$-th element equals to one.

## 2. Related Work

**Distributionally Robust MDPs.** The seminal works of Satia & Lave Jr (1973); Iyengar (2005); Nilim & El Ghaoui (2005) proposed the framework of DRMDP. There are several lines of works studying DRMDPs under different settings. Zhou et al. (2021); Panaganti et al. (2022; 2024b); Shi & Chi (2024); Liu & Xu (2025) studied the offline DRMDP assuming access to an offline dataset and provided sample complexity bounds under the coverage assumption on the offline dataset. Liu & Xu (2024a); Liu et al. (2024); Lu et al. (2024) studied the online DRMDP where an agent learns robust policies by actively interacting with the nominal environment. Blanchet et al. (2024); Panaganti et al. (2022) stud-

---

[1]The general $f$-divergence includes widely studied divergences such as Total Variation, Kullback-Leibler, and $\chi^2$ divergences.

ied the DRMDP with general function approximation, they focused on the offline setting with the $(s, a)$-rectangularity assumption. Ma et al. (2022); Liu & Xu (2024b); Wang et al. (2024a) studied the offline $d$-DRMDP, they proposed provably efficient and computationally tractable algorithms and provided sample complexity bounds under different kinds of coverage assumptions on the offline dataset.

**RRMDPs.** The work of Yang et al. (2023); Zhang et al. (2024) proposed the RRMDP, which can be regarded as a generalization of the DRMDP by substituting the uncertainty set constraint in DRMDP with the regularization term defined as the divergence between the perturbed model and the nominal model. In particular, Yang et al. (2023) studied the tabular RRMDP and proposed a model-free algorithms assuming access to a simulator. Zhang et al. (2024) studied the offline RRMDP, they established connections between RRMDPs with risk sensitive MDPs, and derived the policy gradient principle. Moreover, they studied general function approximation and proposed a computationally efficient algorithm, RFZI, for RRMDPs with KL-divergence defined regularization terms. Zhang et al. (2024) firstly discovered that the duality of the robust value function has a closed expression under the KL-divergence. Panaganti et al. (2024a) studied the offline RRMDP with regularization terms defined by the general $f$-divergence. They studied general function approximation and provided sample complexity results. They further proposed a hybrid algorithm, which learns robust policies with both historical data and interactive data collection, for RRMDPs with TV-divergence defined regularization term. Existing works focus on the $(s, a)$-rectangularity uncertainty regularization, which is different from ours.

## 3. Problem Formulation

In this section, we provide preliminaries for RRMDPs.

**Markov decision process (MDP).** We first introduce the concept of MDPs, which is the basis of our settings. Specifically, we denote $\text{MDP}(\mathcal{S}, \mathcal{A}, H, P^0, r)$ as a finite horizon MDP, where $\mathcal{S}$ is the state space, $\mathcal{A}$ is the action space, $H$ is the horizon length, $P^0 = \{P_h^0\}_{h=1}^H$ are nominal transitional kernels, and the $r(s, a) \in [0, 1]$ is the deterministic reward function assumed to be known in advance. For any policy $\pi$, the value function and Q-function at time step $h$ are defined as $V_h^\pi(s) = \mathbb{E}^{P^0}\left[\sum_{t=h}^H r_t(s_t, a_t) | s_h = s, \pi\right]$, and $Q_h^\pi(s, a) = \mathbb{E}^{P^0}\left[\sum_{t=h}^H r_t(s_t, a_t) | s_h = s, a_h = a, \pi\right]$.

**Robust regularized MDP (RRMDP)** We define a finite horizon RRMDP as $\text{RRMDP}(\mathcal{S}, \mathcal{A}, H, P^0, r, \lambda, D, \mathcal{F})$, where $\lambda$ is the regularizer, $D$ is the probability divergence metric, and $\mathcal{F}$ is the feasible set of all perturbed transition kernels. For any pol-

icy $\pi$, the robust regularized value function is defined as $V_h^{\pi,\lambda}(s) = \inf_{P \in \mathcal{F}} \mathbb{E}^P\left[\sum_{t=h}^H \left[r_t(s_t, a_t) + \lambda D(P_t(\cdot|s_t, a_t) \| P_t^0(\cdot|s_t, a_t))\right] | s_h = s, \pi\right]$ and robust Q-function as $Q_h^{\pi,\lambda}(s, a) = \inf_{P \in \mathcal{F}} \mathbb{E}^P\left[\sum_{t=h}^H \left[r_t(s_t, a_t) + \lambda D(P_t(\cdot|s_t, a_t) \| P_t^0(\cdot|s_t, a_t))\right] | s_h = s, a_h = a, \pi\right]$.

The RRMDP framework has been referred to by different names in the literature, including the penalized robust MDP (Yang et al., 2023), the soft robust MDP (Zhang et al., 2024), and the robust $\phi$-regularized MDP (Panaganti et al., 2024a). For consistency, we adopt the term RRMDP in this work. In RRMDPs, the perturbed transition kernel class $\mathcal{F}$ typically encompasses all possible kernels. However, for environments with large state-action spaces, $\mathcal{F}$ may be overly broad, including transitions that are unrealistic or irrelevant. To address this, we introduce latent structures on transition kernels and design regularization terms that penalize changes in the latent structure, sharing similar ideas with the design of $r$-rectangular (Goyal & Grand-Clement, 2023) and $d$-rectangular (Ma et al., 2022) uncertainty sets.

**The $d$-rectangular linear RRMDP ($d$-RRMDP).** In this paper, we propose the novel $d$-RRMDP, which admits a linear structure of the feasible set and reward function. Specifically, a $d$-RRMDP is a RRMDP where the nominal environment $P^0$ is a special case of linear MDP with a simplex feature space (Jin et al., 2020, Example 2.2), and the feasible set $\mathcal{F}$ involves kernels defined based on the linear structure of the nominal transition kernel. We make the following assumption on reward functions and transition kernels:

**Assumption 3.1** (Jin et al. (2020)). Given a known state-action feature mapping $\phi : \mathcal{S} \times \mathcal{A} \to \mathbb{R}^d$ satisfying $\sum_{i=1}^d \phi_i(s, a) = 1, \phi_i(s, a) \geq 0$, we assume the reward function $\{r_h\}_{h=1}^H$ and the nominal transition kernels $\{P_h^0\}_{h=1}^H$ admit linear structures. Specifically, for all $(h, s, a) \in [H] \times \mathcal{S} \times \mathcal{A}$, we have $r_h(s, a) = \langle \phi(s, a), \theta_h \rangle$, and $P_h^0(\cdot|s, a) = \langle \phi(s, a), \mu_h^0(\cdot) \rangle$, where $\{\theta_h\}_{h=1}^H$ are known vectors with bounded norm $\|\theta_h\|_2 \leq \sqrt{d}$ and $\mu_h^0 = (\mu_{h,1}^0, \mu_{h,2}^0, \cdots, \mu_{h,d}^0), \mu_{h,i}^0(\cdot) \in \Delta(\mathcal{S}), \forall i \in [d]$.

With Assumption 3.1, the robust regularized value function and Q-function are defined as

$$V_h^{\pi,\lambda}(s) = \inf_{\mu_t \in \Delta(\mathcal{S})^d, P_t = \langle \phi, \mu_t \rangle} \mathbb{E}^{\{P_t\}_{t=h}^H}\left[\sum_{t=h}^H \left[r_t(s_t, a_t)\right.\right.$$
$$\left.\left. + \lambda \langle \phi(s_t, a_t), \boldsymbol{D}(\mu_t \| \mu_t^0) \rangle\right] | s_h = s, \pi\right], \quad (3.1)$$

$$Q_h^{\pi,\lambda}(s, a) = \inf_{\mu_t \in \Delta(\mathcal{S})^d, P_t = \langle \phi, \mu_t \rangle} \mathbb{E}^{\{P_t\}_{t=h}^H}\left[\sum_{t=h}^H \left[r_t(s_t, a_t)\right.\right.$$
$$\left.\left. + \lambda \langle \phi(s_t, a_t), \boldsymbol{D}(\mu_t \| \mu_t^0) \rangle\right] | s_h = s, a_h = a, \pi\right],$$

where $\boldsymbol{D}(\mu \| \mu^0) = [D(\mu_i \| \mu_i^0)]_{i \in [d]}$. In other words, we

only consider perturbed kernels in the linear feasible set

$$\mathcal{F}_{\mathrm{L}} = \big\{ P = \{P_h\}_{h=1}^H | P_h(\cdot|s,a) = \langle \phi(s,a), \boldsymbol{\mu}_h(\cdot) \rangle,$$
$$\boldsymbol{\mu}_h = (\mu_{h,1}, \mu_{h,2}, ... \mu_{h,d})^\top, \mu_{h,i}(\cdot) \in \Delta(\mathcal{S}), \forall i \in [d] \big\}.$$

The optimal robust regularized value function and Q-function are defined as:

$$V_h^{\star,\lambda}(s) = \sup_\pi V_h^{\pi,\lambda}(s), \tag{3.2}$$
$$Q_h^{\star,\lambda}(s,a) = \sup_\pi Q_h^{\pi,\lambda}(s,a).$$

Based on (3.2), the optimal robust policy is defined as the policy that achieves the optimal robust regularized value function, $\pi^{\star,\lambda} = \operatorname{argmax}_\pi V_1^{\pi,\lambda}(s)$, $\forall s \in \mathcal{S}$.

**Dynamic programming principles for $d$-RRMDPs** For completeness, we first show that the dynamic programming principles (Sutton & Barto, 2018) hold for $d$-RRMDPs.

**Proposition 3.2.** (Robust Regularized Bellman Equation) Under the $d$-rectangular linear RRMDP, for any policy $\pi$ and any $(h,s,a) \in [H] \times \mathcal{S} \times \mathcal{A}$, we have

$$Q_h^{\pi,\lambda}(s,a) = r_h(s,a) + \inf_{\boldsymbol{\mu}_h \in \Delta(\mathcal{S})^d, P_h = \langle \phi, \boldsymbol{\mu}_h \rangle} \Big[$$
$$\mathbb{E}_{s' \sim P_h(\cdot|s,a)}\big[V_{h+1}^{\pi,\lambda}(s')\big] + \lambda \langle \phi(s,a), \boldsymbol{D}(\boldsymbol{\mu}_h \| \boldsymbol{\mu}_h^0) \rangle \Big],$$
$$V_h^{\pi,\lambda}(s) = \mathbb{E}_{a \sim \pi(\cdot|s)}\big[Q_h^{\pi,\lambda}(s,a)\big]. \tag{3.3}$$

Next, we show that the optimal robust policy is deterministic and stationary. Hence, we can restrict the policy class $\Pi$ to the deterministic and stationary one.

**Proposition 3.3.** Under the $d$-rectangular linear RRMDP, there exists a deterministic and stationary policy $\pi^\star$, such that for any $(h,s,a) \in [H] \times \mathcal{S} \times \mathcal{A}$, $V_h^{\pi^\star,\lambda}(s) = V_h^{\star,\lambda}(s)$, and $Q_h^{\pi^\star,\lambda}(s,a) = Q_h^{\star,\lambda}(s,a)$.

With Proposition 3.2 and Proposition 3.3, we can derive the following robust regularized Bellman optimality equation:

$$Q_h^{\star,\lambda}(s,a) = r_h(s,a) + \inf_{\boldsymbol{\mu}_h \in \Delta(\mathcal{S})^d, P_h = \langle \phi, \boldsymbol{\mu}_h \rangle} \Big[$$
$$\mathbb{E}_{s' \sim P_h(\cdot|s,a)}\big[V_{h+1}^{\star,\lambda}(s')\big] + \lambda \langle \phi(s,a), \boldsymbol{D}(\boldsymbol{\mu}_h \| \boldsymbol{\mu}_h^0) \rangle \Big],$$
$$V_h^{\star,\lambda}(s) = \max_{a \in \mathcal{A}} Q_h^{\star,\lambda}(s,a). \tag{3.4}$$

A direct consequence of (3.4) is the optimal policy $\pi^{\star,\lambda} = \{\pi_h^{\star,\lambda}\}_{h=1}^H$ is the greedy policy with respect to the optimal robust Q-functions $\{Q_h^{\star,\lambda}\}_{h=1}^H$. Thus, in order to estimate $\pi^{\star,\lambda}$, it suffices to estimate $Q_h^{\star,\lambda}, \forall h \in [H]$.

**Offline dataset and learning goal.** An agent works with an offline dataset $\mathcal{D}$ with $K$ i.i.d. trajectories collected from the nominal environment by a behavior policy $\pi^b$. Specifically, for the $\tau$-th trajectory $\{(s_h^\tau, a_h^\tau, r_h^\tau)\}_{h=1}^H$, we have

$a_h^\tau \sim \pi_h^b(\cdot|s_h^\tau)$, $r_h^\tau = r_h(s_h^\tau, a_h^\tau)$, and $s_{h+1}^\tau \sim P_h^0(\cdot|s_h^\tau, a_h^\tau)$ for any $h \in [H]$. The agent aims to learn the optimal robust policy $\pi^\star$ from the offline dataset $\mathcal{D}$. Given a learned policy $\hat{\pi}$, we evaluate $\hat{\pi}$ by the suboptimality gap defined as follows

$$\text{SubOpt}(\hat{\pi}, s_1, \lambda) := V_1^{\star,\lambda}(s_1) - V_1^{\hat{\pi},\lambda}(s_1). \tag{3.5}$$

# 4. Robust Regularized Pessimistic Value Iteration (R2PVI)

In this section, we first develop a meta-algorithm for $d$-RRMDPs with general $f$-divergence defined regularization. To instantiate the meta-algorithm under specific $f$-divergences, we provide exact dual formulations of Q-functions with TV, KL and $\chi^2$-divergence defined regularization, respectively.

We first show that robust Q-functions admit linear representations under $d$-RRMDPs.

**Proposition 4.1.** Under Assumption 3.1, for any tuple $(\pi, s, a, h)$, we have $Q_h^{\pi,\lambda}(s,a) = \langle \phi(s,a), \boldsymbol{\theta}_h + \boldsymbol{w}_h^{\pi,\lambda} \rangle$, where $\boldsymbol{w}_h^{\pi,\lambda} = (w_{h,1}^{\pi,\lambda}, w_{h,2}^{\pi,\lambda}, \cdots, w_{h,d}^{\pi,\lambda})^\top \in \mathbb{R}^d$, and $w_{h,i}^{\pi,\lambda} := \inf_{\mu_{h,i} \in \Delta(\mathcal{S})}\big[\mathbb{E}^{\mu_{h,i}}[V_{h+1}^{\pi,\lambda}(s)] + \lambda D(\mu_{h,i}\|\mu_{h,i}^0)\big]$.

The linear representation of the robust Q-function enables linear function approximation for parameter estimation. The definition of parameter $\boldsymbol{w}_h^\lambda$ involves a regularized optimization. For any function $V : \mathcal{S} \to \mathbb{R}$, the dual formulation of the regularized optimization problem (Yang et al., 2023) is:

$$\inf_{\mu \in \Delta(S)} \mathbb{E}_{s \sim \mu} V(s) + \lambda D_f(\mu \| \mu^0)$$
$$= \sup_{\alpha \in R} \Big[ -\lambda \mathbb{E}_{s \sim \mu^0}\Big[f^*\Big(\frac{\alpha - V(s)}{\lambda}\Big)\Big] + \alpha \Big],$$

where $f^*$ is the conjugate function of $f$. We propose to estimate $w_h^\lambda$ through the ridge regression. We define the intermediate variable $w_{h,i}^\lambda(\alpha) := \mathbb{E}_{s \sim \mu^0}[f^*(\frac{\alpha - V(s)}{\lambda})]$ and obtain an estimation $\hat{w}_{h,i}^\lambda(\alpha) := \big[\operatorname{argmin}_{\boldsymbol{w} \in R^d} \sum_{\tau=1}^K (f^*(\frac{\alpha - \hat{V}_{h+1}^\lambda(s)}{\lambda}) - \phi(s_h^\tau, a_h^\tau)^\top \boldsymbol{w})^2 + \lambda \|\boldsymbol{w}\|_2^2\big]^i = \big[\boldsymbol{\Lambda}_h^{-1}[\sum_{\tau=1}^K \phi(s_h^\tau, a_h^\tau) f^*(\frac{\alpha - \hat{V}_{h+1}^\lambda(s)}{\lambda})]\big]^i$. We then estimate $w_{h,i}^\lambda$ by $\hat{w}_{h,i}^\lambda := \sup_{\alpha \in \mathbb{R}}\{-\lambda \hat{w}_{h,i}^\lambda(\alpha) + \alpha\}$. Leveraging Proposition 3.2 and the pessimism principle (Jin et al., 2021) developed to take account for the distribution shift arising from the offline dataset, we propose the meta-algorithm in Algorithm 1.

**Remark 4.2.** We emphasize that this general framework may encounter numerical challenges when computing the supremum over $\alpha$, especially depending on the choice of the divergence function $f$. In particular, the smoothness and curvature of the conjugate function $f^*$ can significantly affect the stability and efficiency of the optimization. For instance, some divergences lead to non-smooth or non-strongly convex conjugates, making the maximization problem harder to

---

**Algorithm 1** R2PVI under general $f$-divergence

---

**Require:** Dataset $\mathcal{D}$, Regularizer $\lambda > 0$
1: init $\hat{V}_{H+1}^{\lambda}(\cdot) = 0$
2: **for** episode $h = H, \cdots, 1$ **do**
3:      Compute $\mathbf{\Lambda}_h \leftarrow \sum_{\tau=1}^{K} \phi(s_h^{\tau}, a_h^{\tau}) \phi(s_h^{\tau}, a_h^{\tau})^{\top} + \gamma \mathbf{I}$
4:      $\hat{w}_{h,i}^{\lambda}(\alpha) \leftarrow \left[\mathbf{\Lambda}_h^{-1}[\sum_{\tau=1}^{K} \phi(s_h^{\tau}, a_h^{\tau}) f^*(\frac{\alpha - \hat{V}_{h+1}^{\lambda}(s)}{\lambda})]\right]^i$
         $\triangleright$ Duality Estimation for general $f$-divergence
5:      $\hat{w}_{h,i}^{\lambda} \leftarrow \sup_{\alpha \in \mathbb{R}}\{-\lambda \hat{w}_{h,i}^{\lambda}(\alpha) + \alpha\}$
6:      Construct the penalty $\Gamma_h(\cdot, \cdot)$.
7:      Estimate $\hat{Q}_h^{\lambda}(\cdot, \cdot) \leftarrow \min\{\langle \phi(\cdot, \cdot), \boldsymbol{\theta}_h + \hat{w}_h^{\lambda}\rangle - \Gamma_h(\cdot, \cdot), H - h + 1\}^+$.
8:      Construct $\hat{\pi}_h(\cdot|\cdot) \leftarrow \operatorname{argmax}_{\pi_h} \langle \hat{Q}_h^{\lambda}(\cdot, \cdot), \hat{\pi}_h(\cdot|\cdot)\rangle_{\mathcal{A}}$
         and $\hat{V}_h^{\lambda}(\cdot) \leftarrow \langle \hat{Q}_h^{\lambda}(\cdot, \cdot), \hat{\pi}_h(\cdot|\cdot)\rangle_{\mathcal{A}}$.
9: **end for**

---

solve accurately. Therefore, while this framework is general, we highlight that divergence-specific algorithm designs are necessary to ensure tractability and numerical stability.

Next, we instantiate the $f$-divergence with TV, KL and $\chi^2$-divergences respectively, and specify the estimation procedure corresponding to different divergences.

**4.1. R2PVI with the TV-Divergence**

In this section, we show how to get the estimation in Line 4 and Line 5 of Algorithm 1 for TV divergence defined regularization. We first present the following duality result.

**Proposition 4.3.** Given any probability measure $\mu^0 \in \Delta(\mathcal{S})$ and value function $V : \mathcal{S} \rightarrow [0, H]$, if the distance $D$ is chosen as the TV-divergence, the dual formulation of the original regularized optimization problem is formed as: $\inf_{\mu \in \Delta(S)} \mathbb{E}_{s \sim \mu} V(s) + \lambda D_{\mathrm{TV}}(\mu\|\mu^0) = \mathbb{E}_{s \sim \mu^0}[V(s)]_{V_{\min} + \lambda}$.

**Remark 4.4.** We compare the duality of the regularized problem in Proposition 4.3 with the duality of the constraint problem in DRMDPs with TV-divergence defined uncertainty sets (Shi & Chi, 2024): $\inf_{P \in \mathcal{U}_{\mathrm{TV}}^{\rho}(P^0)} \mathbb{E}^P V(s) = \max_{\alpha \in [V_{\min}, V_{\max}]} \left\{\mathbb{E}^{P^0}[V(s)]_{\alpha} - \rho(\alpha - \min_{s'}[V(s')]_{\alpha})\right\}$. The former has a *closed form*, while the later involves an optimization over the dual variable $\alpha$. We show later this distinction makes R2PVI much more computationally efficient compared to algorithms designed for DRMDPs.

Next, we present the parameter estimation procedure. Given an estimated robust value function $\hat{V}_{h+1}^{\lambda}$, we denote $\alpha_{h+1} = \min_{s'} \hat{V}_{h+1}^{\lambda}(s') + \lambda$. By the linear representation in Proposition 4.1, the duality for TV-divergence in Proposition 4.3 and the linearly structured nominal kernel in Assumption 3.1,

we estimate the parameter $\boldsymbol{w}_h^{\lambda}$ as follows

$$\hat{\boldsymbol{w}}_h^{\lambda} = \operatorname{argmin}_{\boldsymbol{w} \in R^d} \sum_{\tau=1}^{K} \left([\hat{V}_{h+1}^{\lambda}(s_{h+1}^{\tau})]_{\alpha_{h+1}}\right. \tag{4.1}$$
$$\left. - \phi(s_h^{\tau}, a_h^{\tau})^{\top} \boldsymbol{w}\right)^2 + \gamma \|\boldsymbol{w}\|_2^2,$$

where $\gamma$ is the regularizer in the ridge regression.

**Remark 4.5.** Thanks to the closed form expression of the duality for TV in Proposition 4.3, R2PVI does not need the dual optimization oracle as the DRPVI algorithm proposed for the $d$-DRMDP (Liu & Xu, 2024b, see their equation (4.4) and Algorithm 1 for more details). DRPVI needs to solve the dual optimization oracle separately for each dimension in each iteration, which is not necessary in our algorithm.

**4.2. R2PVI with the KL-Divergence**

Similar to the TV-divergence, we next derive the estimation in Line 4 and Line 5 of Algorithm 1 for KL divergence defined regularization. We first present the duality result.

**Proposition 4.6.** (Zhang et al., 2024, Example 1) Given any probability measure $\mu^0 \in \Delta(\mathcal{S})$ and value function $V : \mathcal{S} \rightarrow [0, H]$, if the probability divergence $D$ is chosen as the KL-divergence, then the dual formulation of the original regularized optimization problem is: $\inf_{\mu \in \Delta(S)} \mathbb{E}_{s \sim \mu} V(s) + \lambda D_{\mathrm{KL}}(\mu\|\mu^0) = -\lambda \log \mathbb{E}_{s \sim \mu^0}\left[e^{-V(s)/\lambda}\right]$.

The duality of KL also has a closed form. we will shown in Section 5 and Section 6 the closed form solution will reduce the computational cost and also ease the theoretical analysis. Next, we present the parameter estimation procedure. According to the linear representation of Q-functions in Proposition 4.1, the duality for KL-divergence in Proposition 4.6 and the linearly structured nominal kernel in Assumption 3.1, we estimate the parameter $\boldsymbol{w}_h^{\lambda}$ by a two-step procedure. Given an estimated robust value function $\hat{V}_{h+1}^{\lambda}$, we first estimate $\mathbb{E}_{s \sim \mu^0} e^{-\hat{V}_{h+1}^{\lambda}(s)/\lambda}$ by $\hat{\boldsymbol{w}}_h' = \operatorname{argmin}_{\boldsymbol{w} \in \mathbb{R}^d} \sum_{\tau=1}^{K} \left(e^{-\hat{V}_{h+1}^{\lambda}(s_{h+1}^{\tau})/\lambda} - \phi(s_h^{\tau}, a_h^{\tau})^{\top} \boldsymbol{w}\right)^2 + \gamma\|\boldsymbol{w}\|_2^2$. Then we take a log-transformation to get an estimation of $\boldsymbol{w}_h^{\lambda}$:

$$\hat{\boldsymbol{w}}_h^{\lambda} = -\lambda \log \max\{\hat{\boldsymbol{w}}_h', e^{-H/\lambda}\}. \tag{4.2}$$

Note that the max operator is to ensure the ridge-regression estimator is well-defined to take the log-transformation, and $e^{-H/\lambda}$ is the lower bound on $\mathbb{E}_{s \sim \mu^0} e^{-\hat{V}_{h+1}^{\lambda}(s)/\lambda}$.

**Remark 4.7.** The algorithm proposed by Ma et al. (2022) relies on dual optimization oracles under DRMDPs with KL divergence defined uncertainty sets, while our algorithm takes advantages of the closed-form duality solution. Their algorithm also relies on an additional value shift technique to guarantee the estimated parameter is well-defined to take the log-transformation, while our algorithm does not.

### 4.3. R2PVI with the $\chi^2$-Divergence

It remains to derive the estimation in Line 4 and Line 5 of Algorithm 1 for $\chi^2$-divergence defined regularization. We first present a result on the duality of the $\chi^2$-divergence.

**Proposition 4.8.** Given any probability measure $\mu^0 \in \Delta(\mathcal{S})$ and value function $V : \mathcal{S} \to [0, H]$, if $D$ is chosen as the $\chi^2$-divergence, the dual formulation of the original regularized optimization problem is:

$$\inf_{\mu \in \Delta(S)} \mathbb{E}_{s \sim \mu} V(s) + \lambda D_{\chi^2}(\mu \| \mu^0) = \quad (4.3)$$

$$\sup_{\alpha \in [V_{\min}, V_{\max}]} \left\{ \mathbb{E}_{s \sim \mu^0}[V(s)]_\alpha - \frac{1}{4\lambda} \operatorname{Var}_{s \sim \mu^0}[V(s)]_\alpha \right\}.$$

Next, we present the parameter estimation procedure. According to the linear representation of the Q-function in Proposition 4.1, the duality for $\chi^2$-divergence in Proposition 4.8 and the linear structure of the nominal kernel in Assumption 3.1, we estimate the parameter $\boldsymbol{w}_h^\lambda$ as follows. First, we propose a new method motivated by the variance estimation in Liu & Xu (2024b) to estimate the variance of the value function in (4.3). Specifically, given an estimated robust value function $\hat{V}_{h+1}^\lambda$ and dual variable $\alpha$, the estimations of $\mathbb{E}_{s \sim \mu^0}[\hat{V}_{h+1}^\lambda(s)]_\alpha$ and $\mathbb{E}_{s \sim \mu^0}[\hat{V}_{h+1}^\lambda(s)]_\alpha^2$ are:

$$\hat{\mathbb{E}}^{\mu_{h,i}^0}[\hat{V}_{h+1}^\lambda(s)]_\alpha = \left[ \underset{\boldsymbol{w} \in R^d}{\operatorname{argmin}} \sum_{\tau=1}^K ([\hat{V}_{h+1}^\lambda(s_{h+1}^\tau)]_\alpha \right.$$
$$\left. - \boldsymbol{\phi}(s_h^\tau, a_h^\tau)^\top \boldsymbol{w})^2 + \gamma \|\boldsymbol{w}\|_2^2 \right]_{[0,H]}^i, \quad (4.4)$$

$$\hat{\mathbb{E}}^{\mu_{h,i}^0}[\hat{V}_{h+1}^\lambda(s)]_\alpha^2 = \left[ \underset{\boldsymbol{w} \in R^d}{\operatorname{argmin}} \sum_{\tau=1}^K ([\hat{V}_{h+1}^\lambda(s_{h+1}^\tau)]_\alpha^2 \right.$$
$$\left. - \boldsymbol{\phi}(s_h^\tau, a_h^\tau)^\top \boldsymbol{w})^2 + \gamma \|\boldsymbol{w}\|_2^2 \right]_{[0,H^2]}^i, \quad (4.5)$$

where the superscript $i$ represents the $i$-th element of a vector. Then we construct the estimator $\hat{\boldsymbol{w}}_h^\lambda$ element-wisely:

$$\hat{w}_{h,i}^\lambda = \underset{\alpha \in [(\hat{V}_{h+1}^\lambda)_{\min}, (\hat{V}_{h+1}^\lambda)_{\max}]}{\max} \left\{ \hat{\mathbb{E}}^{\mu_{h,i}^0}[\hat{V}_{h+1}^\lambda(s)]_\alpha + \right.$$
$$\left. \frac{1}{4\lambda}(\hat{\mathbb{E}}^{\mu_{h,i}^0}[\hat{V}_{h+1}^\lambda(s)]_\alpha)^2 - \frac{1}{4\lambda}\hat{\mathbb{E}}^{\mu_{h,i}^0}[\hat{V}_{h+1}^\lambda(s)]_\alpha^2 \right\}. \quad (4.6)$$

We note that the above parameter estimation procedure involves an optimization, which is distinct from that of TV and KL, since the duality of $\chi^2$ does not admit a closed form expression. Specifically, it estimates the parameter $\boldsymbol{w}_h^\lambda$ element-wisely. For each dimension, it solves an optimization problem over an estimated dual formulation. This parameter estimation procedure shares a similar spirit with that in $d$-DRMDPs with TV divergence defined uncertainty sets (Liu & Xu, 2024b; Wang et al., 2024a).

To conclude, we summarize the TV, KL and $\chi^2$ divergences instantiation of Algorithm 1 in Algorithm 2.

## 5. Suboptimality Analysis

In this section, we establish theoretical guarantees for algorithms proposed in Section 4. First, we derive instance-dependent upper bounds on the suboptimality gap of policies learned by the instantiated algorithms. Next, under a partial coverage assumption on the offline dataset, we present instance-independent upper bounds for the suboptimality gap and compare them with results from previous works. Finally, we provide an information-theoretic lower bound to highlight the intrinsic characteristics of offline $d$-RRMDPs.

---

**Algorithm 2** R2PVI under TV, KL and $\chi^2$ divergence

---

**Require:** Dataset $\mathcal{D}$, Regularizer $\lambda > 0$
1: init $\hat{V}_{H+1}^\lambda(\cdot) = 0$
2: **for** episode $h = H, \cdots, 1$ **do**
3:      Compute $\boldsymbol{\Lambda}_h \leftarrow \sum_{\tau=1}^K \boldsymbol{\phi}(s_h^\tau, a_h^\tau)\boldsymbol{\phi}(s_h^\tau, a_h^\tau)^\top + \gamma \mathbf{I}$
4:      Obtain the parameter estimation $\hat{\boldsymbol{w}}_h^\lambda$ as follows:
        TV-divergence: use (4.1)
        KL-divergence: use (4.2)     ▷ Duality Estimation
        $\chi^2$-divergence: use (4.6)
5:      Construct the penalty $\Gamma_h(\cdot, \cdot)$.     ▷ Pessimism
6:      Estimate $\hat{Q}_h^\lambda(\cdot, \cdot) \leftarrow \min\{\langle \boldsymbol{\phi}(\cdot, \cdot), \boldsymbol{\theta}_h + \hat{\boldsymbol{w}}_h^\lambda \rangle - \Gamma_h(\cdot, \cdot), H - h + 1\}^+$.
7:      Construct $\hat{\pi}_h(\cdot|\cdot) \leftarrow \operatorname{argmax}_{\pi_h} \langle \hat{Q}_h^\lambda(\cdot, \cdot), \hat{\pi}_h(\cdot|\cdot) \rangle_\mathcal{A}$ and $\hat{V}_h^\lambda(\cdot) \leftarrow \langle \hat{Q}_h^\lambda(\cdot, \cdot), \hat{\pi}_h(\cdot|\cdot) \rangle_\mathcal{A}$.
8: **end for**

---

### 5.1. Instance-Dependent Upper Bound

**Theorem 5.1.** Suppose Assumption 3.1 holds. We set $\gamma = 1$ and $\Gamma_h(s, a) = \beta \sum_{i=1}^d \|\phi_i(\cdot, \cdot)\mathbf{1}_i\|_{\boldsymbol{\Lambda}_h^{-1}}$ in Algorithm 2. Let $\delta \in (0, 1)$. $\beta$ is chosen as follows.

- (TV) $\beta = 16Hd\sqrt{\xi_{\mathrm{TV}}}$,
- (KL) $\beta = 16d\lambda e^{H/\lambda}\sqrt{(H/\lambda + \xi_{\mathrm{KL}})}$,
- ($\chi^2$) $\beta = 8dH^2(1 + 1/\lambda)\sqrt{\xi_{\chi^2}}$,

where $\xi_{\mathrm{TV}} = 2\log(1024Hd^{1/2}K^2/\delta)$, $\xi_{\mathrm{KL}} = \log(1024d\lambda^2 K^3 H/\delta)$ and $\xi_{\chi^2} = \log(192K^5 H^6 d^3(1 + H/2\lambda)^3/\delta)$. Then with probability at least $1 - \delta$, for any $s \in \mathcal{S}$, we have $\operatorname{SubOpt}(\hat{\pi}, s, \lambda) \leq 2\beta \sup_{P \in \mathcal{U}^\lambda(P^0)} \sum_{h=1}^H \mathbb{E}^{\pi^\star, P}[\sum_{i=1}^d \|\phi_i(s, a)\mathbf{1}_i\|_{\boldsymbol{\Lambda}_h^{-1}} | s_1 = s]$, where $\mathcal{U}^\lambda(P^0)$ is the robustly admissible set defined as

$$\mathcal{U}^\lambda(P^0) = \bigotimes_{(h,s,a) \in [H] \times \mathcal{S} \times \mathcal{A}} \mathcal{U}_h^\lambda(s, a; \boldsymbol{\mu}_h^0), \quad (5.1)$$

and $\mathcal{U}_h^\lambda(s, a; \boldsymbol{\mu}_h^0) = \{ \sum_{i=1}^d \phi_i(s, a)\mu_{h,i}(\cdot) : D(\mu_{h,i} \| \mu_{h,i}^0) \leq \max_{s \in \mathcal{S}} V_{h+1}^{\star, \lambda}(s)/\lambda, \forall i \in [d] \}$.

**Remark 5.2.** Theorem 5.1 provides instance-dependent upper bounds on the suboptimality gap, closely resembling the bounds established for algorithms tailored to $d$-DRMDPs with TV divergence defined uncertainty sets (Liu & Xu,

2024b; Wang et al., 2024a). Notably, $\mathcal{U}^\lambda(P^0)$ in Theorem 5.1 represents a subset of the feasible set $\mathcal{F}_L$ in the $d$-RRMDP. While the RRMDP framework does not impose explicit uncertainty set constraints, this term naturally arises from our theoretical analysis (see Lemma C.1 and its proof for details). Specifically, we show that only distributions within $\mathcal{U}^\lambda(P^0)$ are relevant when considering the infimum in the robust regularized value and Q-functions (3.1). Intuitively, the regularization term in (3.1) should not exceed the change in expected cumulative rewards, so it should be upper bounded by the optimal value function. Similar terms are also found in Zhang et al. (2024, Definition 1) and Panaganti et al. (2022, Assumption 1).

### 5.2. Instance-Independent Upper Bound

Next, we derive instance-independent upper bounds on the suboptimality gap, building on Theorem 5.1. To achieve this, we adapt the robust partial coverage assumption on the offline dataset, originally proposed for $d$-DRMDPs (Assumption A.2 of Blanchet et al. (2024)). This adaptation is straightforward and involves replacing the uncertainty set in the $d$-DRMDP framework with the robustly admissible set defined in (5.1).

**Assumption 5.3** (Robust Regularized Partial Coverage). For the offline dataset $\mathcal{D}$, we assume that there exists some constant $c^\dagger > 0$, such that $\forall (h, s, P) \in [H] \times \mathcal{S} \times \mathcal{U}^\lambda(P^0)$,

$$\mathbf{\Lambda}_h \succeq \gamma \mathbf{I} + K \cdot c^\dagger \cdot \mathbb{E}^{\pi^\star, P}\big[\phi_i^2(s, a)\mathbf{1}_i\mathbf{1}_i^\top | s_1 = s\big].$$

Intuitively, Assumption 5.3 assumes that the offline dataset has good coverage on the $(s, a)$-space visited by the optimal robust policy $\pi^\star$ under any transition kernel in the robustly admissible set. With Assumption 5.3 and Theorem 5.1, we present instance-independent bounds as follows.

**Corollary 5.4.** Under the same setting as Theorem 5.1, if we further assume Assumption 5.3 holds, then for any $\delta \in (0, 1)$ and $s \in \mathcal{S}$, with probability at least $1 - \delta$, we have

- (TV) $\text{SubOpt}(\hat{\pi}, s, \lambda) \le 16H^2d^2\sqrt{\xi_{\text{TV}}}/\sqrt{c^\dagger K}$;
- (KL) $\text{SubOpt}(\hat{\pi}, s, \lambda) \le 16\lambda e^{\frac{H}{\lambda}}d^2H(\frac{H}{\lambda}+\xi_{\text{KL}})^{\frac{1}{2}}/\sqrt{c^\dagger K}$;
- ($\chi^2$) $\text{SubOpt}(\hat{\pi}, s, \lambda) \le 8d^2H^3(1+1/\lambda)\sqrt{\xi_{\chi^2}}/\sqrt{c^\dagger K}$.

We compare Algorithm 2 with algorithms proposed in previous works for the offline $d$-DRMDP in Table 1. For the case with TV-divergence, the suboptimality bound of R2PVI matches that of P2MPO (Blanchet et al., 2024) in terms of $d$ and $H$. DRPVI (Liu & Xu, 2024b) and DROP (Wang et al., 2024a) admit tighter bounds on the suboptimality gap, simply because their bounds are derived based on advanced techniques, such as reference-advantage decomposition (Xiong et al., 2022). We remark that our analysis can be tailored to adopt the same techniques and assumption, and thus get tighter bounds.

For the case with KL-divergence, existing theoretical results (Ma et al., 2022; Blanchet et al., 2024) rely on an additional regularity assumption regarding the KL dual variable, stating that the optimal dual variable for the KL duality admits a positive lower bound $\underline{\beta}$ under any feasible transition kernel (see Blanchet et al. (2024, Assumption F.1)). However, this assumption presents the following drawbacks. First, it is challenging to verify the assumption's validity in practice; second, even if such a lower bound holds, there is no straightforward method to determine the magnitude of the lower bound. It can be seen from Table 1 that the suboptimality bound of R2PVI matches that of DRVI-L (Ma et al., 2022) in terms of $d$ and $H$. However, our result depends on $\lambda$ which is the regularization parameter and can be arbitrarily chosen, while the result of DRVI-L depends on $\underline{\beta}$ which can be extremely small such that $\sqrt{\underline{\beta}}e^{H/\underline{\beta}} \gg \sqrt{\lambda}e^{H/\lambda}$. Moreover, Zhang et al. (2024)[2] studied the RRMDP with the regularization term defined by the KL-divergence in their Theorem 5 and the suboptimality bound also depends on the term $\sqrt{\lambda}e^{H/\lambda}$. Further, comparing the bounds of P2MPO (Blanchet et al., 2024) and R2PVI, we can qualitatively conclude that the regularization parameter $\lambda$ in $d$-RRMDPs plays a role analogous to $1/\rho$ in $d$-DRMDPs. This relation aligns with the intuition that a smaller $\lambda$ in $d$-RRMDPs or a larger $\rho$ in $d$-DRMDPs can induce a more robust policy.

For the case with $\chi^2$ divergence, our bound is the first result in literature. Compared with the TV divergence, the complexity is higher due to the more complex geometry and dual formulation of $\chi^2$ divergence. This observation aligns with findings of tabular DRMDPs with TV and $\chi^2$ divergence defined uncertainty sets (Shi et al., 2024). While existing works have focused on the $(s, a)$-rectangular structured regularization, our work fills the theoretical gap in RRMDPs by introducing the $d$-rectangular structured regularization, a contribution that may be of independent interest.

### 5.3. Information-Theoretic Lower Bound

We highlight that in Theorem 5.1, suboptimality bounds under cases with TV, KL, $\chi^2$-divergence share the same term $\sup_{P \in \mathcal{U}^\lambda(P^0)} \sum_{h=1}^H \mathbb{E}^{\pi^\star, P}\big[\sum_{i=1}^d \|\phi_i(s_h, a_h)\mathbf{1}_i\|_{\mathbf{\Lambda}_h^{-1}} | s_1 = s\big]$. In this section, we establish information theoretic lower bounds to show that this term is intrinsic in offline $d$-RRMDPs.

In order to give a formal presentation of the information-theoretical lower bound, we define $\mathcal{M}$ as a class of $d$-RRMDPs and $\text{SubOpt}(M, \hat{\pi}, s, \rho)$ as the suboptimality gap specific to one $d$-RRMDP instance $M \in \mathcal{M}$. We state the information-theoretic lower bound in the following theorem.

---

[2]Zhang et al. (2024) studied the infinite horizon RRMDP with a discounted factor $\gamma$, we replace the effective horizon length $\frac{1}{1-\gamma}$ by the horizon length $H$ in the finite horizon setting.

*Table 1.* Comparison of the suboptimality gap between this and previous works. The $\star$ symbol denotes results that require an additional assumption (Assumption 4.4 of Ma et al. (2022) and Assumption F.1 of Blanchet et al. (2024)) on the KL dual variable, an assumption not required by our R2PVI algorithm. The parameter $\rho$ represents the uncertainty level in DRMDPs, while $\lambda$ represents the regularization term in RRMDPs. The Coverage column indicates the assumption used to derive the suboptimality gap: the robust partial coverage assumption refers to Assumption A.2 of Blanchet et al. (2024), and the regularized partial coverage assumption represents Assumption 5.3.

| Algorithm | Setting | Divergence | Coverage | Suboptimality Gap |
|---|---|---|---|---|
| DRPVI (Liu & Xu, 2024b) | $d$-DRMDP | TV | full | $\tilde{O}(dH^2 K^{-1/2})$ |
| DROP (Wang et al., 2024a) | $d$-DRMDP | TV | robust partial | $\tilde{O}(d^{3/2} H^2 K^{-1/2})$ |
| P2MPO (TV) (Blanchet et al., 2024) | $d$-DRMDP | TV | robust partial | $\tilde{O}(d^2 H^2 K^{-1/2})$ |
| R2PVI-TV **(ours)** | $d$-RRMDP | TV | regularized partial | $\tilde{O}(d^2 H^2 K^{-1/2})$ |
| DRVI-L (Ma et al., 2022) | $d$-DRMDP | KL | robust partial | $\tilde{O}(\sqrt{\underline{\beta}} e^{H/\underline{\beta}} d^2 H^{3/2} K^{-1/2})^{\star}$ |
| P2MPO (KL) (Blanchet et al., 2024) | $d$-DRMDP | KL | robust partial | $\tilde{O}(e^{H/\underline{\beta}} d^2 H^2 \rho^{-1} K^{-1/2})^{\star}$ |
| R2PVI-KL **(ours)** | $d$-RRMDP | KL | regularized partial | $\tilde{O}(\sqrt{\lambda} e^{H/\lambda} d^2 H^{3/2} K^{-1/2})$ |
| R2PVI-$\chi^2$ **(ours)** | $d$-RRMDP | $\chi^2$ | regularized partial | $\tilde{O}(d^2 H^3 (1+\lambda^{-1}) K^{-1/2})$ |

**Theorem 5.5.** Let $K > \max\{\tilde{O}(d^6), \tilde{O}(d^3 H^2/\lambda^2)\}$ be the sample size, where we have regularizer $\lambda$, dimension $d$, horizon length $H$. There exists a class of $d$-rectangular linear RRMDPs $\mathcal{M}$ and an offline dataset $\mathcal{D}$ of size $K$ such that for any $\delta \in (0,1)$, $s \in \mathcal{S}$, divergence $D$ among $D_{\mathrm{TV}}, D_{\mathrm{KL}}$ and $D_{\chi^2}$, with probability at least $1 - \delta$, we have $\inf_{\hat{\pi}} \sup_{M \in \mathcal{M}} \mathrm{SubOpt}(M, \hat{\pi}, s, \lambda, D) \geq c \cdot \sup_{P \in \mathcal{U}^{\lambda}(P^0)} \sum_{h=1}^{H} \mathbb{E}^{\pi^\star, P}[\sum_{i=1}^{d} \|\phi_i(s_h, a_h) \mathbf{1}_i\|_{\mathbf{\Lambda}_h^{-1}} | s_1 = s]$, where $c$ is a universal constant.

Theorem 5.5 is a universal information theoretic lower bound for $d$-RRMDPs with all three divergences studied in Section 5. Theorem 5.5 shows that the instance-dependent term is actually intrinsic to the offline $d$-RRMDPs, and Algorithm 2 is near-optimal up to a factor $\beta$, for which the definition varies among different divergence metric $D$ as shown in Theorem 5.1. The proof outline of Theorem 5.5 is inspired by that of Theorem 6.1 in Liu & Xu (2024b), but here we need careful treatment on bounding the robust regularized value function by duality under different choices of $f$-divergences. We provide more details on the hard instance construction, the proof techniques, and the comparison with existing results in Appendix D.

## 6. Experiment

In this section, we conduct numerical experiments to explore (1) the robustness of R2PVI regarding dynamics shifts, (2) how the regularizer $\lambda$ affects the robustness of R2PVI, and (3) the computation cost of R2PVI. We evaluate our algorithm in two off-dynamics problems that have

been used in the literature (Ma et al., 2022; Liu & Xu, 2024a). All experiments are conducted on a machine with an 11th Gen Intel(R) Core(TM) i5-11300H @ 3.10GHz processor, featuring 8 logical CPUs, 4 physical cores, and 2 threads per core. The implementation of our R2PVI algorithm is available at https://github.com/panxulab/Robust-Regularized-Pessimistic-Value-Iteration.

**Baselines.** We compare our algorithms with three types of baseline frameworks: (1) non-robust pessimism-based algorithm: PEVI (Jin et al., 2021), (2) algorithms for $d$-DRMDPs with TV divergence defined uncertainty sets: DR-PVI (Liu & Xu, 2024b), (3) algorithms for $d$-DRMDPs with KL divergence defined uncertainty sets: DRVI-L (Ma et al., 2022). We do not implement P2MPO and DROP mentioned in Table 1 in our experiment, due to the lack of code base and numerical experiment in their works.

### 6.1. Simulated Linear MDPs

We borrow the simulated linear MDP constructed in Liu & Xu (2024a) and adapt it to the offline setting. We set the behavior policy $\pi^b$ such that it chooses actions uniformly at random. The sample size of the offline dataset is set to 100. For completeness, we present more details on the experiment set up and results in Appendix A.

In Figure 1(a), we compare R2PVI with its non-robust counterpart PEVI (Jin et al., 2021). We conclude that PEVI outperforms R2PVI when the perturbation of the environment is small, but underperforms when the environment encounters a significant shift, which verifies the robustness

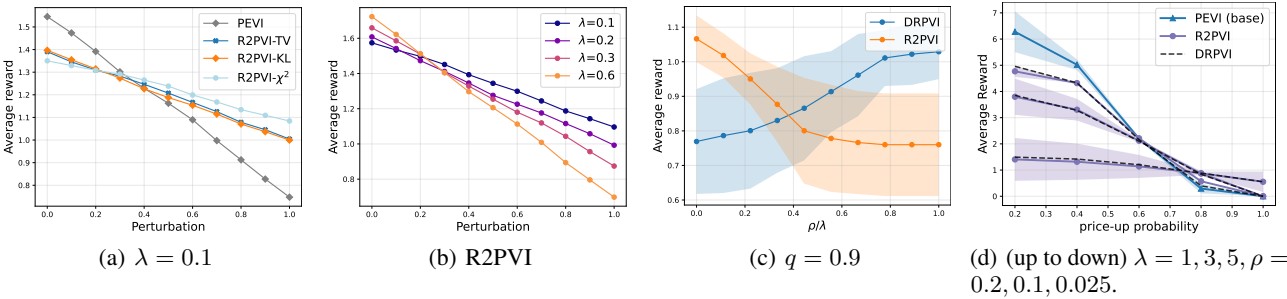

(a) $\lambda = 0.1$      (b) R2PVI      (c) $q = 0.9$      (d) (up to down) $\lambda = 1, 3, 5, \rho = 0.2, 0.1, 0.025$.

*Figure 1.* Simulated results for linear MDP. In Figure 1(a) and Figure 1(b), the $x$-axis refers to the perturbation in the testing environment. In Figure 1(c), the $x$-axis represents different robust level $\rho$ and regularized penalty $\lambda$, respectively. Figure 1(d) shows the robustness of algorithms under different robust level $\rho$ (DRPVI) or regularization penalty $\lambda$ (R2PVI).

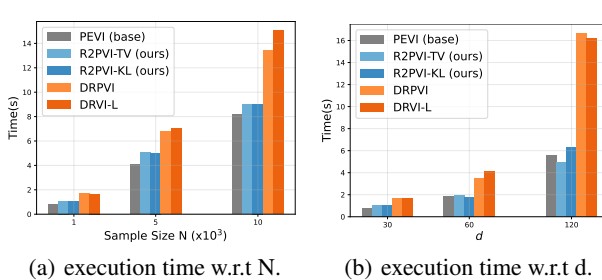

(a) execution time w.r.t N.      (b) execution time w.r.t d.

*Figure 2.* Simulation results for the simulated American put option task. Figure 2(a) shows the computation time of R2PVI with respect to the sample size $N$. Figure 2(b) shows the computation time of algorithms with respect to the feature dimension $d$.

of R2PVI. The regularizer $\lambda$ controls the extent of robustness of R2PVI by determining the magnitude of the penalty as shown in Proposition 3.2. By Figure 1(b), we conclude that a smaller $\lambda$ leads to a more robust policy. To illustrate the relation between the $d$-RRMDP and the $d$-DRMDP, we fix a target environment, and then test R2PVI with different $\lambda$ and DRPVI (Liu & Xu, 2024b) with different $\rho$. We find from Figure 1(c) that the ranges of the average reward are about the same for the two algorithms, though the behaviors w.r.t. $\lambda$ and $\rho$ are opposite. Thus, we verify that the regularizer $\lambda$ plays a similar role in the RRMDP as the inverted robustness parameter $1/\rho$ in the DRMDP.

### 6.2. Simulated American Put Option

In this section, we test our algorithm in a simulated American Put Option environment (Tamar et al., 2014; Zhou et al., 2021) that does not belong to the $d$-rectangular linear RRMDP. This environment is a finite horizon MDP with $H = 20$, and is controlled by a hyperparameter $p_0$, which is set to be 0.5 in the nominal environment. We collect the offline data from the nominal environment by a uniformly random behavior policy. An agent uses the collected offline

dataset to learn a policy which decides at each state whether or not to exercise the option. To implement our algorithm, we use a manually designed feature mapping of dimension $d$. For more details, we refer readers to Appendix A.2.

All experiment results are shown in Figure 2. In particular, from Figure 2(a) and Figure 2(b), we can conclude that the computation cost of R2PVI is as low as its non-robust counterpart PEVI (Jin et al., 2021), and improves that of DRPVI (Liu & Xu, 2024b) and DRVI-L (Ma et al., 2022) designed for the $d$-DRMDP. This is due to the closed form duality of TV and KL under the $d$-RRMDP framework. From Figure 1(d), we conclude that R2PVI not only demonstrates robustness to environment perturbations but also matches DRPVI's performance for appropriate values of the robust regularizer $\lambda$ and uncertainty level $\rho$.

## 7. Conclusion

We introduced the $d$-rectangular linear Robust Regularized Markov Decision Process ($d$-RRMDP) framework to address limitations of the $d$-rectangular DRMDP framework for robust policy learning in literature, improving both theoretical robustness and computational efficiency. We developed R2PVI, a provably effective algorithm for learning robust policies from offline datasets using $f$-divergence-based regularization. Our results highlight the advantages of $d$-RRMDPs, particularly in simplifying the duality oracle. Experiments confirm R2PVI's robustness and efficiency. It remains an intriguing open question to improve the current upper and lower bounds to study the fundamental hardness of $d$-RRMDPs.

## Impact Statement

This paper presents work whose goal is to advance the field of Machine Learning. There are many potential societal consequences of our work, none which we feel must be specifically highlighted here.

## Acknowledgements

Z. Liu and P. Xu are supported in part by the National Science Foundation (DMS-2323112) and the Whitehead Scholars Program at the Duke University School of Medicine.

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

# A. Additional Details on Experiments

In this section, we provide details on experiment setup.

## A.1. Simulated Linear MDPs

**Construction of the Simulated Linear MDP**   We leverage the simulated linear MDP instance proposed by Liu & Xu (2024a). The state space is $\mathcal{S} = \{x_1, \cdots, x_5\}$ and the action space is $\mathcal{A} = \{-1, 1\}^4 \subset \mathbb{R}^4$. At each episode, the initial state is always $x_1$. From $x_1$, the next state can be $x_2, x_4, x_5$ with probability defined on the arrows. Both $x_4$ and $x_5$ are absorbing states. $x_4$ is the fail state with 0 reward and $x_5$ is the goal state with reward 1. The hyperparameter $\boldsymbol{\xi} \in \mathbb{R}^4$ is designed to determine the reward functions and transition probabilities and $\delta$ is the parameter defined to determine the environment. We perturb the transition probability at the initial stage to construct the source environment. The extend of perturbation is controlled by the hyperparameter $q \in (0, 1)$. For more details on the simulated linear DRMDP, we refer readers to the Supplementary A.1 in Liu & Xu (2024a).

**Hyperparameters**   The hyper-parameters in our setting are shown in Table 2. The horizon is 3, the $\beta, \gamma, \delta$ are set the same in all tasks, the $\|\boldsymbol{\xi}\|_1$ is set as 0.3, 0.2, 0.1 in Figure 1 in order to illustrate the versatility of our algorithms.

*Table 2.* Hyper-parameters.

| Hyper-parameters | Value |
| --- | --- |
| $H$ (Horizon) | 3 |
| $\beta$ (pessimism parameter) | 1 |
| $\gamma$ | 0.1 |
| $\delta$ | 0.3 |
| $\|\boldsymbol{\xi}\|_1$ | 0.3, 0.2, 0.1 |

## A.2. Simulated American Put Option

**Construction of the Simulated American Put Option**   In each episode, there are $H = 20$ stages, and each state $h$, the dynamics evolves following the Bernoulli distribution:

$$s_{h+1} = \begin{cases} 1.02 s_h, & \text{w.p } p_0 \\ 0.98 s_h, & \text{w.p } 1 - p_0 \end{cases}, \tag{A.1}$$

where $p_0 \in (0, 1)$ is the probability of price up. At each step, the agent has two actions to take: exercise the option $a_h = 1$ or not exercise $a = 0$. If exercising the option $a_h = 0$, the agent will obtain reward $r_h = \max\{0, 100 - s_h\}$ and the state comes to an end. If not exercising the option $a_h = 1$, The state will continue to transit based on (A.1) and no reward will be received. To implement our algorithms, we use the following feature mapping:

$$\phi(s_h, a) = \begin{cases} [\varphi_1(s_h), \cdots, \varphi_d(s_h), 0] & \text{if } a = 1 \\ [0, \cdots, 0, \max\{0, 100 - s_h\}] & \text{if } a = 0 \end{cases},$$

where $\varphi_i(s) = \max\{0, 1 - |s_h - s_i| / \Delta\}$, $\{s_i\}_{i=1}^d$ are anchor states, $s_1 = 80$ $s_{i+1} - s_i = \Delta$ and $\Delta = 60/d$. For more details on the simulated American put option environment, we refer readers to the Appendix C of Ma et al. (2022).

**Offline Dataset and Hyperparameters**   We set $p_0 = 0.5$ in the nominal environment, from which trajectories are collected by fixed behavior policy, which chooses $a_h = 0$. The $\beta = 0.1$ and $\gamma = 1$ are set hyper-parameters in all tasks. For the time efficiency comparison in Figure 2(a) and Figure 2(b), we counted the time it took for the agent to train once and repeated 5 times to take the average.

# B. Proof of Properties of $d$-RRMDPs

In this section, we provide the proofs of results in Sections 3 and 4, namely, the robust regularized Bellman equation, the existence of the optimal robust policy, and the linear representation of the robust regularized Q-function under the

$d$-rectangular linear RRMDP.

## B.1. Proof of Proposition 3.2

*Proof.* We prove the a stronger proposition by induction from the last stage $H$. Specifically, besides the equations in Proposition 3.2 hold, we further assume that there exist transition kernels $\{\hat{\mu}_t\}_{t=1}^H$, $\hat{P}_t = \langle \phi, \hat{\mu}_t \rangle$, such that for any $(h, s) \in [H] \times \mathcal{S}$,

$$V_h^{\pi,\lambda}(s) = \mathbb{E}^{\{\hat{P}_t\}_{t=h}^H}\left[\sum_{t=h}^H \left[r_t(s_t, a_t) + \lambda\langle\phi(s_t, a_t), \boldsymbol{D}(\hat{\mu}_t||\boldsymbol{\mu}_t^0)\rangle\right| s_h = s, \pi\right]. \tag{B.1}$$

As there is no transitional kernel involved, the base case holds trivially. Suppose the conclusion holds for stage $h + 1$, that is to say, there exists $\hat{P}_t, t = h+1, h+2, \cdots, H$ such that

$$V_{h+1}^{\pi,\lambda}(s) = \mathbb{E}^{\{\hat{P}_i\}_{i=h+1}^H}\left[\sum_{t=h+1}^H \left[r_t(s_t, a_t) + \lambda\langle\phi(s_t, a_t), \boldsymbol{D}(\hat{\mu}_t||\boldsymbol{\mu}_t^0)\rangle\right| s_{h+1} = s, \pi\right].$$

For the case of $h$, recall the definition of $Q_h^\pi$, we have

$$\begin{aligned}
Q_h^{\pi,\lambda}(s,a) &= \inf_{\boldsymbol{\mu}_t \in \Delta(\mathcal{S})^d, P_t = \langle\phi,\boldsymbol{\mu}_t\rangle} \mathbb{E}^{\{P_t\}_{t=h}^H}\left[\sum_{t=h}^H \left[r_t(s_t, a_t) + \lambda\langle\phi(s_t, a_t), \boldsymbol{D}(\boldsymbol{\mu}_t||\boldsymbol{\mu}_t^0)\rangle\right| s_h = s, a_h = a, \pi\right] \\
&= r_h(s,a) + \inf_{\boldsymbol{\mu}_t \in \Delta(\mathcal{S})^d, P_t = \langle\phi,\boldsymbol{\mu}_t\rangle} \lambda\langle\phi(s_h, a_h), \boldsymbol{D}(\boldsymbol{\mu}_h||\boldsymbol{\mu}_h^0)\rangle \\
&\quad + \int_{\mathcal{S}} P_h(ds'|s,a)\mathbb{E}^{\{P_t\}_{t=h+1}^H}\left[\sum_{t=h+1}^H \left[r_t(s_t, a_t) + \lambda\langle\phi(s_t, a_t), \boldsymbol{D}(\boldsymbol{\mu}_t||\boldsymbol{\mu}_t^0)\rangle\right| s_{h+1} = s', \pi\right] \\
&\leq r_h(s,a) + \inf_{\boldsymbol{\mu}_h \in \Delta(\mathcal{S})^d, P_h = \langle\phi,\boldsymbol{\mu}_h\rangle} \lambda\langle\phi(s_h, a_h), \boldsymbol{D}(\boldsymbol{\mu}_h||\boldsymbol{\mu}_h^0)\rangle \\
&\quad + \int_{\mathcal{S}} P_h(ds'|s,a)\mathbb{E}^{\{\hat{P}_t\}_{t=h+1}^H}\left[\sum_{t=h+1}^H \left[r_t(s_t, a_t) + \lambda\langle\phi(s_t, a_t), \boldsymbol{D}(\hat{\mu}_t||\boldsymbol{\mu}_t^0)\rangle\right| s_{h+1} = s', \pi\right] \\
&= r_h(s,a) + \inf_{\boldsymbol{\mu}_h \in \Delta(\mathcal{S})^d, P_h = \langle\phi,\boldsymbol{\mu}_h\rangle} \lambda\langle\phi(s_h, a_h), \boldsymbol{D}(\boldsymbol{\mu}_h||\boldsymbol{\mu}_h^0)\rangle + \mathbb{E}_{s'\sim P_h(\cdot|s,a)}[V_{h+1}^{\pi,\lambda}(s')], \tag{B.2}
\end{aligned}$$

where (B.2) follows by the inductive hypothesis of $V_{h+1}^{\pi,\lambda}(s)$. On the other hand, we can lower bound $Q_h^{\pi,\lambda}(s,a)$ as

$$\begin{aligned}
&Q_h^{\pi,\lambda}(s,a) \\
&= r_h(s,a) + \inf_{\boldsymbol{\mu}_t \in \Delta(\mathcal{S})^d, P_t = \langle\phi,\boldsymbol{\mu}_t\rangle} \lambda\langle\phi(s_h, a_h), \boldsymbol{D}(\boldsymbol{\mu}_h||\boldsymbol{\mu}_h^0)\rangle \\
&\quad + \int_{\mathcal{S}} P_h(ds'|s,a)\mathbb{E}^{\{P_t\}_{t=h+1}^H}\left[\sum_{t=h+1}^H \left[r_t(s_t, a_t) + \lambda\langle\phi(s_t, a_t), \boldsymbol{D}(\boldsymbol{\mu}_t||\boldsymbol{\mu}_t^0)\rangle\right| s_{h+1} = s', \pi\right] \\
&\geq r_h(s,a) + \inf_{\boldsymbol{\mu}_h \in \Delta(\mathcal{S})^d, P_h = \langle\phi,\boldsymbol{\mu}_h\rangle} \lambda\langle\phi(s_h, a_h), \boldsymbol{D}(\boldsymbol{\mu}_h||\boldsymbol{\mu}_h^0)\rangle \tag{B.3} \\
&\quad + \int_{\mathcal{S}} P_h^\pi(ds'|s,a) \inf_{\boldsymbol{\mu}_t \in \Delta(\mathcal{S})^d, P_t = \langle\phi,\boldsymbol{\mu}_t\rangle} \mathbb{E}^{\{P_t\}_{t=h+1}^H}\left[\sum_{t=h+1}^H \left[r_t(s_t, a_t) + \lambda\langle\phi(s_t, a_t), \boldsymbol{D}(\boldsymbol{\mu}_t||\boldsymbol{\mu}_t^0)\rangle\right| s_{h+1} = s', \pi\right] \\
&= r_h(s,a) + \inf_{\boldsymbol{\mu}_h \in \Delta(\mathcal{S})^d, P_h = \langle\phi,\boldsymbol{\mu}_h\rangle} \lambda\langle\phi(s_h, a_h), \boldsymbol{D}(\boldsymbol{\mu}_h||\boldsymbol{\mu}_h^0)\rangle + \mathbb{E}_{s'\sim P_h(\cdot|s,a)}[V_{h+1}^{\pi,\lambda}(s')], \tag{B.4}
\end{aligned}$$

where (B.3) follows by the Fatou's lemma, (B.4) follows by the definition of $V_{h+1}^{\pi,\lambda}(s)$. Hence, combining the two above inequalities, we conclude the proof of the first equation. Next we focus on the proof of the (B.1), by which we aim to proof the existence of transition kernel $\{\hat{P}_t\}_{t=h}^H$. By the fact that

$$Q_h^{\pi,\lambda}(s,a) = r_h(s,a) + \inf_{\boldsymbol{\mu}_h \in \Delta(\mathcal{S})^d, P_h = \langle\phi,\boldsymbol{\mu}_h\rangle} \left[\mathbb{E}_{s'\sim P_h(\cdot|s,a)}\left[V_{h+1}^{\pi,\lambda}(s')\right] + \lambda\langle\phi(s,a), \boldsymbol{D}(\boldsymbol{\mu}_h||\boldsymbol{\mu}_h^0)\rangle\right],$$

we notice that the $\inf$ problem above is constraint by the distance $D$. Therefore by Lagrange duality and the closeness of distribution $\Delta(\mathcal{S})$, there exists $\hat{\boldsymbol{\mu}}_h \in \Delta(\mathcal{S})^d, \hat{P}_h = \langle \boldsymbol{\phi}, \hat{\boldsymbol{\mu}}_h \rangle$ such that

$$Q_h^{\pi,\lambda}(s,a) = r_h(s,a) + \mathbb{E}_{s' \sim \hat{P}_h(\cdot|s,a)}\left[V_{h+1}^{\pi,\lambda}(s')\right] + \lambda \langle \boldsymbol{\phi}(s,a), \boldsymbol{D}(\hat{\boldsymbol{\mu}}_h || \boldsymbol{\mu}_h^0) \rangle. \tag{B.5}$$

Now it remains to proof (3.3). By the definition of $V_h^{\pi,\lambda}(s)$, we have

$$V_h^{\pi,\lambda}(s) \tag{B.6}$$

$$= \inf_{\boldsymbol{\mu}_t \in \Delta(\mathcal{S})^d, P_t = \langle \boldsymbol{\phi}, \boldsymbol{\mu}_t \rangle} \mathbb{E}^{\{P_t\}_{t=h}^H}\left[\sum_{t=h}^H \left[r_t(s_t,a_t) + \lambda \langle \boldsymbol{\phi}(s_t,a_t), \boldsymbol{D}(\boldsymbol{\mu}_t||\boldsymbol{\mu}_t^0) \rangle\right] \Big| s_h = s, \pi\right]$$

$$= \inf_{\boldsymbol{\mu}_t \in \Delta(\mathcal{S})^d, P_t = \langle \boldsymbol{\phi}, \boldsymbol{\mu}_t \rangle} \sum_{a \in \mathcal{A}} \pi(a|s) \mathbb{E}^{\{P_t\}_{t=h}^H}\left[\sum_{t=h}^H \left[r_t(s_t,a_t) + \lambda \langle \boldsymbol{\phi}(s_t,a_t), \boldsymbol{D}(\boldsymbol{\mu}_t||\boldsymbol{\mu}_t^0) \rangle\right] \Big| s_h = s, a_h = a, \pi\right]$$

$$\leq \sum_{a \in \mathcal{A}} \pi(a|s) \mathbb{E}^{\{\hat{P}_t\}_{t=h}^H}\left[\sum_{t=h}^H \left[r_t(s_t,a_t) + \lambda \langle \boldsymbol{\phi}(s_t,a_t), \boldsymbol{D}(\hat{\boldsymbol{\mu}}_t||\boldsymbol{\mu}_t^0) \rangle\right] \Big| s_h = s, a_h = a, \pi\right]$$

$$= \sum_{a \in \mathcal{A}} \pi(a|s) Q_h^{\pi,\lambda}(s,a), \tag{B.7}$$

where (B.7) comes from (B.5) and the inductive hypothesis. On the other hand, by the definition of $Q_h^{\pi,\lambda}(s,a)$, we have

$$\sum_{a \in \mathcal{A}} \pi(a|s) Q_h^{\pi,\lambda}(s,a)$$

$$= \sum_{a \in \mathcal{A}} \pi(a|s) \inf_{\boldsymbol{\mu}_t \in \Delta(\mathcal{S})^d, P_t = \langle \boldsymbol{\phi}, \boldsymbol{\mu}_t \rangle} \mathbb{E}^{\{P_t\}_{t=h}^H}\left[\sum_{t=h}^H \left[r_t(s_t,a_t) + \lambda \langle \boldsymbol{\phi}(s_t,a_t), \boldsymbol{D}(\boldsymbol{\mu}_t||\boldsymbol{\mu}_t^0) \rangle\right] \Big| s_h = s, a_h = a, \pi\right]$$

$$\leq \inf_{\boldsymbol{\mu}_t \in \Delta(\mathcal{S})^d, P_t = \langle \boldsymbol{\phi}, \boldsymbol{\mu}_t \rangle} \sum_{a \in \mathcal{A}} \pi(a|s) \mathbb{E}^{\{P_t\}_{t=h}^H}\left[\sum_{t=h}^H \left[r_t(s_t,a_t) + \lambda \langle \boldsymbol{\phi}(s_t,a_t), \boldsymbol{D}(\boldsymbol{\mu}_t||\boldsymbol{\mu}_t^0) \rangle\right] \Big| s_h = s, a_h = a, \pi\right]$$

$$= V_h^{\pi,\lambda}(s), \tag{B.8}$$

where (B.8) comes from the definition of $V_h^{\pi,\lambda}(s)$. Combining the two inequalities (B.7) and (B.8), we have

$$V_h^{\pi,\lambda}(s) = \mathbb{E}_{a \sim \pi(\cdot|s)}\left[Q_h^{\pi}(s,a)\right].$$

This proves the (3.3) for stage h. Therefore, by using an induction argument, we finish the proof of Proposition 3.2. □

## B.2. Proof of Proposition 3.3

*Proof.* We define the optimal stationary policy $\pi^\star = \{\pi_h^\star\}_{h=1}^H$ as: for all $(h,s) \in [H] \times \mathcal{S}$,

$$\pi_h^\star(s) = \underset{a \in \mathcal{A}}{\operatorname{argmax}}\left[r_h(s,a) + \inf_{\boldsymbol{\mu}_h \in \Delta(\mathcal{S})^d, P_h = \langle \boldsymbol{\phi}, \boldsymbol{\mu}_h \rangle}\left[\mathbb{E}_{s' \sim P_h(\cdot|s,a)}\left[V_{h+1}^{\star,\lambda}(s')\right] + \lambda \langle \boldsymbol{\phi}(s,a), \boldsymbol{D}(\boldsymbol{\mu}_h||\boldsymbol{\mu}_h^0) \rangle\right]\right].$$

Now it remains to show that the regularized robust value function $V_h^{\pi^\star,\lambda}, Q_h^{\pi^\star,\lambda}$ induced by policy $\pi^\star$ is optimal, i.e., for all $(h,s) \in [H] \times \mathcal{S}$,

$$V_h^{\pi^\star,\lambda}(s) = V_h^{\star,\lambda}(s), Q_h^{\pi^\star,\lambda}(s,a) = Q_h^{\star,\lambda}(s,a).$$

By the (3.2), we only need to prove the first equation above, then the optimality of the $Q$ holds trivially. we prove this statement by induction from $H$ to $1$. For stage $H$, the conclusion holds by:

$$V_H^{\star,\lambda}(s) = \sup_\pi V_H^{\pi,\lambda}(s)$$

$$= \sup_\pi \inf_{\boldsymbol{\mu}_H \in \Delta(\mathcal{S})^d, P_H = \langle \boldsymbol{\phi}, \boldsymbol{\mu}_H \rangle} \mathbb{E}^{P_H}\left[\left[r_H(s_H,a_H) + \lambda \langle \boldsymbol{\phi}(s_H,a_H), \boldsymbol{D}(\boldsymbol{\mu}_H||\boldsymbol{\mu}_H^0) \rangle\right] \Big| s_H = s, \pi\right]$$

$$= \sup_{\pi} \Big[ r_H(s_H, \pi_H(s_H)) + \inf_{\boldsymbol{\mu}_H \in \Delta(\mathcal{S})^d, P_H = \langle \boldsymbol{\phi}, \boldsymbol{\mu}_H \rangle} \lambda \langle \boldsymbol{\phi}(s, a), \boldsymbol{D}(\boldsymbol{\mu}_H || \boldsymbol{\mu}_H^0) \rangle \Big]$$

$$= V_H^{\pi^\star, \lambda}(s).$$

Now assume that the conclusion holds by stage $h + 1$. Hence, we have that for all $s \in \mathcal{S}$,

$$V_{h+1}^{\pi^\star, \lambda}(s) = V_{h+1}^{\star, \lambda}(s).$$

For the case of $h$, by (3.2), we have

$$V_h^{\pi^\star, \lambda}(s)$$
$$= \mathbb{E}_{a \sim \pi_h^\star(\cdot|s)} \Big[ Q_h^{\pi^\star, \lambda}(s, a) \Big]$$
$$= \mathbb{E}_{a \sim \pi_h^\star(\cdot|s)} \Big[ r_h(s, a) + \inf_{\boldsymbol{\mu}_h \in \Delta(\mathcal{S})^d, P_h = \langle \boldsymbol{\phi}, \boldsymbol{\mu}_h \rangle} \big[ \mathbb{E}_{s' \sim P_h(\cdot|s,a)} \big[ V_{h+1}^{\pi^\star, \lambda}(s') \big] + \lambda \langle \boldsymbol{\phi}(s, a), \boldsymbol{D}(\boldsymbol{\mu}_h || \boldsymbol{\mu}_h^0) \rangle \big] \Big]$$
$$= \mathbb{E}_{a \sim \pi_h^\star(\cdot|s)} \Big[ r_h(s, a) + \inf_{\boldsymbol{\mu}_h \in \Delta(\mathcal{S})^d, P_h = \langle \boldsymbol{\phi}, \boldsymbol{\mu}_h \rangle} \big[ \mathbb{E}_{s' \sim P_h(\cdot|s,a)} \big[ V_{h+1}^{\star, \lambda}(s') \big] + \lambda \langle \boldsymbol{\phi}(s, a), \boldsymbol{D}(\boldsymbol{\mu}_h || \boldsymbol{\mu}_h^0) \rangle \big] \Big] \tag{B.9}$$
$$= \max_{a \in \mathcal{A}} \Big[ r_h(s, a) + \inf_{\boldsymbol{\mu}_h \in \Delta(\mathcal{S})^d, P_h = \langle \boldsymbol{\phi}, \boldsymbol{\mu}_h \rangle} \big[ \mathbb{E}_{s' \sim P_h(\cdot|s,a)} \big[ V_{h+1}^{\star, \lambda}(s') \big] + \lambda \langle \boldsymbol{\phi}(s, a), \boldsymbol{D}(\boldsymbol{\mu}_h || \boldsymbol{\mu}_h^0) \rangle \big] \Big], \tag{B.10}$$

where (B.9) holds by the inductive hypothesis, (B.10) holds by the definition of $\pi^\star$. On the other hand, recall the definition of $V_h^{\star, \lambda}(s)$, then for any $s \in \mathcal{S}$, by (3.2) we have

$$V_h^{\star, \lambda}(s)$$
$$= \sup_{\pi} V_h^{\pi, \lambda}(s)$$
$$= \sup_{\pi} \mathbb{E}_{a \sim \pi_h(\cdot|s)} \Big[ Q_h^{\pi, \lambda}(s, a) \Big]$$
$$= \sup_{\pi} \mathbb{E}_{a \sim \pi_h(\cdot|s)} \Big[ r_h(s, a) + \inf_{\boldsymbol{\mu}_h \in \Delta(\mathcal{S})^d, P_h = \langle \boldsymbol{\phi}, \boldsymbol{\mu}_h \rangle} \big[ \mathbb{E}_{s' \sim P_h(\cdot|s,a)} \big[ V_{h+1}^{\pi, \lambda}(s') \big] + \lambda \langle \boldsymbol{\phi}(s, a), \boldsymbol{D}(\boldsymbol{\mu}_h || \boldsymbol{\mu}_h^0) \rangle \big] \Big]$$
$$\leq \sup_{\pi} \mathbb{E}_{a \sim \pi_h(\cdot|s)} \Big[ r_h(s, a) + \inf_{\boldsymbol{\mu}_h \in \Delta(\mathcal{S})^d, P_h = \langle \boldsymbol{\phi}, \boldsymbol{\mu}_h \rangle} \big[ \mathbb{E}_{s' \sim P_h(\cdot|s,a)} \big[ V_{h+1}^{\star, \lambda}(s') \big] + \lambda \langle \boldsymbol{\phi}(s, a), \boldsymbol{D}(\boldsymbol{\mu}_h || \boldsymbol{\mu}_h^0) \rangle \big] \Big] \tag{B.11}$$
$$= \max_{a \in \mathcal{A}} \Big[ r_h(s, a) + \inf_{\boldsymbol{\mu}_h \in \Delta(\mathcal{S})^d, P_h = \langle \boldsymbol{\phi}, \boldsymbol{\mu}_h \rangle} \big[ \mathbb{E}_{s' \sim P_h(\cdot|s,a)} \big[ V_{h+1}^{\star, \lambda}(s') \big] + \lambda \langle \boldsymbol{\phi}(s, a), \boldsymbol{D}(\boldsymbol{\mu}_h || \boldsymbol{\mu}_h^0) \rangle \big] \Big]$$
$$= V_h^{\pi^\star, \lambda}(s), \tag{B.12}$$

where (B.11) holds by the fact that $V_{h+1}^{\pi^\star, \lambda}(s) \leq V_{h+1}^{\star, \lambda}(s), \forall s \in \mathcal{S}$, (B.12) holds by (B.10). In turn, we trivially have $V_h^{\star, \lambda}(s) \geq V_h^{\pi^\star, \lambda}(s)$ due to the optimality of the value function. Hence, we obtain $V_h^{\pi^\star, \lambda}(s) = V_h^{\star, \lambda}(s), \forall s \in \mathcal{S}$. Therefore, by the induction argument, we conclude the proof. $\qquad \square$

### B.3. Proof of Proposition 4.1

*Proof.* By Proposition 3.2, we have

$$Q_h^{\pi, \lambda}(s, a) = r_h(s, a) + \inf_{\boldsymbol{\mu}_h \in \Delta(\mathcal{S})^d, P_h = \langle \boldsymbol{\phi}, \boldsymbol{\mu}_h \rangle} \big[ \mathbb{E}_{s' \sim P_h(\cdot|s,a)} \big[ V_{h+1}^{\pi, \lambda}(s') \big] + \lambda \langle \boldsymbol{\phi}(s, a), \boldsymbol{D}(\boldsymbol{\mu}_h || \boldsymbol{\mu}_h^0) \rangle \big]$$

$$= \langle \boldsymbol{\phi}(s, a), \boldsymbol{\theta}_h \rangle + \inf_{\boldsymbol{\mu}_h \in \Delta(\mathcal{S})^d} \Big[ \langle \boldsymbol{\phi}(s, a), \mathbb{E}_{s' \sim \boldsymbol{\mu}_h} [V_{h+1}^{\pi, \lambda}(s')] \rangle + \lambda \sum_{i=1}^d \phi_i(s, a) D(\mu_{h,i} || \mu_{h,i}^0) \Big]$$

$$= \langle \boldsymbol{\phi}(s, a), \boldsymbol{\theta}_h \rangle + \langle \boldsymbol{\phi}(s, a), \boldsymbol{w}_h^{\pi, \lambda} \rangle$$

$$= \langle \boldsymbol{\phi}(s, a), \boldsymbol{\theta}_h + \boldsymbol{w}_h^{\pi, \lambda} \rangle.$$

Hence we conclude the proof. $\qquad \square$

### B.4. Proof of Proposition 4.3

*Proof.* The optimization problem can be formalized as:

$$\inf_{\mu} \mathbb{E}_{s\sim\mu} V(s) + \lambda D_{\text{TV}}(\mu\|\mu^0) \ \text{ subject to } \sum_s \mu(s) = 1, \mu(s) \geq 0.$$

Denote $y(s) = \mu(s) - \mu^0(s)$, the objective function can be rewritten as:

$$
\begin{aligned}
\mathbb{E}_{s\sim\mu} V(s) + \lambda D_{\text{TV}}(\mu\|\mu^0) &= \sum_s \mu(s)V(s) + \lambda/2 \sum_s |\mu(s) - \mu^0(s)| \\
&= \sum_s V(s)(y(s) + \mu^0(s)) + \lambda/2 \sum |y(s)| \\
&= \mathbb{E}_{s\sim\mu^0} V(s) + \sum_s V(s)y(s) + \lambda/2 \sum |y(s)|.
\end{aligned}
$$

Recall the constraint $\sum_s y(s) = 0, y(s) \geq -\mu^0(s)$, by the Lagrange duality, we establish the Lagrangian function:

$$\mathcal{L} = \min_y \max_{\mu\geq 0, r\in R} \Big( \sum_s [y(s)(V(s) - \mu(s) - r)) + \lambda/2|y(s)|] - \sum_s \mu(s)\mu^0(s) \Big).$$

In order to achieve the minimax optimality, for any $s$, term $y(s)(V(s) - \mu(s) - r)) + \lambda/2|y(s)|$ should obtain a bounded lower bound with respect to $y(s)$, which requires that $\mu(s), r$ should satisfy the following conditions:

$$\forall s \in \mathcal{S}, |V(s) - \mu(s) - r| \leq \lambda/2 \Rightarrow \max_{s\in\mathcal{S}}\{V(s) - \mu(s)\} - \min_{s\in\mathcal{S}}\{V(s) - \mu(s)\} \leq \lambda.$$

With the constraint above, we denote $g(s) := V(s) - \mu(s)$, we have

$$
\begin{aligned}
\mathcal{L} &= \max_{\mu\geq 0, r\in R} \min_y \Big\{ \sum_s [y(s)(V(s) - \mu(s) - r)) + \lambda/2|y(s)|] - \sum_s \mu(s)\mu^0(s) \Big\} \\
&= \max_{\max_{s\in\mathcal{S}}(V(s)-\mu(s))-\min_{s\in\mathcal{S}}(V(s)-\mu(s))\leq\lambda} - \sum \mu(s)\mu^0(s) \\
&= \max_{\max_s g(s)-\min_s g(s)\leq\lambda, g(s)\leq V(s)} \Big\{ \sum g(s)\mu^0(s) \Big\} - \mathbb{E}_{s\sim\mu^0} V(s).
\end{aligned}
$$

Thus we have,

$$
\begin{aligned}
&\mathbb{E}_{s\sim\mu} V(s) + \lambda D_{\text{TV}}(\mu\|\mu^0) \\
&= \mathbb{E}_{s\sim\mu^0} V(s) + \max_{\max_s g(s)-\min_s g(s)\leq\lambda, g(s)\leq V(s)} \Big\{ \sum g(s)\mu^0(s) \Big\} - \mathbb{E}_{s\sim\mu^0} V(s) \\
&= \mathbb{E}_{s\sim\mu^0}[V(s)]_{V_{\min}+\lambda},
\end{aligned}
\tag{B.13}
$$

where (B.13) holds by directly solving the max problem. Hence we conclude the proof. $\square$

### B.5. Proof of Proposition 4.8

*Proof.* Similar to the proof of Proposition 4.3, define $y(s) = \mu(s) - \mu^0(s)$, with lagrange duality, we have:

$$
\begin{aligned}
\mathcal{L} &= \sum_s V(s)(y(s) + \mu^0(s)) + \lambda \sum_s \frac{y(s)^2}{\mu^0(s)} - \sum_s (\mu^0(s) + y(s))\mu(s) - r\sum_s y(s) \\
&= \sum_s \Big( \frac{\lambda y(s)^2}{\mu^0(s)} + y(s)(V(s) - \mu(s) - r) + \sum_s V(s)\mu^0(s) - \sum_s \mu^0(s)\mu(s) \Big).
\end{aligned}
$$

Noticing that $\mathcal{L}$ is a quadratic function with respect to $y(s)$, therefore after we fix the term $\mu(s), r$ and compute the min with respect to $y(s)$, we have

$$\mathcal{L} = -\frac{1}{4\lambda} \sum_s \mu^0(s)(V(s) - \mu(s) - r)^2 + \sum_s V(s)\mu^0(s) - \sum_s \mu^0(s)\mu(s)$$

$$= -\frac{1}{4\lambda}\left[\mathbb{E}_{s\sim\mu^0}[V-\mu]^2 - \left(\mathbb{E}_{s\sim\mu^0}[V-\mu]\right)^2\right] + \mathbb{E}_{s\sim\mu^0}[V-\mu] \tag{B.14}$$

$$= \mathbb{E}_{s\sim\mu^0}[V-\mu] - \frac{1}{4\lambda}\mathrm{Var}_{s\sim\mu^0}[V-\mu]$$

$$= \sup_{\alpha\in[V_{\min},V_{\max}]}\mathbb{E}_{s\sim\mu^0}[V(s)]_\alpha - \frac{1}{4\lambda}\mathrm{Var}_{s\sim\mu^0}[V(s)]_\alpha, \tag{B.15}$$

where (B.14) comes from maximizing $r$, and (B.15) comes from maximizing $\mu(s), s \in \mathcal{S}$ and the observation that $\mu(s) = 0$ or $V(s), \forall s \in \mathcal{S}$ when achieving its maximum. Hence, we conclude the proof. $\qquad\square$

## C. Proof of the Upper Bounds of Suboptimality

In this section, we prove Theorem 5.1 and Corollary 5.4. For simplicity, we denote $\phi_h^\tau = \phi(s_h^\tau, a_h^\tau)$. According to the robust regularized Bellman equation in Proposition 3.2, we first define the robust regularized Bellman operator: for any $(h, s, a) \in [H] \times \mathcal{S} \times \mathcal{A}$ and any function $V : \mathcal{S} \times \mathcal{A} \to [0, H]$,

$$\mathcal{T}_h^\lambda V(s,a) := r_h(s,a) + \inf_{\boldsymbol{\mu}_h\in\Delta(\mathcal{S})^d, P_h=\langle\phi,\boldsymbol{\mu}_h\rangle}\left[\mathbb{E}_{s'\sim P_h(\cdot|s,a)}\left[V_{h+1}^{\pi,\lambda}(s')\right] + \lambda\langle\phi(s,a), \boldsymbol{D}(\boldsymbol{\mu}_h\|\boldsymbol{\mu}_h^0)\rangle\right]. \tag{C.1}$$

We have $Q_h^{\pi,\lambda}(s,a) = \mathcal{T}_h^\lambda V_{h+1}^{\pi,\lambda}(s,a)$.

### C.1. Proof of Theorem 5.1

We start from bounding the suboptimality gap by the estimation uncertainty in the following Lemma.

**Lemma C.1.** If the following inequality holds for any $(h, s, a) \in [H] \times \mathcal{S} \times \mathcal{A}$:

$$|\mathcal{T}_h^\lambda \hat{V}_{h+1}^\lambda(s,a) - \langle\phi(s,a), \hat{\boldsymbol{w}}_h^\lambda\rangle| \le \Gamma_h(s,a),$$

then we have

$$\mathrm{SupOpt}(\hat{\pi}, s, \lambda) \le 2 \sup_{P\in\mathcal{U}^\lambda(P^0)} \sum_{h=1}^H \mathbb{E}^{\pi^\star,P}\left[\Gamma_h(s_h,a_h)|s_1=s\right].$$

C.1.1. PROOF OF THEOREM 5.1 - CASE WITH THE TV DIVERGENCE

---

**Algorithm 3** Robust Regularized Pessimistic Value Iteration under TV distance (R2PVI-TV)

---

**Require:** Dataset $\mathcal{D}$, regularizer $\lambda > 0$, $\gamma > 0$ and parameter $\beta$
1: init $\hat{V}_{H+1}^\lambda(\cdot) = 0$
2: **for** episode $h = H, \cdots, 1$ **do**
3: $\quad \boldsymbol{\Lambda}_h \leftarrow \sum_{\tau=1}^K \phi(s_h^\tau, a_h^\tau)(\phi(s_h^\tau, a_h^\tau))^\top + \gamma\mathbf{I}$
4: $\quad \alpha_{h+1} \leftarrow \min_{s\in\mathcal{S}}\{\hat{V}_{h+1}^\lambda(s)\} + \lambda$
5: $\quad \hat{\boldsymbol{w}}_h^\lambda \leftarrow \boldsymbol{\Lambda}_h^{-1}(\sum_{\tau=1}^K \phi(s_h^\tau, a_h^\tau)[\hat{V}_{h+1}^\lambda(s_{h+1}^\tau)]_{\alpha_{h+1}})$ $\qquad\qquad\qquad\qquad$ ◁ Estimated by (4.1)
6: $\quad \Gamma_h(\cdot,\cdot) \leftarrow \beta\sum_{i=1}^d \|\phi_i(\cdot,\cdot)\mathbf{1}_i\|_{\boldsymbol{\Lambda}_h^{-1}}$
7: $\quad \hat{Q}_h^\lambda(\cdot,\cdot) \leftarrow \min\{\phi(\cdot,\cdot)^\top(\boldsymbol{\theta}_h + \hat{\boldsymbol{w}}_h^\lambda) - \Gamma_h(\cdot,\cdot), H - h + 1\}^+$
8: $\quad \hat{\pi}_h(\cdot|\cdot) \leftarrow \arg\max_{\pi_h}\langle\hat{Q}_h^\lambda(\cdot,\cdot), \pi_h(\cdot|\cdot)\rangle_{\mathcal{A}}$ and $\hat{V}_h^\lambda(\cdot) \leftarrow \langle\hat{Q}_h^\lambda(\cdot,\cdot), \hat{\pi}_h(\cdot|\cdot)\rangle_{\mathcal{A}}$
9: **end for**

---

For completeness, we present R2PVI specific to the TV distance in Algorithm 3, which gives a closed form solution of (4.1). Now we present the upper bound of weights as follows.

**Lemma C.2.** (Bound of weights - TV) For any $h \in [H]$, we have

$$\|\boldsymbol{w}_h^\lambda\|_2 \le H\sqrt{d}, \|\hat{\boldsymbol{w}}_h^\lambda\|_2 \le H\sqrt{\frac{Kd}{\gamma}}.$$

*Proof of Theorem 5.1 - TV.* The R2PVI with TV-divergence is presented in Algorithm 3. We derive the upper bound on the estimation uncertainty $\Gamma_h(s,a)$ to prove the theorem. We first decompose the difference between the regularized robust bellman operator $\mathcal{T}_h^\lambda$ and the empirical regularized robust bellman operator $\hat{\mathcal{T}}_h^\lambda$ as

$$\left|\mathcal{T}_h^\lambda \hat{V}_{h+1}^\lambda(s,a) - \langle \boldsymbol{\phi}(s,a), \hat{\boldsymbol{w}}_h^\lambda \rangle\right| \tag{C.2}$$

$$= \left|\sum_{i=1}^d \phi_i(s,a)(w_{h,i}^\lambda - \hat{w}_{h,i}^\lambda)\right|$$

$$= \left|\sum_{i=1}^d \phi_i(s,a)\mathbf{1}_i(\boldsymbol{w}_h^\lambda - \hat{\boldsymbol{w}}_h^\lambda)\right|$$

$$= \left|\gamma \sum_{i=1}^d \phi_i(s,a)\mathbf{1}_i \boldsymbol{\Lambda}_h^{-1}\boldsymbol{w}_h^\lambda + \sum_{i=1}^d \phi_i(s,a)\mathbf{1}_i \boldsymbol{\Lambda}_h^{-1}\sum_{\tau=1}^K \boldsymbol{\phi}(s_h^\tau, a_h^\tau)\eta_h^\tau([\hat{V}_{h+1}^\lambda(s)]_{\alpha_{h+1}})\right| \tag{C.3}$$

$$\leq \gamma \sum_{i=1}^d \|\phi_i(s,a)\mathbf{1}_i\|_{\boldsymbol{\Lambda}_h^{-1}}\underbrace{\|\boldsymbol{w}_h^\lambda\|_{\boldsymbol{\Lambda}_h^{-1}}}_{\text{(i)}} + \sum_{i=1}^d \|\phi_i(s,a)\mathbf{1}_i\|_{\boldsymbol{\Lambda}_h^{-1}}\underbrace{\|\sum_{\tau=1}^K \boldsymbol{\phi}(s_h^\tau, a_h^\tau)\eta_h^\tau([\hat{V}_{h+1}^\lambda(s)]_{\alpha_{h+1}})\|_{\boldsymbol{\Lambda}_h^{-1}}}_{\text{(ii)}}, \tag{C.4}$$

where (C.3) comes from the definition of $\hat{\boldsymbol{w}}_h^\lambda$, while (C.4) follows by the Cauchy-Schwartz inequality. By Lemma C.2 and the fact that $\hat{V}_{h+1}^\lambda(s) \leq H$ and $\gamma = 1$, we have

$$\text{(i)} = \|\boldsymbol{w}_h^\lambda\|_{\boldsymbol{\Lambda}_h^{-1}} \leq \|\boldsymbol{\Lambda}_h^{-1}\|^{1/2}\|\boldsymbol{w}_h^\lambda\|_2 \leq H\sqrt{d},$$

where the last inequality comes from the fact that $\|\boldsymbol{\Lambda}_h^{-1}\| \leq \gamma^{-1}$. Now it remains to bound term (ii), as $\hat{V}_{h+1}^\lambda$ depends on data, which makes it difficult to bound it directly by concentration equality. Instead, we consider focus on the function class $\mathcal{V}_h(R_0, B_0, \gamma)$:

$$\mathcal{V}_h(R_0, B_0, \gamma) = \{V_h(x; \boldsymbol{\theta}, \beta, \boldsymbol{\Lambda}): \mathcal{S} \to [0,H], \|\boldsymbol{\theta}\|_2 \leq R_0, \beta \in [0, B_0], \gamma_{\min}(\boldsymbol{\Lambda}_h) \geq \gamma\},$$

where $V_h(x; \boldsymbol{\theta}, \beta, \boldsymbol{\Lambda}) = \max_{a \in \mathcal{A}}[\boldsymbol{\phi}(s,a)^\top \boldsymbol{\theta} - \beta \sum_{i=1}^d \|\phi_i(s,a)\|_{\boldsymbol{\Lambda}_h^{-1}}]_{[0,H-h+1]}$. By Lemma C.2 and the definition of $\hat{V}_{h+1}^\lambda$, when we set $R_0 = H\sqrt{Kd/\gamma}, B_0 = \beta = 16Hd\sqrt{\xi_{\text{TV}}}$, it suffices to show that $\hat{V}_{h+1}^\lambda \in \mathcal{V}_{h+1}(R_0, B_0, \gamma)$. Next we aim to find a union cover of the $\mathcal{V}_{h+1}(R_0, B_0, \gamma)$, hence the term (ii) can be upper bounded. Let $\mathcal{N}_h(\epsilon; R_0, B_0, \gamma)$ be the minimum $\epsilon$-cover of $\mathcal{V}_h(R_0, B_0, \lambda)$ with respect to the supreme norm, $\mathcal{N}_h([0,H])$ be the minimum $\epsilon$-cover of $[0,H]$ respectively. In other words, for any function $V \in \mathcal{V}_h(R_0, B_0, \gamma), \alpha_{h+1} \in [0,H]$, there exists a function $V' \in \mathcal{V}_h(R_0, B_0, \gamma)$ and a real number $\alpha_\epsilon \in [0,H]$ such that:

$$\sup_{s \in \mathcal{S}} |V(s) - V'(s)| \leq \epsilon, |\alpha_\epsilon - \alpha_{h+1}| \leq \epsilon.$$

By Cauchy-Schwartz inequality and the fact that $\|a+b\|_{\boldsymbol{\Lambda}_h^{-1}}^2 \leq 2\|a\|_{\boldsymbol{\Lambda}_h^{-1}}^2 + 2\|b\|_{\boldsymbol{\Lambda}_h^{-1}}^2$ and the definition of the term (ii), we have

$$\text{(ii)}^2 \leq 2\left\|\sum_{\tau=1}^K \boldsymbol{\phi}(s_h^\tau, a_h^\tau)\eta_h^\tau([\hat{V}_{h+1}^\lambda]_{\alpha_\epsilon})\right\|_{\boldsymbol{\Lambda}_h^{-1}}^2 + 2\left\|\sum_{\tau=1}^K \boldsymbol{\phi}(s_h^\tau, a_h^\tau)\eta_h^\tau([\hat{V}_{h+1}^\lambda]_{\alpha_{h+1}} - [\hat{V}_{h+1}^\lambda]_{\alpha_\epsilon})\right\|_{\boldsymbol{\Lambda}_h^{-1}}^2$$

$$\leq 4\left\|\sum_{\tau=1}^K \boldsymbol{\phi}(s_h^\tau, a_h^\tau)\eta_h^\tau([V'_{h+1}]_{\alpha_\epsilon})\right\|_{\boldsymbol{\Lambda}_h^{-1}}^2 + 4\left\|\sum_{\tau=1}^K \boldsymbol{\phi}(s_h^\tau, a_h^\tau)\eta_h^\tau([\hat{V}_{h+1}^\lambda]_{\alpha_\epsilon} - [V'_{h+1}]_{\alpha_\epsilon})\right\|_{\boldsymbol{\Lambda}_h^{-1}}^2 + \frac{2\epsilon^2 K^2}{\gamma}, \tag{C.5}$$

where (C.5) follows by the fact that

$$2\left\|\sum_{\tau=1}^K \boldsymbol{\phi}(s_h^\tau, a_h^\tau)\eta_h^\tau([\hat{V}_{h+1}^\lambda]_{\alpha_{h+1}} - [\hat{V}_{h+1}^\lambda]_{\alpha_\epsilon})\right\|_{\boldsymbol{\Lambda}_h^{-1}}^2 \leq 2\epsilon^2 \sum_{\tau=1,\tau'=1}^K |\boldsymbol{\phi}_h^\tau \boldsymbol{\Lambda}_h^{-1}\boldsymbol{\phi}_h^{\tau\top}| \leq \frac{2\epsilon^2 K^2}{\gamma}.$$

Meanwhile, by the fact that $|[\hat{V}_{h+1}^\lambda]_{\alpha_\epsilon} - [V_{h+1}']_{\alpha_\epsilon}| \le |\hat{V}_{h+1}^\lambda - V_{h+1}'|$, we have

$$4\Big\| \sum_{\tau=1}^K \phi(s_h^\tau, a_h^\tau)\eta_h^\tau([\hat{V}_{h+1}^\lambda]_{\alpha_\epsilon} - [V_{h+1}']_{\alpha_\epsilon}) \Big\|_{\Lambda_h^{-1}}^2 \tag{C.6}$$

$$\le 4 \sum_{\tau=1,\tau'=1}^K |\phi_h^\tau \Lambda_h^{-1} \phi_h^{\tau'}| \max |\eta_h^\tau([\hat{V}_{h+1}^\lambda]_{\alpha_\epsilon} - [V_{h+1}']_{\alpha_\epsilon})|^2$$

$$\le 4\epsilon^2 \sum_{\tau=1,\tau'=1}^K |\phi_h^\tau \Lambda_h^{-1} \phi_h^{\tau'}|$$

$$\le \frac{4\epsilon^2 K^2}{\gamma}. \tag{C.7}$$

By applying the (C.7) into (C.5), we have

$$(\text{ii})^2 \le 4 \sup_{V' \in \mathcal{N}_h(\epsilon; R_0, B_0, \gamma), \alpha_\epsilon \in \mathcal{N}_h([0,H])} \Big\| \sum_{\tau=1}^K \phi(s_h^\tau, a_h^\tau)\eta_h^\tau([V_{h+1}']_{\alpha_\epsilon}) \Big\|_{\Lambda_h^{-1}}^2 + \frac{6\epsilon^2 K^2}{\gamma}. \tag{C.8}$$

By Lemma F.3, applying a union bound over $\mathcal{N}_h(\epsilon; R_0, B_0, \gamma)$ and $\mathcal{N}_h([0,H])$, with probability at least $1 - \delta/2H$, we have

$$4 \sup_{V' \in \mathcal{N}_h(\epsilon; R_0, B_0, \gamma), \alpha_\epsilon \in \mathcal{N}_h([0,H])} \Big\| \sum_{\tau=1}^K \phi(s_h^\tau, a_h^\tau)\eta_h^\tau([V_{h+1}']_{\alpha_\epsilon}) \Big\|_{\Lambda_h^{-1}}^2 + \frac{6\epsilon^2 K^2}{\gamma}$$

$$\le 4H^2 \Big( 2\log \frac{2H|\mathcal{N}_h(\epsilon; R_0, B_0, \gamma)| |\mathcal{N}_h([0,H])|}{\delta} + d\log(1 + K/\gamma) \Big) + \frac{6\epsilon^2 K^2}{\gamma}. \tag{C.9}$$

Applying Lemma F.1, we have

$$\log|\mathcal{N}_h(\epsilon; R_0, B_0, \lambda)| \le d\log(1 + 4R_0/\epsilon) + d^2\log(1 + 8d^{1/2}B_0^2/\gamma\epsilon^2)$$

$$= d\log(1 + 4K^{3/2}d^{-1/2}) + d^2\log(1 + 8d^{-3/2}B_0^2 K^2 H^{-2})$$

$$\le 2d^2\log(1 + 8d^{-3/2}B_0^2 K^2 H^{-2}). \tag{C.10}$$

Similarly, by Lemma F.2, we have

$$|\mathcal{N}_h([0,H])| \le 3H/\epsilon.$$

Combining (C.10) with (C.8) and (C.9), by setting $\epsilon = dH/K$, we have

$$(\text{ii})^2 \le 4H^2 \Big( 2\log \frac{2H|\mathcal{N}_h(\epsilon; R_0, B_0, \gamma)| |\mathcal{N}_h([0,H])|}{\delta} + d\log(1 + K/\gamma) \Big) + \frac{6\epsilon^2 K^2}{\gamma}$$

$$\le 4H^2(4d^2\log(1 + 8d^{-3/2}B_0^2 K^2 H^{-2}) + \log(3K/d) + d\log(1+K) + 2\log 2H/\delta) + 6d^2 H^2$$

$$\le 16H^2 d^2(\log(1 + 8d^{-3/2}B_0^2 K^2 H^{-2}) + \log(1+K)/d + 3/8 + \log H/\delta)$$

$$\le 32H^2 d^2 \log 8d^{-3/2}B_0^2 K^2 H^{-1}/\delta$$

$$= 32H^2 d^2 \log 1024 H d^{1/2} K^2 \xi_{\text{TV}}/\delta$$

$$= 32H^2 d^2(\log 1024 H d^{1/2} K^2/\delta + \log \xi_{\text{TV}})$$

$$\le 64H^2 d^2 \xi_{\text{TV}} := \frac{\beta^2}{4}.$$

Recall the upper bound in (C.4), we have with probability at least $1 - \delta$,

$$|\mathcal{T}_h^\lambda \hat{V}_{h+1}^\lambda(s,a) - \langle \phi(s,a), \hat{w}_h^\lambda \rangle|$$

$$\le \gamma \sum_{i=1}^d \|\phi_i(s,a)\mathbf{1}_i\|_{\Lambda_h^{-1}} \|w_h^\lambda\|_{\Lambda_h^{-1}} + \sum_{i=1}^d \|\phi_i(s,a)\mathbf{1}_i\|_{\Lambda_h^{-1}} \Big\| \sum_{\tau=1}^K \phi(s_h^\tau, a_h^\tau)\eta_h^\tau([\hat{V}_{h+1}^\lambda(s)]_{\alpha_\epsilon}) \Big\|_{\Lambda_h^{-1}}$$

$$\leq \sum_{i=1}^{d} \|\phi_i(s,a)\mathbf{1}_i\|_{\mathbf{\Lambda}_h^{-1}} (H\sqrt{d} + \beta/2)$$

$$\leq \beta \sum_{i=1}^{d} \|\phi_i(s,a)\mathbf{1}_i\|_{\mathbf{\Lambda}_h^{-1}}, \tag{C.11}$$

where (C.11) follows by the fact that $2H\sqrt{d} \leq \beta$. Hence, the prerequisite is satisfied in Lemma C.1, we can upper bound the suboptimality gap as:

$$\text{SubOpt}(\hat{\pi}, s, \lambda) \leq 2 \sup_{P \in \mathcal{U}^\lambda(P^0)} \sum_{h=1}^{H} \mathbb{E}^{\pi^\star, P}\big[\Gamma_h(s_h, a_h)|s_1 = s\big]$$

$$= 2\beta \cdot \sup_{P \in \mathcal{U}^\lambda(P^0)} \sum_{h=1}^{H} \mathbb{E}^{\pi^\star, P}\Big[\sum_{i=1}^{d} \|\phi_i(s,a)\mathbf{1}_i\|_{\mathbf{\Lambda}_h^{-1}}|s_1 = s\Big].$$

This concludes the proof. $\qquad\square$

### C.1.2. PROOF OF THEOREM 5.1 - CASE WITH KL DIVERGENCE

---

**Algorithm 4** Robust Regularized Pessimistic Value Iteration under KL distance (R2PVI-KL)

---
**Require:** Dataset $\mathcal{D}$, regularizer $\lambda > 0, \gamma > 0$ and parameter $\beta$
1: init $\hat{V}_{H+1}^\lambda(\cdot) = 0$
2: **for** episode $h = H, \cdots, 1$ **do**
3: $\quad \mathbf{\Lambda}_h \leftarrow \sum_{\tau=1}^{K} \phi(s_h^\tau, a_h^\tau)(\phi(s_h^\tau, a_h^\tau))^\top + \gamma\mathbf{I}$
4: $\quad \hat{\boldsymbol{w}}_h' \leftarrow \mathbf{\Lambda}_h^{-1}\big(\sum_{\tau=1}^{K} \phi(s_h^\tau, a_h^\tau)e^{-\frac{\hat{V}_{h+1}^\lambda(s_{h+1}^\tau)}{\lambda}}\big)$
5: $\quad \hat{\boldsymbol{w}}_h^\lambda \leftarrow -\lambda \log \max\{\hat{\boldsymbol{w}}_h', e^{-H/\lambda}\}$
6: $\quad \Gamma_h(\cdot, \cdot) \leftarrow \beta \sum_{i=1}^{d} \|\phi_i(\cdot, \cdot)\mathbf{1}_i\|_{\mathbf{\Lambda}_h^{-1}}$
7: $\quad \hat{Q}_h^\lambda(\cdot, \cdot) \leftarrow \min\{\phi(\cdot, \cdot)^\top(\boldsymbol{\theta}_h + \hat{\boldsymbol{w}}_h^\lambda) - \Gamma_h(\cdot, \cdot), H - h + 1\}^+$
8: $\quad \hat{\pi}_h(\cdot|\cdot) \leftarrow \text{argmax}_{\pi_h}\langle\hat{Q}_h^\lambda(\cdot, \cdot), \pi_h(\cdot|\cdot)\rangle_{\mathcal{A}}$ and $\hat{V}_h^\lambda(\cdot) \leftarrow \langle\hat{Q}_h^\lambda(\cdot, \cdot), \hat{\pi}_h(\cdot|\cdot)\rangle_{\mathcal{A}}$
9: **end for**

---

For completeness, we present the R2PVI algorithm specific to the KL distance in Algorithm 4. Note that we have used the following closed form solution

$$\hat{\boldsymbol{w}}_h' = \underset{\boldsymbol{w} \in \mathbb{R}^d}{\text{argmin}} \sum_{\tau=1}^{K} \Big(e^{-\frac{\hat{V}_{h+1}^\lambda(s_{h+1}^\tau)}{\lambda}} - \phi(s_h^\tau, a_h^\tau)^\top \boldsymbol{w}\Big)^2 + \gamma\|\boldsymbol{w}\|_2^2. \tag{C.12}$$

$$= \mathbf{\Lambda}_h^{-1}\Big(\sum_{\tau=1}^{K} \phi(s_h^\tau, a_h^\tau)e^{-\frac{\hat{V}_{h+1}^\lambda(s_{h+1}^\tau)}{\lambda}}\Big). \tag{C.13}$$

Our proof relies on the following lemmas on bounding the regression parameter and $\epsilon$-covering number of the robust value function class.

**Lemma C.3** (Bound of weights - KL). *For any $h \in [H]$,*

$$\|\boldsymbol{w}_h^\lambda\|_2 \leq \sqrt{d}, \|\hat{\boldsymbol{w}}_h'\|_2 \leq \sqrt{\frac{Kd}{\gamma}}.$$

**Lemma C.4** (Bound of covering number - KL). *For any $h \in [H]$, let $\mathcal{V}_h$ denote a class of functions mapping from $\mathcal{S}$ to $\mathbb{R}$ with the following form:*

$$V_h(x; \boldsymbol{\theta}, \beta, \mathbf{\Lambda}_h) = \max_{a \in \mathcal{A}}\Big\{\phi(s,a)^\top\boldsymbol{\theta} - \lambda \log\Big(1 + \beta \sum_{i=1}^{d} \|\phi_i(\cdot, \cdot)\mathbf{1}_i\|_{\mathbf{\Lambda}_h^{-1}}\Big)\Big\}_{[0, H-h+1]},$$

the parameters $(\boldsymbol{\theta}, \beta, \boldsymbol{\Lambda}_h)$ satisfy $\|\boldsymbol{\theta}\|_2 \leq L, \beta \in [0, B], \gamma_{\min}(\boldsymbol{\Lambda}_h) \geq \gamma$. Let $\mathcal{N}_h(\epsilon)$ be the $\epsilon$-covering number of $\mathcal{V}$ with respect to the distance $\text{dist}(V_1, V_2) = \sup_x |V_1(x) - V_2(x)|$. Then

$$\log |\mathcal{N}_h(\epsilon)| \leq d \log(1 + 4L/\epsilon) + d^2 \log(1 + 8\lambda^2 d^{1/2} B^2/\gamma\epsilon^2).$$

*Proof.* The R2PVI with KL-divergence is presented in Algorithm 4. Similar to the proof of TV divergence, we decompose the estimation uncertainty between $\mathcal{T}_h^\lambda$ and $\hat{\mathcal{T}}_h^\lambda$ as:

$$
\begin{aligned}
|\mathcal{T}_h^\lambda \hat{V}_{h+1}^\lambda(s, a) - \langle \boldsymbol{\phi}(s, a), \hat{\boldsymbol{w}}_h^\lambda \rangle| &= \left| \boldsymbol{\phi}(s, a)^\top \left( \boldsymbol{\theta}_h - \lambda \log \boldsymbol{w}_h^\lambda - \boldsymbol{\theta}_h + \lambda \log \max\{\hat{\boldsymbol{w}}_h', e^{-H/\lambda}\} \right) \right| \\
&= \left| \boldsymbol{\phi}(s, a)^\top \left( \lambda \log \max\{\hat{\boldsymbol{w}}_h', e^{-H/\lambda}\} - \lambda \log \boldsymbol{w}_h^\lambda \right) \right| \\
&= \lambda \left| \sum_{i=1}^d \phi_i(s, a) \log \frac{\max\{\hat{w}_{h,i}', e^{-H/\lambda}\}}{w_{h,i}^\lambda} \right| \\
&\leq \lambda \sum_{i=1}^d \phi_i(s, a) \left| \log \frac{\max\{\hat{w}_{h,i}', e^{-H/\lambda}\}}{w_{h,i}^\lambda} \right| \\
&\leq \lambda \sum_{i=1}^d \phi_i(s, a) \left| \log \left(1 + e^{H/\lambda} |\max\{\hat{w}_{h,i}', e^{-H/\lambda}\} - w_{h,i}^\lambda| \right) \right| && \text{(C.14)} \\
&\leq \lambda \sum_{i=1}^d \phi_i(s, a) \log \left(1 + e^{H/\lambda} |\hat{w}_{h,i}' - w_{h,i}^\lambda| \right) && \text{(C.15)} \\
&\leq \lambda \log \left( \sum_{i=1}^d \phi_i(s, a) + e^{H/\lambda} \sum_{i=1}^d \phi_i(s, a) |\hat{w}_{h,i}' - w_{h,i}^\lambda| \right) && \text{(C.16)} \\
&= \lambda \log \left(1 + e^{H/\lambda} \sum_{i=1}^d \phi_i(s, a) \mathbf{1}_i^\top |\hat{\boldsymbol{w}}_h' - \boldsymbol{w}_h^\lambda| \right),
\end{aligned}
$$

where (C.14) and (C.15) comes from the fact that:

$$w_{h,i}^\lambda = \mathbb{E}_{s' \sim \mu_{h,i}^0} \left[ e^{-\frac{\hat{V}_h^\lambda(s')}{\lambda}} \right] \geq \mathbb{E}_{s' \sim \mu_{h,i}^0}[e^{-H/\lambda}] = e^{-H/\lambda}, |\log A - \log B| = \log \left(1 + \frac{|A - B|}{\min\{A, B\}}\right),$$

and (C.16) comes from the Jensen's inequality applying to function $\log(x)$. Therefore, our next goal is to bound the term $\sum_{i=1}^d \phi_i(s, a) \mathbf{1}_i^\top |\hat{\boldsymbol{w}}_h' - \boldsymbol{w}_h^\lambda|$. Specifically, we have

$$
\begin{aligned}
&\sum_{i=1}^d \phi_i(s, a) \mathbf{1}_i^\top |\hat{\boldsymbol{w}}_h' - \boldsymbol{w}_h^\lambda| \\
&= \sum_{i=1}^d \phi_i(s, a) \mathbf{1}_i^\top \left| \boldsymbol{w}_h^\lambda - \boldsymbol{\Lambda}_h^{-1} \sum_{\tau=1}^K (\boldsymbol{\phi}_h^\tau) e^{-\frac{\hat{V}_{h+1}^\lambda(s_{h+1})}{\lambda}} \right| \\
&= \sum_{i=1}^d \phi_i(s, a) \mathbf{1}_i^\top \left| \boldsymbol{w}_h^\lambda - \boldsymbol{\Lambda}_h^{-1} \sum_{\tau=1}^K \boldsymbol{\phi}_h^\tau (\boldsymbol{\phi}_h^\tau)^\top \boldsymbol{w}_h^\lambda + \boldsymbol{\Lambda}_h^{-1} \sum_{\tau=1}^K \boldsymbol{\phi}_h^\tau (\boldsymbol{\phi}_h^\tau)^\top \boldsymbol{w}_h^\lambda - \boldsymbol{\Lambda}_h^{-1} \sum_{\tau=1}^K (\boldsymbol{\phi}_h^\tau) e^{-\frac{\hat{V}_{h+1}^\lambda(s_{h+1})}{\lambda}} \right| \\
&= \sum_{i=1}^d \phi_i(s, a) \mathbf{1}_i^\top \left( \underbrace{\left| \boldsymbol{w}_h^\lambda - \boldsymbol{\Lambda}_h^{-1} \sum_{\tau=1}^K \boldsymbol{\phi}_h^\tau (\boldsymbol{\phi}_h^\tau)^\top \boldsymbol{w}_h^\lambda \right|}_{\text{(i)}} + \underbrace{\left| \boldsymbol{\Lambda}_h^{-1} \sum_{\tau=1}^K \boldsymbol{\phi}_h^\tau (\boldsymbol{\phi}_h^\tau)^\top \boldsymbol{w}_h^\lambda - \boldsymbol{\Lambda}_h^{-1} \sum_{\tau=1}^K (\boldsymbol{\phi}_h^\tau) e^{-\frac{\hat{V}_{h+1}^\lambda(s_{h+1})}{\lambda}} \right|}_{\text{(ii)}} \right).
\end{aligned}
$$

Next, we upper bound term (i) and (ii), respectively. For the first term, we have:

$$\sum_{i=1}^d \phi_i(s, a) \mathbf{1}_i^\top \cdot \text{(i)} = \sum_{i=1}^d \phi_i(s, a) \mathbf{1}_i^\top \left( |\boldsymbol{w}_h^\lambda - \boldsymbol{\Lambda}_h^{-1} (\boldsymbol{\Lambda}_h - \gamma \mathbf{I}) \boldsymbol{w}_h^\lambda| \right)$$

$$= \gamma \sum_{i=1}^{d} \phi_i(s,a) \mathbf{1}_i^{\top} \mathbf{\Lambda}_h^{-1} |\boldsymbol{w}_h^{\lambda}|$$

$$\leq \gamma \sum_{i=1}^{d} \|\phi_i(s,a)\mathbf{1}_i\|_{\mathbf{\Lambda}_h^{-1}} \|\boldsymbol{w}_h^{\lambda}\|_{\mathbf{\Lambda}_h^{-1}} \tag{C.17}$$

$$\leq \gamma \sqrt{d} \sum_{i=1}^{d} \|\phi_i(s,a)\mathbf{1}_i\|_{\mathbf{\Lambda}_h^{-1}}, \tag{C.18}$$

where (C.17) follows from the Cauchy-Schwartz inequality, (C.18) follows from the fact that:

$$\|\boldsymbol{w}_h^{\lambda}\|_{\mathbf{\Lambda}_h^{-1}} \leq \|\mathbf{\Lambda}_h^{-1}\|^{1/2} \|\boldsymbol{w}_h^{\lambda}\|_2 \leq \sqrt{d/\gamma},$$

where the last inequality follows from Lemma C.3 and the fact that $\|\mathbf{\Lambda}_h^{-1}\| \leq \gamma^{-1}$. Now it remains to bound the term (ii), by the definition of $\eta_h^{\tau}(f) = \mathbb{E}_{s' \sim P_h^0(\cdot|s_h^{\tau}, a_h^{\tau})}[f(s')] - f(s_{h+1}^{\tau})$, the term (ii) can be rewritten as:

$$(\text{ii}) = \sum_{i=1}^{d} \phi_i(s,a)\mathbf{1}_i^{\top} \left| \mathbf{\Lambda}_h^{-1} \sum_{\tau=1}^{K} \phi_h^{\tau} \left[ (\phi_h^{\tau})^{\top} \boldsymbol{w}_h^{\lambda} - e^{-\frac{\hat{V}_{h+1}^{\lambda}(s_{h+1})}{\lambda}} \right] \right|$$

$$= \sum_{i=1}^{d} \phi_i(s,a)\mathbf{1}_i^{\top} \left| \mathbf{\Lambda}_h^{-1} \sum_{\tau=1}^{K} \phi_h^{\tau} \eta_h^{\tau} \left( e^{-\frac{\hat{V}_{h+1}^{\lambda}(s)}{\lambda}} \right) \right|$$

$$\leq \sum_{i=1}^{d} \|\phi_i(s,a)\mathbf{1}_i\|_{\mathbf{\Lambda}_h^{-1}} \underbrace{\left\| \sum_{\tau=1}^{K} \phi_h^{\tau} \eta_h^{\tau} \left( e^{-\frac{\hat{V}_{h+1}^{\lambda}(s)}{\lambda}} \right) \right\|_{\mathbf{\Lambda}_h^{-1}}}_{(\text{iii})}.$$

For the rest of the proof, it's left to bound the term (iii). As the $\hat{V}_{h+1}^{\lambda}$ depends on the offline dataset, which makes it difficult to upper bound directly from concentration equality due to the dependence issue, we seek for providing a uniform concentration bound applied to the term (iii), i.e. we aim to upper bound the following term:

$$\sup_{V \in \mathcal{V}_{h+1}(R,B,\gamma)} \left\| \sum_{\tau=1}^{K} \phi_h^{\tau} \eta_h^{\tau} (e^{-\frac{V}{\lambda}}) \right\|_{\mathbf{\Lambda}_h^{-1}}.$$

Here for all $h \in [H]$, the function class is defined as:

$$\mathcal{V}_h(R,B,\gamma) = \{V_h(x; \boldsymbol{\theta}, \beta, \mathbf{\Lambda}_h) : \|\boldsymbol{\theta}\|_2 \leq R, \beta \in [0,B], \gamma_{\min}(\mathbf{\Lambda}_h) \geq \gamma\},$$

where $V_h(x; \boldsymbol{\theta}, \beta, \mathbf{\Lambda}_h) = \max_{a \in \mathcal{A}} \{\phi(s,a)^{\top}\boldsymbol{\theta} - \lambda \log(1 + \beta \sum_{i=1}^{d} \|\phi_i(\cdot,\cdot)\mathbf{1}_i\|_{\mathbf{\Lambda}_h^{-1}})\}_{[0,H-h+1]}$. In order to ensure $\hat{V}_{h+1}^{\lambda} \in \mathcal{V}_{h+1}(R_0, B_0, \lambda)$, we need to bound $\hat{\boldsymbol{\theta}}_h = \boldsymbol{\theta}_h - \lambda \log \max\{\hat{\boldsymbol{w}}_h', e^{-H/\lambda}\}$. Following the fact that:

$$\|\hat{\boldsymbol{\theta}}_h\|_2 \leq \|\boldsymbol{\theta}_h\|_2 + \lambda \|\log \max\{\hat{\boldsymbol{w}}_h', e^{-H/\lambda}\}\|_2.$$

By Lemma C.3, $e^{-H/\lambda} \leq \max\{\hat{w}_{h,i}', e^{-H/\lambda}\} \leq \max\{\|\hat{\boldsymbol{w}}_h'\|, e^{-H/\lambda}\} \leq \max\{\sqrt{Kd/\lambda}, e^{-H/\lambda}\}$, therefore the term can be bounded as:

$$\|\hat{\boldsymbol{\theta}}_h\|_2 \leq \sqrt{d} + \lambda\sqrt{d} \max\left( \log\sqrt{\frac{Kd}{\lambda}}, H/\lambda \right)$$

$$\leq H\sqrt{d} + d\sqrt{K\lambda}$$

$$\leq 2Hd\sqrt{K\lambda}. \tag{C.19}$$

Hence, we can choose $R_0 = 2Hd\sqrt{K\lambda}$ and $B_0 = \beta = 16d\lambda e^{H/\lambda}\sqrt{(H/\lambda + \xi_{\text{KL}})}$, then we have for all $h \in [H], \hat{V}_{h+1}^{\lambda} \in \mathcal{V}_{h+1}(R_0, B_0, \lambda)$. Next we aim to find a union cover of the $\mathcal{V}_{h+1}(R_0, B_0, \gamma)$, hence the term (iii) can be upper bounded.

For all $\epsilon \in (0, \lambda), h \in [H]$, let $\mathcal{N}_h(\epsilon; R, B, \lambda) := \mathcal{N}_h(\epsilon)$ denote the minimal $\epsilon$-cover of $\mathcal{V}_h(R, B, \lambda)$ with respect to the supreme norm. In other words, for any function $\hat{V}^\lambda \in \mathcal{V}_h(R, B, \lambda)$, there exists a function $V' \in \mathcal{N}_{h+1}(\epsilon)$ such that

$$\sup_{x \in \mathcal{S}} |\hat{V}_{h+1}^\lambda(x) - V_{h+1}'(x)| \le \epsilon.$$

Hence, given $\hat{V}_{h+1}^\lambda, V_{h+1}'$ satisfying the inequality above, recall the definition of $\eta_h^\tau = \eta_h^\tau(f) = \mathbb{E}_{s' \sim P_h^0(\cdot|s_h^\tau, a_h^\tau)}[f(s')] - f(s_{h+1}^\tau)$, we have:

$$\begin{aligned}
&\left| \eta_h^\tau\left( e^{-\frac{\hat{V}_{h+1}^\lambda(s)}{\lambda}} \right) - \eta_h^\tau\left( e^{-\frac{V_{h+1}'(s)}{\lambda}} \right) \right| \\
&\le \left| \mathbb{E}_{s \sim P_h^0(\cdot|s_h^\tau, a_h^\tau)}\left[ e^{-\frac{\hat{V}_{h+1}^\lambda(s)}{\lambda}} - e^{-\frac{V_{h+1}'(s)}{\lambda}} \right] - e^{-\frac{\hat{V}_{h+1}^\lambda(s_{h+1})}{\lambda}} + e^{-\frac{V_{h+1}'(s_{h+1})}{\lambda}} \right| \\
&\le \left| \mathbb{E}_{s \sim P_h^0(\cdot|s_h^\tau, a_h^\tau)}\left[ e^{-\frac{\hat{V}_{h+1}^\lambda(s)}{\lambda}} - e^{-\frac{V_{h+1}'(s)}{\lambda}} \right] \right| + \left| e^{-\frac{\hat{V}_{h+1}^\lambda(s_{h+1})}{\lambda}} - e^{-\frac{V_{h+1}'(s_{h+1})}{\lambda}} \right| \\
&\le 2\epsilon/\lambda + 2\epsilon/\lambda = 4\epsilon/\lambda, \quad\quad\quad\quad\quad\quad\quad\quad\quad\quad\quad\quad\quad\quad\quad\quad\quad (\text{C.20})
\end{aligned}$$

where (C.20) follows from the fact that for any $s \in \mathcal{S}$,

$$\left| e^{-\frac{\hat{V}_{h+1}^\lambda(s)}{\lambda}} - e^{-\frac{V_{h+1}'(s)}{\lambda}} \right| \le e^{\frac{|\hat{V}_{h+1}^\lambda(s) - V_{h+1}'(s)|}{\lambda}} - 1 \le e^{\frac{\epsilon}{\lambda}} - 1 \le 2\epsilon/\lambda,$$

where the last inequality is held by the fact that $\epsilon \in (0, \lambda)$. By the Cauchy-Schwartz inequality, for any two vectors $a, b \in \mathbb{R}^d$ and positive definite matrix $\Lambda \in \mathbb{R}^{d \times d}$, it holds that $\|a + b\|_\Lambda^2 \le 2\|a\|_\Lambda^2 + 2\|b\|_\Lambda^2$, hence for all $h \in [H]$, we have:

$$\begin{aligned}
|(\text{iii})|^2 &\le 2 \left\| \sum_{\tau=1}^K \phi_h^\tau \eta_h^\tau\left( e^{-\frac{V_{h+1}'(s)}{\lambda}} \right) \right\|_{\Lambda_h^{-1}}^2 + 2 \left\| \sum_{\tau=1}^K \phi_h^\tau \left[ \eta_h^\tau\left( e^{-\frac{V_{h+1}'(s)}{\lambda}} \right) - \eta_h^\tau\left( e^{-\frac{\hat{V}_{h+1}^\lambda(s)}{\lambda}} \right) \right] \right\|_{\Lambda_h^{-1}}^2 \\
&\le 2 \left\| \sum_{\tau=1}^K \phi_h^\tau \eta_h^\tau\left( e^{-\frac{V_{h+1}'(s)}{\lambda}} \right) \right\|_{\Lambda_h^{-1}}^2 + 32\epsilon^2/\lambda^2 \sum_{\tau, \tau'=1}^K |\phi_h^\tau \Lambda_h^{-1} \phi_h^{\tau'}| \\
&\le 2 \sup_{V \in \mathcal{N}_{h+1}(\epsilon)} \left\| \sum_{\tau=1}^K \phi_h^\tau \eta_h^\tau\left( e^{-\frac{V(s)}{\lambda}} \right) \right\|_{\Lambda_h^{-1}}^2 + \frac{32\epsilon^2 K^2}{\lambda^2 \gamma}. \quad\quad\quad\quad\quad (\text{C.21})
\end{aligned}$$

We set $f(s) = e^{-\frac{V(s)}{\lambda}}$, by applying Lemma F.3, for any fixed $h \in [H], \delta \in (0, 1)$, we have:

$$P\left( \sup_{V \in \mathcal{N}_{h+1}(\epsilon)} \left\| \sum_{\tau=1}^K \phi_h^\tau \eta_h^\tau\left( e^{-\frac{V(s)}{\lambda}} \right) \right\|_{\Lambda_h^{-1}}^2 \ge 4\left( 2\log\frac{H|\mathcal{N}_{h+1}(\epsilon)|}{\delta} + d\log\left(1 + \frac{K}{\gamma}\right) \right) \right) \le \delta/H. \quad (\text{C.22})$$

Hence, combining (C.22) with (C.21) and let $\gamma = 1$, then for all $h \in [H]$, it holds that

$$\left\| \sum_{\tau=1}^K \phi_h^\tau \eta_h^\tau\left( e^{-\frac{\hat{V}_{h+1}^\lambda(s)}{\lambda}} \right) \right\|_{\Lambda_h^{-1}}^2 \le 8\left( 2\log\frac{H|\mathcal{N}_{h+1}(\epsilon)|}{\delta} + d\log(1 + K) + \frac{4\epsilon^2 K^2}{\lambda^2} \right), \quad (\text{C.23})$$

with probability at least $1 - \delta$. By Lemma C.4, recall $L = R_0 = 2Hd\sqrt{K\lambda}$ in this setting, we have

$$\log(|\mathcal{N}_{h+1}(\epsilon)|) \le d\log(1 + 4R_0/\epsilon) + d^2\log(1 + 8\lambda^2 d^{1/2}B^2/\epsilon^2). \quad (\text{C.24})$$

We then set $\epsilon = d\lambda/K \in (0, \lambda)$ and define $\beta' = \beta/\lambda e^{H/\lambda} = 16d\sqrt{(H/\lambda + \xi_{\text{KL}})}$ for brevity, then (C.24) can be bounded as:

$$\begin{aligned}
\log(|\mathcal{N}_{h+1}(\epsilon)|) &\le d\log(1 + 4R_0 K/d\lambda) + d^2\log(1 + 8\lambda^2 K^2 d^{-3/2} e^{\frac{2H}{\lambda}} \beta'^2) \\
&= d\log(1 + 4R_0 K/d\lambda) + d^2\log(e^{-2H/\lambda} + 8\lambda^2 K^2 d^{-3/2} \beta'^2) + 2d^2 H/\lambda \\
&\le 2d^2\log(8\lambda^2 K^2 d^{-3/2} \beta'^2) + 2d^2 H/\lambda.
\end{aligned}$$

Therefore, by combining the result with the inequality (C.23), we can get

$$\Big\| \sum_{\tau=1}^{K} \phi_h^\tau \eta_h^\tau \Big( e^{-\frac{\hat{V}_{h+1}^\lambda(s)}{\lambda}} \Big) \Big\|_{\Lambda_h^{-1}}^2 \leq 8 \Big( 2 \log \frac{H}{\delta} + 4d^2 H/\lambda + 4d^2 \log(8\lambda^2 K^2 d^{-3/2} \beta'^2) + 4d^2 + d \log(1+K) \Big)$$

$$\leq 8(4d^2 H/\lambda + 4d^2 \log(8\lambda^2 K^3 H d^{-3/2} \beta'^2/\delta)) \tag{C.25}$$

$$= 8(4d^2 H/\lambda + 4d^2 \log(8\lambda^2 K^3 H d^{-3/2}/\delta) + 4d^2 \log(\beta'^2))$$

$$\leq \beta'^2/4, \tag{C.26}$$

where (C.25) follows by the fact that $2\log\frac{H}{\delta} + 4d^2 + d\log(1+K) \leq 4d^2 \log(\frac{HK}{\delta})$, and (C.26) is held due to the fact that

$$\beta'^2/4 = 64d^2 \Big( H/\lambda + \log \frac{1024 d\lambda^2 K^3 H}{\delta} \Big)$$

$$= 8 \Big( 8d^2 H/\lambda + 4d^2 \log(8\lambda^2 K^3 H d^{-3/2}/\delta) + 4d^2 \log \frac{1024 d^{7/2} \lambda^2 K^3 H}{\delta} + 4d^2 \log(128) \Big)$$

$$\geq 8 \Big( 4d^2 H/\lambda + 4d^2 \log(8\lambda^2 K^3 H d^{-3/2}/\delta) + 4d^2 \log(\beta'^2) \Big), \tag{C.27}$$

where (C.27) holds by

$$\log(\beta'^2) = \log \Big( 256 d^2 \Big( H/\lambda + \log \frac{1024 d\lambda^2 K^3 H}{\delta} \Big) \Big)$$

$$\leq \log(256 d^2) + \Big( H/\lambda + \log \frac{1024 d\lambda^2 K^3 H}{\delta} \Big)$$

$$\leq \log(128) + \log \frac{1024 d^{7/2} \lambda^2 K^3 H}{\delta} + H/\lambda.$$

By the bound on (i), (ii), (iii), for all $h \in H$ and $(s,a) \in \mathcal{S} \times \mathcal{A}$, with probability at least $1 - \delta$, it holds that

$$|\mathcal{T}_h^\lambda \hat{V}_{h+1}^\lambda(s,a) - \langle \phi(s,a), \hat{\boldsymbol{w}}_h^\lambda \rangle| \leq \lambda \log \Big( 1 + e^{H/\lambda}(\sqrt{d} + \beta'/2) \sum_{i=1}^{d} \|\phi_i(s,a)\mathbf{1}_i\|_{\Lambda_h^{-1}} \Big)$$

$$\leq \lambda \log \Big( 1 + e^{H/\lambda} \beta' \sum_{i=1}^{d} \|\phi_i(s,a)\mathbf{1}_i\|_{\Lambda_h^{-1}} \Big) \tag{C.28}$$

$$\leq \beta \sum_{i=1}^{d} \|\phi_i(s,a)\mathbf{1}_i\|_{\Lambda_h^{-1}}, \tag{C.29}$$

where (C.28) follows by the fact that $\beta' \geq 2\sqrt{d}$, (C.29) follows by the fact that $\log(1+x) \leq x$ holds for any positive $x$. Thus, by Lemma C.1, we can upper bound the suboptimality gap as:

$$\text{SubOpt}(\hat{\pi}_1, s) \leq 2 \sup_{P \in \mathcal{U}^\lambda(P^0)} \sum_{h=1}^{H} \mathbb{E}^{\pi^\star, P} [\Gamma_h(s_h, a_h) | s_1 = s]$$

$$= 2\beta \sup_{P \in \mathcal{U}^\lambda(P^0)} \sum_{h=1}^{H} \mathbb{E}^{\pi^\star, P} \Big[ \sum_{i=1}^{d} \|\phi_i(s,a)\mathbf{1}_i\|_{\Lambda_h^{-1}} | s_1 = s \Big].$$

Therefore, we conclude the proof. $\qquad\square$

### C.1.3. PROOF OF THEOREM 5.1 - CASE WITH $\chi^2$ DIVERGENCE

For completeness, we present the R2PVI algorithm specific to the $\chi^2$ distance in Algorithm 5, which gives closed form solution of (4.4) and (4.5). Before the proof, we first present the bound on weights under $\chi^2$-divergence:

**Lemma C.5** (Bound of weights - $\chi^2$)**.** For any $h \in [H]$,

$$\|\hat{\boldsymbol{w}}_h^\lambda\|_2 \leq \sqrt{d} \Big( H + \frac{H^2}{2\lambda} \Big).$$

---

**Algorithm 5** Robust Regularized Pessimistic Value Iteration under $\chi^2$ distance (R2PVI-$\chi^2$)

---

**Require:** Dataset $\mathcal{D}$, regularizer $\lambda > 0, \gamma > 0$ and parameter $\beta$

1: init $\hat{V}_{H+1}^{\lambda}(\cdot) = 0$
2: **for** episode $h = H, \cdots, 1$ **do**
3:      $\boldsymbol{\Lambda}_h \leftarrow \sum_{\tau=1}^K \boldsymbol{\phi}(s_h^\tau, a_h^\tau)(\boldsymbol{\phi}(s_h^\tau, a_h^\tau))^\top + \gamma \mathbf{I}$
4:      $\hat{\mathbb{E}}^{\mu_{h,i}^0}[\hat{V}_{h+1}^\lambda(s)]_\alpha \leftarrow [\boldsymbol{\Lambda}_h^{-1}(\sum_{\tau=1}^K \boldsymbol{\phi}(s_h^\tau, a_h^\tau)^\top [\hat{V}_{h+1}^\lambda(s_{h+1}^\tau)]_\alpha)]_{[0,H]}$      ◁ Estimated by (4.4)
5:      $\hat{\mathbb{E}}^{\mu_{h,i}^0}[\hat{V}_{h+1}^\lambda(s)]_\alpha^2 \leftarrow [\boldsymbol{\Lambda}_h^{-1}(\sum_{\tau=1}^K \boldsymbol{\phi}(s_h^\tau, a_h^\tau)^\top [\hat{V}_{h+1}^\lambda(s_{h+1}^\tau)]_\alpha^2)]_{[0,H^2]}$      ◁ Estimated by (4.5)
6:      Estimate $\hat{w}_{h,i}^\lambda$ according to (4.6)
7:      $\Gamma_h(\cdot, \cdot) \leftarrow \beta \sum_{i=1}^d \|\phi_i(\cdot, \cdot)\|_{\boldsymbol{\Lambda}_h^{-1}}$
8:      $\hat{Q}_h^\lambda(\cdot, \cdot) \leftarrow \min\{\boldsymbol{\phi}(\cdot, \cdot)^\top(\boldsymbol{\theta}_h + \hat{\boldsymbol{w}}_h^\lambda) - \Gamma_h(\cdot, \cdot), H - h + 1\}^+$
9:      $\hat{\pi}_h(\cdot|\cdot) \leftarrow \arg\max_{\pi_h}\langle \hat{Q}_h^\lambda(\cdot, \cdot), \pi_h(\cdot|\cdot)\rangle_{\mathcal{A}}$ and $\hat{V}_h^\lambda(\cdot) \leftarrow \langle \hat{Q}_h^\lambda(\cdot, \cdot), \hat{\pi}_h(\cdot|\cdot)\rangle_{\mathcal{A}}$
10: **end for**

---

*Proof of Theorem 5.1 - $\chi^2$.* The R2PVI with $\chi^2$-divergence is presented in Algorithm 5. By the definition of $\mathcal{T}_h^\lambda, \hat{\mathcal{T}}_h^\lambda$, we have

$$
\begin{aligned}
&\mathcal{T}_h^\lambda \hat{V}_{h+1}^\lambda(s, a) - \langle \boldsymbol{\phi}(s, a), \hat{\boldsymbol{w}}_h^\lambda \rangle \\
&= \boldsymbol{\phi}(s, a)^\top (\boldsymbol{\theta}_h + \boldsymbol{w}_h^\lambda - \boldsymbol{\theta}_h - \hat{\boldsymbol{w}}_h^\lambda) \\
&= \sum_{i=1}^d \phi_i(s, a)(w_{h,i}^\lambda - \hat{w}_{h,i}') \\
&= \sum_{i=1}^d \phi_i(s, a)\Big[ \sup_{\alpha \in [0,H]} \Big\{ \mathbb{E}_{s \sim \mu_{h,i}^0}[\hat{V}_{h+1}^\lambda(s)]_\alpha + \frac{1}{4\lambda}(\mathbb{E}_{s \sim \mu_{h,i}^0}[\hat{V}_{h+1}^\lambda(s)]_\alpha)^2 - \frac{1}{4\lambda}\mathbb{E}_{s \sim \mu_{h,i}^0}[\hat{V}_{h+1}^\lambda(s)]_\alpha^2 \Big\} \\
&\quad - \sup_{\alpha \in [0,H]} \Big\{ \hat{\mathbb{E}}^{\mu_{h,i}^0}[\hat{V}_{h+1}^\lambda(s)]_\alpha + \frac{1}{4\lambda}(\hat{\mathbb{E}}^{\mu_{h,i}^0}[\hat{V}_{h+1}^\lambda(s)]_\alpha)^2 - \frac{1}{4\lambda}\hat{\mathbb{E}}^{\mu_{h,i}^0}[\hat{V}_{h+1}^\lambda(s)]_\alpha^2 \Big\} \Big].
\end{aligned}
\tag{C.30}
$$

To continue, for any $i \in [d]$, we denote

$$
\alpha_i = \arg\max_{\alpha \in [0,H]} \Big\{ \mathbb{E}_{s \sim \mu_{h,i}^0}[\hat{V}_{h+1}^\lambda(s)]_\alpha + \frac{1}{4\lambda}(\mathbb{E}_{s \sim \mu_{h,i}^0}[\hat{V}_{h+1}^\lambda(s)]_\alpha)^2 - \frac{1}{4\lambda}\mathbb{E}_{s \sim \mu_{h,i}^0}[\hat{V}_{h+1}^\lambda(s)]_\alpha^2 \Big\}.
$$

Hence, (C.30) can be further upper bounded as

$$
\begin{aligned}
&\mathcal{T}_h^\lambda \hat{V}_{h+1}^\lambda(s, a) - \langle \boldsymbol{\phi}(s, a), \hat{\boldsymbol{w}}_h^\lambda \rangle \\
&\leq \underbrace{\sum_{i=1}^d \phi_i(s, a)\big(\mathbb{E}_{s \sim \mu_{h,i}^0}[\hat{V}_{h+1}^\lambda(s)]_{\alpha_i} - \hat{\mathbb{E}}^{\mu_{h,i}^0}[\hat{V}_{h+1}^\lambda(s)]_{\alpha_i}\big)\Big( \frac{1}{4\lambda}\big(\mathbb{E}_{s \sim \mu_{h,i}^0}[\hat{V}_{h+1}^\lambda(s)]_{\alpha_i} + \hat{\mathbb{E}}^{\mu_{h,i}^0}[\hat{V}_{h+1}^\lambda(s)]_{\alpha_i}\big) + 1 \Big)}_{\text{(i)}} \\
&\quad - \underbrace{\sum_{i=1}^d \phi_i(s, a)\big(\mathbb{E}_{s \sim \mu_{h,i}^0}[\hat{V}_{h+1}^\lambda(s)]_{\alpha_i}^2 - \hat{\mathbb{E}}^{\mu_{h,i}^0}[\hat{V}_{h+1}^\lambda(s)]_{\alpha_i}^2\big)}_{\text{(ii)}}.
\end{aligned}
\tag{C.31}
$$

Next, we bound (i) and (ii), respectively.

**Bounding term (i).** We define

$$
\tilde{\mathbb{E}}^{\mu_{h,i}^0}[\hat{V}_{h+1}^\lambda(s)]_\alpha = \Big[ \arg\min_{\boldsymbol{w} \in R^d} \sum_{\tau=1}^K ([\hat{V}_{h+1}^\lambda(s_{h+1}^\tau)]_\alpha - \boldsymbol{\phi}(s_h^\tau, a_h^\tau)^\top \boldsymbol{w})^2 + \gamma \|\boldsymbol{w}\|_2^2 \Big]^i.
$$

Considering the gap between the $\hat{\mathbb{E}}^{\mu_{h,i}^0}[\hat{V}_{h+1}^\lambda(s)]_{\alpha_i}$ and $\tilde{\mathbb{E}}^{\mu_{h,i}^0}[\hat{V}_{h+1}^\lambda(s)]_{\alpha_i}$ due to the definition that $\hat{\mathbb{E}}^{\mu_{h,i}^0}[\hat{V}_{h+1}^\lambda(s)]_{\alpha_i} = [\tilde{\mathbb{E}}^{\mu_{h,i}^0}[\hat{V}_{h+1}^\lambda(s)]_{\alpha_i}]_{[0,H]}$, we eliminate the clip operator at first. We rewrite (i) as follows:

$$
\text{(i)} = \sum_{i=1}^d \phi_i(s,a)\big(\mathbb{E}_{s\sim\mu_{h,i}^0}[\hat{V}_{h+1}^\lambda(s)]_{\alpha_i} - \hat{\mathbb{E}}^{\mu_{h,i}^0}[\hat{V}_{h+1}^\lambda(s)]_{\alpha_i}\big)\Big(\frac{1}{4\lambda}\big(\mathbb{E}_{s\sim\mu_{h,i}^0}[\hat{V}_{h+1}^\lambda(s)]_{\alpha_i} + \hat{\mathbb{E}}^{\mu_{h,i}^0}[\hat{V}_{h+1}^\lambda(s)]_{\alpha_i}\big) + 1\Big)
$$

$$
= \sum_{i=1}^d \phi_i(s,a)\big(\mathbb{E}_{s\sim\mu_{h,i}^0}[\hat{V}_{h+1}^\lambda(s)]_{\alpha_i} - \tilde{\mathbb{E}}^{\mu_{h,i}^0}[\hat{V}_{h+1}^\lambda(s)]_{\alpha_i}\big)
$$

$$
\times \underbrace{\Big(\frac{1}{4\lambda}\big(\mathbb{E}_{s\sim\mu_{h,i}^0}[\hat{V}_{h+1}^\lambda(s)]_{\alpha_i} + \hat{\mathbb{E}}^{\mu_{h,i}^0}[\hat{V}_{h+1}^\lambda(s)]_{\alpha_i}\big) + 1\Big)\frac{\mathbb{E}^{\mu_{h,i}^0}[\hat{V}_{h+1}^\lambda(s)]_{\alpha_i} - \hat{\mathbb{E}}^{\mu_{h,i}^0}[\hat{V}_{h+1}^\lambda(s)]_{\alpha_i}}{\mathbb{E}^{\mu_{h,i}^0}[\hat{V}_{h+1}^\lambda(s)]_{\alpha_i} - \tilde{\mathbb{E}}^{\mu_{h,i}^0}[\hat{V}_{h+1}^\lambda(s)]_{\alpha_i}}}_{:=C_i}.
$$

We claim that $|C_i| \leq 1 + H/2\lambda, \forall i \in [H]$. We prove the claim by discussing the value of $\tilde{\mathbb{E}}^{\mu_{h,i}^0}[\hat{V}_{h+1}^\lambda(s)]_{\alpha_i}$ in the following three cases:

**Case I.** $\tilde{\mathbb{E}}^{\mu_{h,i}^0}[\hat{V}_{h+1}^\lambda(s)]_{\alpha_i} \leq 0$. By the fact that $\mathbb{E}_{s\sim\mu_{h,i}^0}[\hat{V}_{h+1}^\lambda(s)]_{\alpha_i} \leq H$, we have:

$$
|C_i| = \left|\Big(\frac{1}{4\lambda}\mathbb{E}_{s\sim\mu_{h,i}^0}[\hat{V}_{h+1}^\lambda(s)]_{\alpha_i} + 1\Big)\frac{\mathbb{E}_{s\sim\mu_{h,i}^0}[\hat{V}_{h+1}^\lambda(s)]_{\alpha_i}}{\mathbb{E}_{s\sim\mu_{h,i}^0}[\hat{V}_{h+1}^\lambda(s)]_{\alpha_i} - \tilde{\mathbb{E}}^{\mu_{h,i}^0}[\hat{V}_{h+1}^\lambda(s)]_{\alpha_i}}\right| \leq 1 + H/4\lambda,
$$

where the equality holds by $\frac{1}{4\lambda}\mathbb{E}_{s\sim\mu_{h,i}^0}[\hat{V}_{h+1}^\lambda(s)]_{\alpha_i} + 1 \leq 1 + H/4\lambda$. Hence the claim holds by **Case I**.

**Case II.** $0 \leq \tilde{\mathbb{E}}^{\mu_{h,i}^0}[\hat{V}_{h+1}^\lambda(s)]_{\alpha_i} \leq H$. The claim holds trivially, as we have:

$$
|C_i| = \frac{1}{4\lambda}\big(\mathbb{E}_{s\sim\mu_{h,i}^0}[\hat{V}_{h+1}^\lambda(s)]_{\alpha_i} + \tilde{\mathbb{E}}^{\mu_{h,i}^0}[\hat{V}_{h+1}^\lambda(s)]_{\alpha_i}\big) + 1 \leq 1 + H/2\lambda.
$$

Hence, we conclude the claim.

**Case III.** $\tilde{\mathbb{E}}^{\mu_{h,i}^0}[\hat{V}_{h+1}^\lambda(s)]_{\alpha_i} > H$. Notice that

$$
|C_i| = \left|\Big(\frac{1}{4\lambda}\big(\mathbb{E}_{s\sim\mu_{h,i}^0}[\hat{V}_{h+1}^\lambda(s)]_{\alpha_i} + H\big) + 1\Big)\frac{\mathbb{E}_{s\sim\mu_{h,i}^0}[\hat{V}_{h+1}^\lambda(s)]_{\alpha_i} - H}{\mathbb{E}_{s\sim\mu_{h,i}^0}[\hat{V}_{h+1}^\lambda(s)]_{\alpha_i} - \tilde{\mathbb{E}}^{\mu_{h,i}^0}[\hat{V}_{h+1}^\lambda(s)]_{\alpha_i}}\right|
$$

$$
= \Big(\frac{1}{4\lambda}\big(\mathbb{E}_{s\sim\mu_{h,i}^0}[\hat{V}_{h+1}^\lambda(s)]_{\alpha_i} + H\big) + 1\Big)\frac{H - \mathbb{E}_{s\sim\mu_{h,i}^0}[\hat{V}_{h+1}^\lambda(s)]_{\alpha_i}}{\tilde{\mathbb{E}}^{\mu_{h,i}^0}[\hat{V}_{h+1}^\lambda(s)]_{\alpha_i} - \mathbb{E}_{s\sim\mu_{h,i}^0}[\hat{V}_{h+1}^\lambda(s)]_{\alpha_i}}
$$

$$
\leq H/2\lambda + 1, \tag{C.32}
$$

where (C.32) holds by the fact that $\tilde{\mathbb{E}}^{\mu_{h,i}^0}[\hat{V}_{h+1}^\lambda(s)]_{\alpha_i} > H$.

With the upper bound for $C_i$, we can upper bound (i) as

$$
|\text{(i)}| = \Big|\sum_{i=1}^d \phi_i(s,a)(\mathbb{E}_{s\sim\mu_{h,i}^0}[\hat{V}_{h+1}^\lambda(s)]_{\alpha_i} - \tilde{\mathbb{E}}^{\mu_{h,i}^0}[\hat{V}_{h+1}^\lambda(s)]_{\alpha_i})C_i\Big|
$$

$$
= \Big|\gamma\sum_{i=1}^d \phi_i(s,a)\mathbf{1}_i\mathbf{\Lambda}_h^{-1}\mathbb{E}^{\mu_h^0}[\hat{V}_h^\lambda(s)]_{\alpha_i}C_i + \sum_{i=1}^d \phi_i(s,a)\mathbf{1}_i\mathbf{\Lambda}_h^{-1}\sum_{\tau=1}^K \phi(s_h^\tau,a_h^\tau)\eta_h^\tau([\hat{V}_{h+1}^\lambda]_{\alpha_i})C_i\Big|
$$

$$
\leq (1 + H/2\lambda)\sum_{i=1}^d \|\phi_i(s,a)\mathbf{1}_i\|_{\mathbf{\Lambda}_h^{-1}}\Big(\gamma H + \Big\|\sum_{\tau=1}^K \phi(s_h^\tau,a_h^\tau)\eta_h^\tau([\hat{V}_{h+1}^\lambda]_{\alpha_i})\Big\|_{\mathbf{\Lambda}_h^{-1}}\Big). \tag{C.33}
$$

**Bounding term (ii).** Similar to bounding (i), we can deduce that:

$$
|(\text{ii})| = \Big| \sum_{i=1}^{d} \phi_i(s,a)(\mathbb{E}_{s\sim\mu_{h,i}^0}[\hat{V}_{h+1}^{\lambda}(s)]_{\alpha_i}^2 - \hat{\mathbb{E}}^{\mu_{h,i}^0}[\hat{V}_{h+1}^{\lambda}(s)]_{\alpha_i}^2) \Big|
$$

$$
\leq \gamma H^2 \sum_{i=1}^{d} \|\phi_i(s,a)\mathbf{1}_i\|_{\mathbf{\Lambda}_h^{-1}} + \sum_{i=1}^{d} \|\phi_i(s,a)\mathbf{1}_i\|_{\mathbf{\Lambda}_h^{-1}} \Big\| \sum_{\tau=1}^{K} \phi(s_h^\tau, a_h^\tau)\eta_h^\tau([\hat{V}_{h+1}^{\lambda}]_{\alpha_i}^2) \Big\|_{\mathbf{\Lambda}_h^{-1}}, \tag{C.34}
$$

where (C.34) follows by the Cauchy Schwartz inequality and the fact that $\mathbb{E}_{s\sim\mu_{h,i}^0}[\hat{V}_h^{\lambda}(s)]_{\alpha_i}^2 \leq H^2, \forall i \in [d]$. Hence combining (C.31), (C.33) and (C.34), we have

$$
\mathcal{T}_h^{\lambda}\hat{V}_{h+1}^{\lambda}(s,a) - \langle \phi(s,a), \hat{\boldsymbol{w}}_h^{\lambda}\rangle
$$

$$
\leq (1 + H/2\lambda)\Big(\gamma H \sum_{i=1}^{d} \|\phi_i(s,a)\mathbf{1}_i\|_{\mathbf{\Lambda}_h^{-1}} + \sum_{i=1}^{d} \|\phi_i(s,a)\mathbf{1}_i\|_{\mathbf{\Lambda}_h^{-1}} \Big\| \sum_{\tau=1}^{K} \phi(s_h^\tau, a_h^\tau)\eta_h^\tau([\hat{V}_{h+1}^{\lambda}]_{\alpha_i'}) \Big\|_{\mathbf{\Lambda}_h^{-1}} \Big)
$$

$$
+ \gamma H^2 \sum_{i=1}^{d} \|\phi_i(s,a)\mathbf{1}_i\|_{\mathbf{\Lambda}_h^{-1}} + \sum_{i=1}^{d} \|\phi_i(s,a)\mathbf{1}_i\|_{\mathbf{\Lambda}_h^{-1}} \Big\| \sum_{\tau=1}^{K} \phi(s_h^\tau, a_h^\tau)\eta_h^\tau([\hat{V}_{h+1}^{\lambda}]_{\alpha_i'}^2) \Big\|_{\mathbf{\Lambda}_h^{-1}}.
$$

On the other hand, we can similarly deduce that there exists $\alpha_i'$ s.t.

$$
\langle \phi(s,a), \hat{\boldsymbol{w}}_h^{\lambda}\rangle - \mathcal{T}_h^{\lambda}\hat{V}_{h+1}^{\lambda}(s,a)
$$

$$
\leq (1 + H/2\lambda)\Big(\gamma H \sum_{i=1}^{d} \|\phi_i(s,a)\mathbf{1}_i\|_{\mathbf{\Lambda}_h^{-1}} + \sum_{i=1}^{d} \|\phi_i(s,a)\mathbf{1}_i\|_{\mathbf{\Lambda}_h^{-1}} \Big\| \sum_{\tau=1}^{K} \phi(s_h^\tau, a_h^\tau)\eta_h^\tau([\hat{V}_{h+1}^{\lambda}]_{\alpha_i'}) \Big\|_{\mathbf{\Lambda}_h^{-1}} \Big)
$$

$$
+ \gamma H^2 \sum_{i=1}^{d} \|\phi_i(s,a)\mathbf{1}_i\|_{\mathbf{\Lambda}_h^{-1}} + \sum_{i=1}^{d} \|\phi_i(s,a)\mathbf{1}_i\|_{\mathbf{\Lambda}_h^{-1}} \Big\| \sum_{\tau=1}^{K} \phi(s_h^\tau, a_h^\tau)\eta_h^\tau([\hat{V}_{h+1}^{\lambda}]_{\alpha_i'}^2) \Big\|_{\mathbf{\Lambda}_h^{-1}}.
$$

Then for all $i \in [d]$, there exists $\hat{\alpha}_i \in \{\alpha_i, \alpha_i'\}$, such that

$$
|\mathcal{T}_h^{\lambda}\hat{V}_{h+1}^{\lambda}(s,a) - \langle \phi(s,a), \hat{\boldsymbol{w}}_h^{\lambda}\rangle|
$$

$$
\leq (1 + H/2\lambda)\Big(\gamma H \sum_{i=1}^{d} \|\phi_i(s,a)\mathbf{1}_i\|_{\mathbf{\Lambda}_h^{-1}}\Big) + \gamma H^2 \sum_{i=1}^{d} \|\phi_i(s,a)\mathbf{1}_i\|_{\mathbf{\Lambda}_h^{-1}}
$$

$$
+ \sum_{i=1}^{d} \|\phi_i(s,a)\mathbf{1}_i\|_{\mathbf{\Lambda}_h^{-1}} \Big( (1 + H/2\lambda)\Big\| \sum_{\tau=1}^{K} \phi(s_h^\tau, a_h^\tau)\eta_h^\tau([\hat{V}_{h+1}^{\lambda}]_{\hat{\alpha}_i}) \Big\|_{\mathbf{\Lambda}_h^{-1}} + \Big\| \sum_{\tau=1}^{K} \phi(s_h^\tau, a_h^\tau)\eta_h^\tau([\hat{V}_{h+1}^{\lambda}]_{\hat{\alpha}_i}^2) \Big\|_{\mathbf{\Lambda}_h^{-1}} \Big).
$$

Now it remains to bound the terms

$$
\underbrace{\Big\| \sum_{\tau=1}^{K} \phi(s_h^\tau, a_h^\tau)\eta_h^\tau([\hat{V}_{h+1}^{\lambda}]_{\hat{\alpha}_i}) \Big\|_{\mathbf{\Lambda}_h^{-1}}}_{(\text{iii})} \text{ and } \underbrace{\Big\| \sum_{\tau=1}^{K} \phi(s_h^\tau, a_h^\tau)\eta_h^\tau([\hat{V}_{h+1}^{\lambda}]_{\hat{\alpha}_i}^2) \Big\|_{\mathbf{\Lambda}_h^{-1}}}_{(\text{iv})}.
$$

Similar to the proof in KL divergence, we aim to find a union function class $\mathcal{V}_{h+1}(R_0, B_0, \lambda)$, which holds uniformly that $\hat{V}_{h+1}^{\lambda} \in \mathcal{V}_{h+1}(R_0, B_0, \lambda)$, here for all $h \in [H]$, the function class is defined as:

$$
\mathcal{V}_h(R_0, B_0, \lambda) = \{V_h(x; \boldsymbol{\theta}, \beta, \mathbf{\Lambda}) : \mathcal{S} \to [0, H], \|\boldsymbol{\theta}\|_2 \leq R_0, \beta \in [0, B_0], \gamma_{\min}(\mathbf{\Lambda}_h) \geq \gamma\},
$$

where $V_h(x; \boldsymbol{\theta}, \beta, \mathbf{\Lambda}) = \max_{a\in\mathcal{A}}[\phi(s,a)^\top \boldsymbol{\theta} - \beta \sum_{i=1}^{d} \|\phi_i(s,a)\|_{\mathbf{\Lambda}_h^{-1}}]_{[0,H-h+1]}$. By Lemma C.5, when we set $R_0 = \sqrt{d}(H + H^2/2\lambda), B_0 = \beta = 8dH^2(1 + 1/\lambda)\sqrt{\xi_{\chi^2}}$, it suffices to show that $\hat{V}_{h+1}^{\lambda} \in \mathcal{V}_{h+1}(R_0, B_0, \gamma)$. Next we aim to find a union cover of the $\mathcal{V}_{h+1}(R_0, B_0, \gamma)$, hence the term (iii) and (iv) can be upper bounded. Let $\mathcal{N}_h(\epsilon; R_0, B_0, \gamma)$ be the minimum $\epsilon$-cover of $\mathcal{V}_h(R, B, \lambda)$ with respect to the supreme norm, $\mathcal{N}_h([0, H])$ be the minimum $\epsilon$-cover of $[0, H]$

respectively. In other words, for any function $V \in \mathcal{V}_h(R, B, \lambda), \alpha \in [0, H]$, there exists a function $V' \in \mathcal{V}_h(R, B, \lambda)$ and a real number $\alpha_\epsilon \in [0, H]$ such that:

$$\sup_{s \in \mathcal{S}} |V(s) - V'(s)| \le \epsilon, |\alpha - \alpha_\epsilon| \le \epsilon.$$

Recall the definition of (iii) and (iv). By Cauchy-Schwartz inequality and the fact that $\|a + b\|^2_{\Lambda_h^{-1}} \le 2\|a\|^2_{\Lambda_h^{-1}} + 2\|b\|^2_{\Lambda_h^{-1}}$, we have

$$(\text{iii})^2 \le 2\Big\| \sum_{\tau=1}^{K} \phi(s_h^\tau, a_h^\tau) \eta_h^\tau([\hat{V}_{h+1}^\lambda]_{\alpha_\epsilon}) \Big\|^2_{\Lambda_h^{-1}} + 2\Big\| \sum_{\tau=1}^{K} \phi(s_h^\tau, a_h^\tau) \eta_h^\tau([\hat{V}_{h+1}^\lambda]_{\hat{\alpha}} - [\hat{V}_{h+1}^\lambda]_{\alpha_\epsilon}) \Big\|^2_{\Lambda_h^{-1}}$$

$$\le 4\Big\| \sum_{\tau=1}^{K} \phi(s_h^\tau, a_h^\tau) \eta_h^\tau([V_{h+1}']_{\alpha_\epsilon}) \Big\|^2_{\Lambda_h^{-1}} + 4\Big\| \sum_{\tau=1}^{K} \phi(s_h^\tau, a_h^\tau) \eta_h^\tau([\hat{V}_{h+1}^\lambda]_{\alpha_\epsilon} - [V_{h+1}']_{\alpha_\epsilon}) \Big\|^2_{\Lambda_h^{-1}} + \frac{2\epsilon^2 K^2}{\gamma}, \quad \text{(C.35)}$$

where (C.35) follows by the fact that

$$2\Big\| \sum_{\tau=1}^{K} \phi(s_h^\tau, a_h^\tau) \eta_h^\tau([\hat{V}_{h+1}^\lambda]_{\hat{\alpha}} - [\hat{V}_{h+1}^\lambda]_{\alpha_\epsilon}) \Big\|^2_{\Lambda_h^{-1}} \le 2\epsilon^2 \sum_{\tau=1, \tau'=1}^{K} |\phi_h^\tau \Lambda_h^{-1} \phi_h^{\tau'}| \le \frac{2\epsilon^2 K^2}{\gamma}.$$

Meanwhile, by the fact that $|[\hat{V}_{h+1}^\lambda]_{\alpha_\epsilon} - [V_{h+1}']_{\alpha_\epsilon}| \le |\hat{V}_{h+1}^\lambda - V_{h+1}'|$, we have

$$4\Big\| \sum_{\tau=1}^{K} \phi(s_h^\tau, a_h^\tau) \eta_h^\tau([\hat{V}_{h+1}^\lambda]_{\alpha_\epsilon} - [V_{h+1}']_{\alpha_\epsilon}) \Big\|^2_{\Lambda_h^{-1}}$$

$$\le 4 \sum_{\tau=1, \tau'=1}^{K} |\phi_h^\tau \Lambda_h^{-1} \phi_h^{\tau'}| \max |\eta_h^\tau([\hat{V}_{h+1}^\lambda]_{\alpha_\epsilon} - [V_{h+1}']_{\alpha_\epsilon})|^2$$

$$\le 4\epsilon^2 \sum_{\tau=1, \tau'=1}^{K} |\phi_h^\tau \Lambda_h^{-1} \phi_h^{\tau'}|$$

$$\le \frac{4\epsilon^2 K^2}{\gamma}.$$

By applying the above two inequalities and the union bound into (C.35), we have

$$(\text{iii})^2 \le 4 \sup_{V' \in \mathcal{N}_h(\epsilon; R_0, B_0, \gamma), \alpha_\epsilon \in \mathcal{N}_h([0, H])} \Big\| \sum_{\tau=1}^{K} \phi(s_h^\tau, a_h^\tau) \eta_h^\tau([V_{h+1}']_{\alpha_\epsilon}) \Big\|^2_{\Lambda_h^{-1}} + \frac{6\epsilon^2 K^2}{\gamma}.$$

By Lemma F.3, applying a union bound over $\mathcal{N}_h(\epsilon; R_0, B_0, \gamma)$ and $\mathcal{N}_h([0, H])$, with probability at least $1 - \delta/2H$, we have

$$4 \sup_{V' \in \mathcal{N}_h(\epsilon; R_0, B_0, \gamma), \alpha_\epsilon \in \mathcal{N}_h([0, H])} \Big\| \sum_{\tau=1}^{K} \phi(s_h^\tau, a_h^\tau) \eta_h^\tau([V_{h+1}']_{\alpha_\epsilon}) \Big\|^2_{\Lambda_h^{-1}} + \frac{6\epsilon^2 K^2}{\gamma}$$

$$\le 4H^2 \Big( 2 \log \frac{2H |\mathcal{N}_h(\epsilon; R_0, B_0, \gamma)| |\mathcal{N}_h([0, H])|}{\delta} + d \log(1 + K/\gamma) \Big) + \frac{6\epsilon^2 K^2}{\gamma}.$$

Similarly, by the fact that $\|a + b\|^2_{\Lambda_h^{-1}} \le 2\|a\|^2_{\Lambda_h^{-1}} + 2\|b\|^2_{\Lambda_h^{-1}}$, noticing (iv) has the almost same form as (iii), we have

$$(\text{iv})^2 \le 2\Big\| \sum_{\tau=1}^{K} \phi(s_h^\tau, a_h^\tau) \eta_h^\tau([\hat{V}_{h+1}^\lambda]^2_{\alpha_\epsilon}) \Big\|^2_{\Lambda_h^{-1}} + 2\Big\| \sum_{\tau=1}^{K} \phi(s_h^\tau, a_h^\tau) \eta_h^\tau([\hat{V}_{h+1}^\lambda]^2_{\hat{\alpha}} - [\hat{V}_{h+1}^\lambda]^2_{\alpha_\epsilon}) \Big\|^2_{\Lambda_h^{-1}}$$

$$\le 4\Big\| \sum_{\tau=1}^{K} \phi(s_h^\tau, a_h^\tau) \eta_h^\tau([V_{h+1}']^2_{\alpha_\epsilon}) \Big\|^2_{\Lambda_h^{-1}} + \frac{24 H^2 \epsilon^2 K^2}{\gamma}, \quad \text{(C.36)}$$

where (C.36) follows by the fact that

$$2\Big\| \sum_{\tau=1}^{K} \phi(s_h^\tau, a_h^\tau) \eta_h^\tau([\hat{V}_{h+1}^\lambda]_{\hat{\alpha}}^2 - [\hat{V}_{h+1}^\lambda]_{\alpha_\epsilon}^2) \Big\|_{\Lambda_h^{-1}}^2 \le 8H^2\epsilon^2 \sum_{\tau=1,\tau'=1}^{K} |\phi_h^\tau \Lambda_h^{-1} \phi_h^{\tau'}| \le \frac{8H^2\epsilon^2 K^2}{\gamma},$$

and

$$4\| \sum_{\tau=1}^{K} \phi(s_h^\tau, a_h^\tau) \eta_h^\tau([\hat{V}_{h+1}^\lambda]_{\alpha_\epsilon}^2 - [V_{h+1}']_{\alpha_\epsilon}^2) \|_{\Lambda_h^{-1}}^2$$

$$\le 4 \sum_{\tau=1,\tau'=1}^{K} |\phi_h^\tau \Lambda_h^{-1} \phi_h^{\tau'}| \max |\eta_h^\tau([\hat{V}_{h+1}^\lambda]_{\alpha_\epsilon} - [V_{h+1}']_{\alpha_\epsilon})|^2$$

$$\le 16H^2\epsilon^2 \sum_{\tau=1,\tau'=1}^{K} |\phi_h^\tau \Lambda_h^{-1} \phi_h^{\tau'}|$$

$$\le \frac{16H^2\epsilon^2 K^2}{\gamma}.$$

We apply the union bound and Lemma F.3, with probability at least $1 - \delta/2H$

$$(\text{iv})^2 \le 4H^4\Big(2\log \frac{2H|\mathcal{N}_h(\epsilon; R_0, B_0, \gamma)\|\mathcal{N}_h([0, H])|}{\delta} + d\log(1 + K/\gamma)\Big) + \frac{24H^2\epsilon^2 K^2}{\gamma}.$$

Therefore, with probability at least $1 - \delta$,

$$|\mathcal{T}_h^\lambda \hat{V}_{h+1}^\lambda(s, a) - \langle \phi(s, a), \hat{\boldsymbol{w}}_h^\lambda \rangle| \tag{C.37}$$

$$\le \gamma H(1 + H/2\lambda) \sum_{i=1}^{d} \|\phi_i(s, a)\mathbf{1}_i\|_{\Lambda_h^{-1}} + \gamma H^2 \sum_{i=1}^{d} \|\phi_i(s, a)\mathbf{1}_i\|_{\Lambda_h^{-1}}$$

$$+ \sum_{i=1}^{d} \|\phi_i(s, a)\mathbf{1}_i\|_{\Lambda_h^{-1}} \Big((1 + H/2\lambda)\Big\| \sum_{\tau=1}^{K} \phi(s_h^\tau, a_h^\tau)\eta_h^\tau([\hat{V}_{h+1}^\lambda]_{\hat{\alpha}_i}) \Big\|_{\Lambda_h^{-1}} + \Big\| \sum_{\tau=1}^{K} \phi(s_h^\tau, a_h^\tau)\eta_h^\tau([\hat{V}_{h+1}^\lambda]_{\hat{\alpha}_i}^2) \Big\|_{\Lambda_h^{-1}}\Big)$$

$$\le \sum_{i=1}^{d} \|\phi_i(s, a)\mathbf{1}_i\|_{\Lambda_h^{-1}} \Big[\gamma H(1 + H + H/2\lambda)$$

$$+ (1 + H/2\lambda)\sqrt{4H^2\Big(2\log \frac{2H|\mathcal{N}_h(\epsilon; R_0, B_0, \gamma)\|\mathcal{N}_h([0, H])|}{\delta} + d\log(1 + K/\gamma)\Big) + \frac{6\epsilon^2 K^2}{\gamma}}$$

$$+ \sqrt{4H^4\Big(2\log \frac{2H|\mathcal{N}_h(\epsilon; R_0, B_0, \gamma)\|\mathcal{N}_h([0, H])|}{\delta} + d\log(1 + K/\gamma)\Big) + \frac{24H^2\epsilon^2 K^2}{\gamma}}\Big]. \tag{C.38}$$

By the fact that $R_0 = \sqrt{d}(H + H^2/2\lambda)$, Lemma F.1 and Lemma F.2, we can upper bound the term $|\mathcal{N}_h(\epsilon; R_0, B_0, \gamma)|$ and $|\mathcal{N}_h([0, H])|$ as follows:

$$\log |\mathcal{N}_h(\epsilon; R_0, B_0, \gamma)| \le d\log(1 + 4R_0/\epsilon) + d^2\log(1 + 8d^{1/2}B^2/\gamma\epsilon^2), \log |\mathcal{N}_h([0, H])| \le \log(3H/\epsilon).$$

We set $\epsilon = \frac{1}{K}, \gamma = 1$, with the upper bound above, we have

$$4H^2\Big(2\log \frac{2H|\mathcal{N}_h(\epsilon; R_0, B_0, \gamma)\|\mathcal{N}_h([0, H])|}{\delta} + d\log(1 + K/\gamma)\Big) + \frac{6\epsilon^2 K^2}{\gamma}$$

$$\le 4H^2\Big(2\log \frac{6H^2 K}{\delta} + d\log(1 + K) + d\log(1 + d^{1/2}(1 + H/2\lambda)K) + d^2\log(1 + d^{1/2}B^2 K^2) + 3/2\Big)$$

$$\le 4H^2\Big(2d\log \frac{6H^2 K}{\delta} + 2d\log(K) + d\log(d^{1/2}(1 + H/2\lambda)K) + 2d^2\log d^{1/2}B^2 K^2\Big)$$

$$\leq 8H^2 d^2 \Big( \log \frac{6H^2 K}{\delta} + \log(K) + \log(d^{1/2}(1 + H/2\lambda)K) + \log 8d^{1/2} B^2 K^2 \Big)$$
$$= 8H^2 d^2 (\log 48 K^5 H^2 B^2 d(1 + H/2\lambda)/\delta).$$

Similarly, we can upper bound the third term in (C.38) as follows:

$$4H^4 \Big( 2\log \frac{2H|\mathcal{N}_h(\epsilon; R_0, B_0, \gamma)| \|\mathcal{N}_h([0, H])|}{\delta} + d\log(1 + K/\gamma) \Big) + \frac{24H^2 \epsilon^2 K^2}{\gamma}$$
$$= H^2 \Big( 4H^2 (2\log \frac{2H|\mathcal{N}_h(\epsilon; R_0, B_0, \gamma)| \|\mathcal{N}_h([0, H])|}{\delta} + d\log(1 + K/\gamma)) + \frac{24\epsilon^2 K^2}{\gamma} \Big)$$
$$\leq 8H^4 d^2 (\log 48 K^5 H^2 B^2 d(1 + H/2\lambda)/\delta).$$

Hence, we apply this bound into the (C.38), we have

$$|\langle \boldsymbol{\phi}(s, a), \hat{\boldsymbol{w}}_h^\lambda \rangle - \mathcal{T}_h^\lambda \hat{V}_{h+1}^\lambda(s, a)|$$
$$\leq \sum_{i=1}^d \|\boldsymbol{\phi}(s, a)\mathbf{1}_i\|_{\boldsymbol{\Lambda}_h^{-1}} (H(1 + H + H/2\lambda)$$
$$\quad + (1 + H/2\lambda + H)\sqrt{8H^2 d^2 (\log 48 K^5 H^2 B^2 d(1 + H/2\lambda)/\delta)}$$
$$\leq \sum_{i=1}^d \|\boldsymbol{\phi}(s, a)\mathbf{1}_i\|_{\boldsymbol{\Lambda}_h^{-1}} 2(H/\lambda + H)\sqrt{8H^2 d^2 (\log 48 K^5 H^2 B^2 d(1 + H/2\lambda)/\delta)} \tag{C.39}$$
$$\leq \sum_{i=1}^d \|\boldsymbol{\phi}(s, a)\mathbf{1}_i\|_{\boldsymbol{\Lambda}_h^{-1}} 2(H/\lambda + H)Hd\sqrt{8(\log 192 K^5 H^6 d^3 (1 + H/2\lambda)^3/\delta) + \log \xi_{\chi^2}}$$
$$= \sum_{i=1}^d \|\boldsymbol{\phi}(s, a)\mathbf{1}_i\|_{\boldsymbol{\Lambda}_h^{-1}} 2(H/\lambda + H)Hd\sqrt{8(\xi_{\chi^2} + \log \xi_{\chi^2})}$$
$$\leq \beta \sum_{i=1}^d \|\boldsymbol{\phi}(s, a)\mathbf{1}_i\|_{\boldsymbol{\Lambda}_h^{-1}}, \tag{C.40}$$

where (C.39) comes from the fact that $1 + 1/\lambda \leq 1 + H/2\lambda$, the (C.40) comes from the fact that $\log \xi_{\chi^2} \leq \xi_{\chi^2}$. Hence, the prerequisite is satisfied in Lemma C.1, we can upper bound the suboptimality gap as:

$$\text{SubOpt}(\hat{\pi}, s, \lambda) \leq 2 \sup_{P \in \mathcal{U}^\lambda(P^0)} \sum_{h=1}^H \mathbb{E}^{\pi^\star, P} \big[ \Gamma_h(s_h, a_h) | s_1 = s \big]$$
$$= 2\beta \sup_{P \in \mathcal{U}^\lambda(P^0)} \sum_{h=1}^H \mathbb{E}^{\pi^\star, P} \Big[ \sum_{i=1}^d \|\phi_i(s, a)\mathbf{1}_i\|_{\boldsymbol{\Lambda}_h^{-1}} | s_1 = s \Big].$$

This concludes the proof. □

### C.2. Proof of Corollary 5.4

*Proof.* The proof follows the argument in (F.15) and (F.16) of (Blanchet et al., 2024). Specifically, we denote

$$\boldsymbol{\Lambda}_{h,i}^P = \mathbb{E}^{\pi^\star, P} \big[ (\phi_i(s_h, a_h)\mathbf{1}_i)(\phi_i(s_h, a_h)\mathbf{1}_i)^\top | s_1 = s \big], \quad \forall(h, i, P) \in [H] \times [d] \times \mathcal{U}^\lambda(P^0). \tag{C.41}$$

By Assumption 5.3, setting $\gamma = 1$, we have

$$\sup_{P \in \mathcal{U}^\lambda(P^0)} \sum_{h=1}^H \mathbb{E}^{\pi^\star, P} \Big[ \sum_{i=1}^d \|\phi_i(s_h, a_h)\mathbf{1}_i\|_{\boldsymbol{\Lambda}_h^{-1}} | s_1 = s \Big]$$
$$= \sup_{P \in \mathcal{U}^\lambda(P^0)} \sum_{h=1}^H \sum_{i=1}^d \mathbb{E}^{\pi^\star, P} \Big[ \sqrt{\text{Tr} \big( (\phi_i(s_h, a_h)\mathbf{1}_i)(\phi_i(s_h, a_h)\mathbf{1}_i)^\top \boldsymbol{\Lambda}_h^{-1} \big)} | s_1 = s \Big]$$

$$\leq \sup_{P\in\mathcal{U}^\lambda(P^0)} \sum_{h=1}^H \sum_{i=1}^d \sqrt{\mathrm{Tr}\left(\mathbb{E}^{\pi^\star,P}\left[(\phi_i(s_h,a_h)\mathbf{1}_i)(\phi_i(s_h,a_h)\mathbf{1}_i)^\top|s_1=s\right]\mathbf{\Lambda}_h^{-1}\right)} \tag{C.42}$$

$$\leq \sup_{P\in\mathcal{U}^\lambda(P^0)} \sum_{h=1}^H \sum_{i=1}^d \sqrt{\mathrm{Tr}\left(\mathbf{\Lambda}_{h,i}^P\cdot\left(\mathbf{I}+K\cdot c^\dagger\cdot\mathbf{\Lambda}_{h,i}^P\right)^{-1}\right)} \tag{C.43}$$

$$= \sup_{P\in\mathcal{U}^\lambda(P^0)} \sum_{h=1}^H \sum_{i=1}^d \sqrt{\frac{(\mathbb{E}^{\pi^\star,P}[\phi_i(s_h,a_h)|s_1=s])^2}{1+c^\dagger\cdot K\cdot(\mathbb{E}^{\pi^\star,P}[\phi_i(s_h,a_h)|s_1=s])^2}}$$

$$\leq \sup_{P\in\mathcal{U}^\lambda(P^0)} \sum_{h=1}^H \sum_{i=1}^d \sqrt{\frac{1}{c^\dagger\cdot K}} \tag{C.44}$$

$$= \frac{dH}{\sqrt{c^\dagger K}},$$

where (C.42) is due to the Jensen's inequality, (C.43) holds by the definition in (C.41) and Assumption 5.3, (C.44) holds by the fact that the only nonzero element of $\mathbf{\Lambda}_{h,i}^P$ is the $i$-th diagonal element. Thus, by Theorem 5.1, with probability at least $1-\delta$, for any $s\in\mathcal{S}$ the suboptimality can be upper bounded as:

$$\mathrm{SubOpt}(\hat{\pi},s,\lambda) \leq \beta \sup_{P\in\mathcal{U}^\lambda(P^0)} \sum_{h=1}^H \mathbb{E}^{\pi^\star,P}\Big[\sum_{i=1}^d \|\phi_i(s,a)\mathbf{1}_i\|_{\mathbf{\Lambda}_h^{-1}}\Big|s_1=s\Big] \leq \frac{\beta dH}{\sqrt{c^\dagger K}},$$

where

$$\beta = \begin{cases} 16Hd\sqrt{\xi_{\mathrm{TV}}}, & \text{if D is TV;} \\ 16d\lambda e^{H/\lambda}\sqrt{(H/\lambda+\xi_{\mathrm{KL}})}, & \text{if D is KL;} \\ 8dH^2(1+1/\lambda)\sqrt{\xi_{\chi^2}}, & \text{if D is } \chi^2. \end{cases}$$

Hence, we conclude the proof. $\qquad\square$

## D. Proof of the Information-Theoretic Lower Bound

In this section, we prove the information-theoretic lower bound. We first introduce the construction of hard instances in Appendix D.1, then we prove Theorem 5.5 in Appendix D.2

### D.1. Construction of Hard Instances

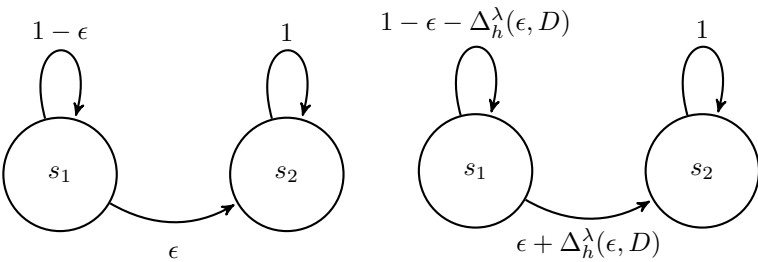

(a) The source MDP environment.    (b) The target MDP environment.

*Figure 3.* The nominal environment and the worst case environment. The value on each arrow represents the transition probability. The MDP has two states, $s_1$ and $s_2$, and $H$ steps. For he nominal environment on the left, the $s_1$ is the good state where the transition is determined by an error term $\epsilon$, and $s_2$ is a fail state with reward 0 and only transitions to itself. The worst case environment on the right is obtained by perturbing the transition probability at each step of the nominal environment. The magnitude of the perturbation $\Delta_h^\lambda(\epsilon,D)$ at each stage $h$ depends on the divergence metric $D$, the regularized $\lambda$ and the parameter $\epsilon$.

The construction of the information-theoretic lower bound relies on a novel family of hard instances. We illustrate one such instance in Figure 3. Both the nominal and target environments satisfy Assumption 3.1. The environment consists of

two states, $s_1$ and $s_2$. In the nominal environment Figure 3(a), $s_1$ represents the good state with a positive reward. For any transition originating from $s_1$, there is a $1 - \epsilon$ probability of transitioning to itself and an $\epsilon$ probability of transitioning to $s_2$ regardless of the action taken, where $\epsilon$ is a parameter to be determined. The state $s_2$ is a fail state with zero reward and can only transition to itself. The worst-case target environment Figure 3(b) is obtained by perturbing the transition probabilities in the nominal environment. The perturbation magnitude $\Delta_h^\lambda(\epsilon, D)$ depends on the stage $h$, regularizer $\lambda$, divergence metric $D$, and parameter $\epsilon$.

The family of $d$-rectangular linear RRMDPs are parameterized by a Boolean vector $\boldsymbol{\xi} = \{\boldsymbol{\xi}_h\}_{h \in [H]}$, where $\boldsymbol{\xi}_h \in \{-1, 1\}^d$. For a given $\boldsymbol{\xi}$ and regularizer $\lambda$, the corresponding $d$-rectangular linear RRMDP $M_{\boldsymbol{\xi}}^\rho$ is constructed as follows. The state space $\mathcal{S} = \{x_1, x_2\}$ and the action space $\mathcal{A} = \{0, 1\}^d$. The initial state distribution $\mu_0$ is defined as

$$\mu_0(s_1) = \frac{d+1}{d+2} \quad \text{and} \quad \mu_0(x_2) = \frac{1}{d+2}.$$

The feature mapping $\phi : \mathcal{S} \times \mathcal{A} \to \mathbb{R}^{d+2}$ is defined as

$$\phi(s_1, a)^\top = \Big(\frac{a_1}{d}, \frac{a_2}{d}, \cdots, \frac{a_d}{d}, 1 - \sum_{i=1}^d \frac{a_i}{d}, 0\Big),$$

$$\phi(s_2, a)^\top = \big(0, 0, \cdots, 0, 0, 1\big),$$

which satisfies $\phi_i(s, a) \geq 0$ and $\sum_{i=1}^d \phi_i(s, a) = 1$. The nominal distributions $\{\boldsymbol{\mu}_h^0\}_{h \in [H]}$ are defined as

$$\boldsymbol{\mu}_h^0 = \Big( \underbrace{(1-\epsilon)\delta_{s_1} + \epsilon\delta_{s_2}, (1-\epsilon)\delta_{s_1} + \epsilon\delta_{s_2}, \cdots, (1-\epsilon)\delta_{s_1} + \epsilon\delta_{s_2}}_{d+1}, \delta_{s_2} \Big)^\top, \forall h \in [H],$$

where $\epsilon$ is an error term injected into the nominal model, which is to be determined later. Thus, the transition is homogeneous and does not depend on action but only on state. The reward parameters $\{\boldsymbol{\theta}_h\}_{h \in [H]}$ are defined as

$$\boldsymbol{\theta}_h^\top = \delta \cdot \Big(\frac{\xi_{h1} + 1}{2}, \frac{\xi_{h2} + 1}{2}, \cdots \frac{\xi_{hd} + 1}{2}, \frac{1}{2}, 0\Big), \forall h \in [H],$$

where $\delta$ is a parameter to control the differences among instances, which is to be determined later. The reward $r_h$ is generated from the normal distribution $r_h \sim \mathcal{N}(r_h(s_h, a_h), 1)$, where $r_h(s, a) = \phi(s, a)^\top \boldsymbol{\theta}_h$. Note that

$$r_h(s_1, a) = \phi(s_1, a)^\top \boldsymbol{\theta}_h = \frac{\delta}{2d}\big(\langle \boldsymbol{\xi}_h, a \rangle + d\big) \geq 0 \quad \text{and} \quad r_h(s_2, a) = \phi(s_2, a)^\top \boldsymbol{\theta}_h = 0, \ \forall a \in \mathcal{A},$$

Thus, the worst case transition kernel should have the highest possible transition probability to $s_2$, and the optimal robust policy should lead to a transition probability to $s_2$ as small as possible. Therefore the optimal action at step $h$ is

$$a_h^\star = \Big(\frac{1 + \xi_{h1}}{2}, \frac{1 + \xi_{h2}}{2} \cdots, \frac{1 + \xi_{hd}}{2}\Big).$$

We illustrate the designed $d$-rectangular linear RRMDP $M_{\boldsymbol{\xi}}^\lambda$ in Figure 3(a) and Figure 3(b).

Finally, the offline dataset is collected by the following procedure: the behavior policy $\pi^b = \{\pi_h^b\}_{h \in [H]}$ is defined as

$$\pi_h^b \sim \text{Unif}\big(\{\boldsymbol{e}_1, \cdots, \boldsymbol{e}_d, \boldsymbol{0}\}\big), \forall h \in [H],$$

where $\{\boldsymbol{e}_i\}_{i \in [d]}$ are the canonical basis vectors in $\mathbb{R}^d$. The initial state is generated according to $\mu_0$ and then the behavior policy interact with the nominal environment $K$ episodes to collect the offline dataset $\mathcal{D}$.

**Remark D.1.** We would like to highlight the difference between our hard instances and the hard instances developed in Liu & Xu (2024b). We find out that instances developed in Liu & Xu (2024b) only allow perturbations measured in TV-divergence. The reason is that in their nominal environment, both $s_1$ and $s_2$ are absorbing states, and thus $P_h^0(\cdot|s, a)$ only has support on $s$, which could be either $s_1$ or $s_2$. In this case, any perturbation to $P_h^0(\cdot|s, a)$ would cause a violation of the absolute continuous condition in the definition of the KL-divergence and the $\chi^2$-divergence[3]. In comparison, we inject a

---

[3]It has been shown in Proposition 2.5 of Lu et al. (2024) that the TV divergence can be extended to allow for two distributions with different support.

small error $\epsilon$ in the nominal kernel such that $P_h^0(\cdot|s_1, a)$ has full support $\{s_1, s_2\}$ when the transition starts from $s_1$. Hence, we can make perturbations on $P_h^0(\cdot|s_1, a)$ safely without violating the absolutely continuous condition. Additionally, Liu & Xu (2024b) only construct perturbation in the first stage, while we admit perturbation in every stage $h$ in order to make our instance more general.

### D.2. Proof of Theorem 5.5

With this family of hard instances, we are ready to prove the information-theoretic lower bound. For any $\boldsymbol{\xi} \in \{-1, 1\}^{dH}$, let $\mathbb{Q}_{\boldsymbol{\xi}}$ denote the distribution of dataset $\mathcal{D}$ collected from the instance $M_{\boldsymbol{\xi}}$. Denote the family of parameters as $\Omega = \{-1, 1\}^{dH}$ and the family of hard instances as $\mathcal{M} = \{M_{\boldsymbol{\xi}} : \boldsymbol{\xi} \in \Omega\}$. Before the proof, we introduce the following lemma bounding the robust value function.

**Lemma D.2.** Under the constructed hard instances in Appendix D.1, let $\delta = d^{3/2}/\sqrt{2K}$ and $K > d^3 H^2/(2\lambda^2)$. For any $h \in [H]$, we have

$$0 \le \frac{\delta}{2d} \sum_{j=h}^{H}(1-\epsilon)^{j-h}\Big(d + \Big(\sum_{i=1}^{d}\xi_{ji}\mathbb{E}^\pi a_{ji}\Big)\Big) - V_h^{\pi,\lambda}(s_1) \le f_h^\lambda(\epsilon), \tag{D.1}$$

where $f_h^\lambda(\epsilon)$ is a error term, which is defined as:

$$f_h^\lambda(\epsilon) = \begin{cases} 0, & \text{if D is TV;} \\ (H-h)\lambda\epsilon(e-1), & \text{if D is KL;} \\ (H-h)\lambda\epsilon(1-\epsilon)/4, & \text{if D is } \chi^2. \end{cases}$$

Furthermore, if we set the $\epsilon$ as

$$\epsilon = \begin{cases} 1 - 2^{-1/H}, & \text{if D is TV;} \\ \min\{1 - 2^{-1/H}, d^{3/2}/(64\lambda\sqrt{2K})\}, & \text{if D is KL;} \\ \min\{1 - 2^{-1/H}, d^{3/2}/(8\lambda\sqrt{2K})\}, & \text{if D is } \chi^2, \end{cases} \tag{D.2}$$

then we have $f_h^\lambda(\epsilon) \le d^{3/2}H/32\sqrt{2K}$.

*Proof of Theorem 5.5.* Invoking Lemma D.2, we have

$$V_1^{\pi^\star,\lambda}(s_1) - V_1^{\pi,\lambda}(s_1) \ge \frac{\delta}{2d}\sum_{j=1}^{H}\sum_{i=1}^{d}(1-\epsilon)^{j-1}\Big(\frac{1+\xi_{ji}}{2} - \xi_{ji}\mathbb{E}^\pi a_{ji}\Big) - f_1^\lambda(\epsilon)$$

$$= \frac{\delta}{4d}\sum_{j=1}^{H}\sum_{i=1}^{d}(1-\epsilon)^{j-1}(1-\xi_{ji}\mathbb{E}^\pi(2a_{ji}-1)) - f_1^\lambda(\epsilon)$$

$$= \frac{\delta}{4d}\sum_{j=1}^{H}\sum_{i=1}^{d}(1-\epsilon)^{j-1}(\xi_{ji} - \mathbb{E}^\pi(2a_{ji}-1))\xi_{ji} - f_1^\lambda(\epsilon)$$

$$= \frac{\delta}{4d}\sum_{j=1}^{H}\sum_{i=1}^{d}(1-\epsilon)^{j-1}|\xi_{ji} - \mathbb{E}^\pi(2a_{ji}-1)| - f_1^\lambda(\epsilon) \tag{D.3}$$

$$\ge \frac{\delta}{4d}(1-\epsilon)^{H-1}\sum_{j=1}^{H}\sum_{i=1}^{d}|\xi_{ji} - \mathbb{E}^\pi(2a_{ji}-1)| - f_1^\lambda(\epsilon), \tag{D.4}$$

where (D.3) follows from the fact that $\xi_{ji} \in \{-1, 1\}$. To continue,

$$\frac{\delta}{4d}\sum_{j=1}^{H}\sum_{i=1}^{d}|\xi_{ji} - \mathbb{E}^\pi(2a_{ji}-1)| \ge \frac{\delta}{4d}\sum_{j=1}^{H}\sum_{i=1}^{d}|\xi_{ji} - \mathbb{E}^\pi(2a_{ji}-1)|\mathbf{1}\{\xi_{hi} \ne \text{sign}(\mathbb{E}(2a_{ji}-1))\}|$$

$$\geq \frac{\delta}{4d} \sum_{j=1}^{H} \sum_{i=1}^{d} \mathbf{1}\{\xi_{hi} \neq \text{sign}(\mathbb{E}(2a_{ji} - 1))\}|$$

$$= \frac{\delta}{4d} D_H(\xi, \xi^\pi), \tag{D.5}$$

where $D_H(\cdot, \cdot)$ is the Hamming distance. Then applying the Assouad's method (Lemma 2.12 in Tsybakov (2009)), we have

$$\inf_{\pi} \sup_{\xi \in \Omega} \mathbb{E}_\xi[D_H(\xi, \xi^\pi)] \geq \frac{dH}{2} \min_{D_H(\xi, \xi^\pi)=1} \inf_{\phi} [\mathbb{Q}_\xi(\psi(\mathcal{D}) \neq \xi) + \mathbb{Q}_{\xi^\pi}(\psi(\mathcal{D}) \neq \xi^\pi)]$$

$$\geq \frac{dH}{2} \left( 1 - \left( \frac{1}{2} \max_{D_H(\xi, \xi^\pi)=1} D_{\text{KL}}(\mathbb{Q}_\xi \| \mathbb{Q}_{\xi^\pi}) \right)^{1/2} \right), \tag{D.6}$$

where $D_{\text{KL}}$ represents the KL divergence. Next we bound $D_{\text{KL}}(\mathbb{Q}_\xi \| \mathbb{Q}_{\xi^\pi})$, according to the definition of $\mathbb{Q}_\xi(\mathcal{D})$, we have

$$\mathbb{Q}_\xi(\mathcal{D}) = \prod_{k=1}^{K} \prod_{\tau=1}^{H} \pi_h^b(a_h^\tau | s_h^\tau) P_h^0(s_{h+1}^\tau | s_h^\tau, a_h^\tau) R_h(r_h^\tau | s_h^\tau, a_h^\tau),$$

where $R_h(r_h^\tau | s_h^\tau, a_h^\tau)$ refers to the density function of $\mathcal{N}(r_h(s_h^\tau, a_h^\tau), 1)$ at $r_h^\tau$. By the fact that the difference between the two distributions $\mathbb{Q}_\xi(\mathcal{D})$ and $\mathbb{Q}_{\xi^\pi}(\mathcal{D})$ lie only in the reward distribution corresponding to the index where $\xi$ and $\xi^\pi$ differ, we have

$$D_{KL}(\mathbb{Q}_\xi(\mathcal{D}) \| \mathbb{Q}_{\xi^\pi}(\mathcal{D})) = \sum_{\tau=1}^{\frac{K}{d+2}} D_{KL}\left( \mathcal{N}\left( \frac{d+1}{2d}\delta, 1 \right) \| \mathcal{N}\left( \frac{d-1}{2d}\delta, 1 \right) \right) = \frac{K}{d+2} \frac{\delta^2}{d^2} \leq \frac{1}{2}, \tag{D.7}$$

where the last inequality follows from the definition of $\delta$. By the fact that $\delta = d^{3/2}/\sqrt{2K}$, we have

$$\inf_{\hat{\pi}} \sup_{M \in \mathcal{M}} \text{subopt}(M, \hat{\pi}, s, \lambda) \geq \frac{\delta H(1-\epsilon)^{H-1}}{4} \left( 1 - \left( \frac{1}{2} \max_{D_H(\xi, \xi^\pi)=1} D_{KL}(\mathbb{Q}_\xi \| \mathbb{Q}_{\xi^\pi}) \right)^{1/2} \right) - f_h^\lambda(\epsilon) \tag{D.8}$$

$$\geq \frac{\delta H(1-\epsilon)^{H-1}}{8} - f_h^\lambda(\epsilon) \tag{D.9}$$

$$= \frac{d^{3/2} H(1-\epsilon)^{H-1}}{8\sqrt{2K}} - f_h^\lambda(\epsilon)$$

$$\geq \frac{d^{3/2} H}{16\sqrt{2K}} - \frac{d^{3/2} H}{32\sqrt{2K}} \tag{D.10}$$

$$\geq \frac{1}{128\sqrt{2}} \sum_{h=1}^{H} \mathbb{E}^{\pi^\star, P} \Big[ \sum_{i=1}^{d} \|\phi_i(s, a)\mathbf{1}_i\|_{\Lambda_h^{-1}} |s_1 = s \Big], \tag{D.11}$$

where (D.8) holds by applying the inequality (D.4), (D.5) and (D.6) in order, (D.9) holds by (D.7), (D.10) holds by the definition of $\epsilon$ in (D.2), and (D.11) holds by Lemma F.4. Hence, it is sufficient for taking $c = 1/128\sqrt{2}$. This concludes the proof.

$\square$

# E. Proof of Technical Lemmas

### E.1. Proof of Lemma C.1

*Proof.* We first decompose the $\text{SubOpt}(\pi, s, \lambda)$ as follows:

$$\text{SubOpt}(\hat{\pi}, s, \lambda) = V_1^{\star, \lambda}(s) - V_1^{\hat{\pi}, \lambda}(s)$$

$$= \underbrace{V_1^{\star, \lambda}(s) - \hat{V}_1^\lambda(s)}_{(i)} + \underbrace{\hat{V}_1^\lambda(s) - V_1^{\hat{\pi}, \lambda}(s)}_{(ii)},$$

where $\hat{V}_1^\lambda(s)$ is computed in the algorithm. We next bound the term (i) and (ii) respectively. For term (i), the error comes from the estimated error of the value function and the Q-function, therefore by (3.2) and the definition of the $\hat{Q}_h^\lambda(s,a)$ in meta-algorithm, for any $h \in [H]$, we can decompose the error as:

$$
\begin{aligned}
V_h^{\pi^\star,\lambda}(s) - \hat{V}_h^\lambda(s) &= Q_h^{\pi^\star,\lambda}(s,\pi_h^\star(s)) - \hat{Q}_h^\lambda(s,\hat{\pi}_h(s)) \\
&\leq Q_h^{\pi^\star,\lambda}(s,\pi_h^\star(s)) - \hat{Q}_h^\lambda(s,\pi^\star(s)) \tag{E.1} \\
&= \mathcal{T}_h^\lambda V_{h+1}^{\pi^\star,\lambda}(s,\pi_h^\star(s)) - \mathcal{T}_h^\lambda \hat{V}_{h+1}^\lambda(s,\pi_h^\star(s)) + \mathcal{T}_h^\lambda \hat{V}_{h+1}^\lambda(s,\pi_h^\star(s)) - \hat{Q}_h^\lambda(s,\pi^\star(s)) \\
&= \mathcal{T}_h^\lambda V_{h+1}^{\pi^\star,\lambda}(s,\pi_h^\star(s)) - \mathcal{T}_h^\lambda \hat{V}_{h+1}^\lambda(s,\pi_h^\star(s)) + \delta_h^\lambda(s,\pi_h^\star(s)), \tag{E.2}
\end{aligned}
$$

where (E.1) comes from the fact that $\hat{\pi}_h$ is the greedy policy with respect to $\hat{Q}_h^\lambda(s,a)$, the regularized robust Bellman update error $\delta_h^\lambda$ is defined as:

$$
\delta_h^\lambda(s,a) := \mathcal{T}_h^\lambda \hat{V}_{h+1}^\lambda(s,a) - \hat{Q}_h^\lambda(s,a), \forall (s,a) \in \mathcal{S} \times \mathcal{A}, \tag{E.3}
$$

which aims to eliminate the clip operator in the definition of $\hat{Q}_h^\lambda(s,a)$. Denote the worst transition kernel w.r.t the regularized Bellman operator as $\hat{P} = \{\hat{P}_h\}_{h \in [H]}$, where $\hat{P}_h$ is defined as:

$$
\begin{aligned}
\hat{P}_h(\cdot|s,a) &= \underset{\boldsymbol{\mu}_h \in \Delta(\mathcal{S})^d, P_h = \langle \boldsymbol{\phi}, \boldsymbol{\mu}_h \rangle}{\operatorname{argmin}} \left[ \mathbb{E}_{s' \sim P_h(\cdot|s,a)}[\hat{V}_{h+1}^\lambda(s')] + \lambda \langle \boldsymbol{\phi}(s,a), \boldsymbol{D}(\boldsymbol{\mu}_h\|\boldsymbol{\mu}_h^0) \rangle \right] \\
&= \sum_{i=1}^d \phi_i(s,a) \underset{\mu_{h,i} \in \Delta(\mathcal{S})}{\operatorname{argmin}} \left[ \mathbb{E}_{s' \sim \mu_{h,i}}[\hat{V}_{h+1}^\lambda(s')] + \lambda D(\mu_{h,i}\|\mu_{h,i}^0) \right] \\
&= \sum_{i=1}^d \phi_i(s,a) \hat{\mu}_{h,i}(\cdot),
\end{aligned}
$$

where the $\hat{\mu}_{h,i}$ is defined as $\hat{\mu}_{h,i} = \operatorname{argmin}_{\mu_{h,i} \in \Delta(\mathcal{S})} \left[ \mathbb{E}_{s' \sim \mu_{h,i}}[\hat{V}_{h+1}(s')] + \lambda D(\mu_{h,i}\|\mu_{h,i}^0) \right]$. Hence the difference between the regularized Bellman operator and the empirical regularized Bellman operator can be bounded as

$$
\begin{aligned}
&\mathcal{T}_h^\lambda V_{h+1}^{\pi^\star,\lambda}(s,\pi_h^\star(s)) - \mathcal{T}_h^\lambda \hat{V}_{h+1}^\lambda(s,\pi_h^\star(s)) \\
&= r_h(s,\pi_h^\star(s)) + \inf_{\boldsymbol{\mu}_h \in \Delta(\mathcal{S})^d, P_h = \langle \boldsymbol{\phi}, \boldsymbol{\mu}_h \rangle} \left[ \mathbb{E}_{s' \sim P_h(\cdot|s,\pi_h^\star(s))}[V_{h+1}^{\pi^\star,\lambda}(s')] + \lambda \langle \boldsymbol{\phi}(s,\pi_h^\star(s)), \boldsymbol{D}(\boldsymbol{\mu}_h\|\boldsymbol{\mu}_h^0) \rangle \right] \\
&\quad - r_h(s,\pi_h^\star(s)) - \inf_{\boldsymbol{\mu}_h \in \Delta(\mathcal{S})^d, P_h = \langle \boldsymbol{\phi}, \boldsymbol{\mu}_h \rangle} \left[ \mathbb{E}_{s' \sim P_h(\cdot|s,\pi_h^\star(s))}[\hat{V}_{h+1}^\lambda(s')] + \lambda \langle \boldsymbol{\phi}(s,\pi_h^\star(s)), \boldsymbol{D}(\boldsymbol{\mu}_h\|\boldsymbol{\mu}_h^0) \rangle \right] \\
&\leq \mathbb{E}_{s' \sim \hat{P}_h(\cdot|s,\pi_h^\star(s))}[\hat{V}_{h+1}^\lambda(s')] - \mathbb{E}_{s' \sim \hat{P}_h(\cdot|s,\pi_h^\star(s))}[V_{h+1}^{\star,\lambda}(s')] \\
&= \mathbb{E}_{s' \sim \hat{P}_h(\cdot|s,\pi_h^\star(s))}[\hat{V}_{h+1}^\lambda(s') - V_{h+1}^{\star,\lambda}(s')]. \tag{E.4}
\end{aligned}
$$

Combining inequality (E.2) and (E.4), we have for any $h \in [H]$

$$
V_h^{\pi^\star,\lambda}(s) - \hat{V}_h^\lambda(s) \leq \mathbb{E}_{s' \sim \hat{P}_h(\cdot|s,\pi_h^\star(s))}[\hat{V}_{h+1}^\lambda(s') - V_{h+1}^{\star,\lambda}(s')] + \delta_h^\lambda(s,\pi_h^\star(s)). \tag{E.5}
$$

Recursively applying (E.5), we have

$$
\text{(i)} = V_1^{\pi^\star,\lambda}(s) - \hat{V}_1^\lambda(s) \leq \sum_{h=1}^H \mathbb{E}^{\pi^\star,\hat{P}}[\delta_h^\lambda(s_h,a_h)\|s_1 = s].
$$

Next we bound term (ii), similar to term (i), by (C.1), the error can be decomposed to

$$
\begin{aligned}
\hat{V}_h^\lambda(s) - V_h^{\hat{\pi},\lambda}(s) &= \hat{Q}_h^\lambda(s,\hat{\pi}_h(s)) - Q_h^{\hat{\pi},\lambda}(s,\hat{\pi}_h(s)) \\
&= \mathcal{T}_h^\lambda \hat{V}_{h+1}^\lambda(s,\hat{\pi}_h(s)) - \delta_h^\lambda(s,\hat{\pi}_h(s)) - \mathcal{T}_h^\lambda V_{h+1}^{\hat{\pi},\lambda}(s,\hat{\pi}_h(s)). \tag{E.6}
\end{aligned}
$$

Denote $P^{\hat{\pi}} = \{P_h^{\hat{\pi}}\}_{h \in [H]}$ where $P_h^{\hat{\pi}}$ is defined as: $\forall (s,a) \in \mathcal{S} \times \mathcal{A}$,

$$
P_h^{\hat{\pi}}(\cdot|s,a) = \underset{\boldsymbol{\mu}_h \in \Delta(\mathcal{S})^d, P_h = \langle \boldsymbol{\phi}, \boldsymbol{\mu}_h \rangle}{\operatorname{argmin}} \left[ \mathbb{E}_{s' \sim P_h(\cdot|s,a)}[\hat{V}_{h+1}^{\hat{\pi}}(s')] + \lambda \langle \boldsymbol{\phi}(s,a), \boldsymbol{D}(\boldsymbol{\mu}_h\|\boldsymbol{\mu}_h^0) \rangle \right].
$$

Hence similar to the bound in (E.4), the difference between the regularized Bellman operator and the empirical regularized Bellman operator can be bounded as

$$\mathcal{T}_h^\lambda \hat{V}_{h+1}^\lambda(s, \hat{\pi}_h(s)) - \mathcal{T}_h^\lambda V_{h+1}^{\hat{\pi},\lambda}(s, \hat{\pi}_h(s)) \le \mathbb{E}_{s' \sim P_h^{\hat{\pi}}(\cdot|s, \hat{\pi}_h(s))}[\hat{V}_{h+1}^\lambda(s') - V_{h+1}^{\hat{\pi},\lambda}(s')]. \tag{E.7}$$

Combining inequality (E.6), (E.7), we have for any $h \in [H]$

$$\hat{V}_h^\lambda(s) - V_h^{\hat{\pi},\lambda}(s) \le \mathbb{E}_{s' \sim P_h^{\hat{\pi}}(\cdot|s, \hat{\pi}_h(s))}[\hat{V}_{h+1}^\lambda(s') - V_{h+1}^{\hat{\pi},\lambda}(s')] - \delta_h^\lambda(s, \hat{\pi}_h(s)). \tag{E.8}$$

Recursively applying (E.8), we have the "pessimisim" of the estimated value function that $\forall h \in [H]$

$$\hat{V}_1^\lambda(s) - V_1^{\hat{\pi},\lambda}(s) \le \sum_{h=1}^H \mathbb{E}^{\hat{\pi}, P^{\hat{\pi}}}\big[ - \delta_h^\lambda(s_h, a_h)\|s_1 = s\big].$$

Therefore combining the two bounds above, we have

$$\text{SubOpt}(\hat{\pi}, s, \lambda) = (\text{i}) + (\text{ii}) \le \sum_{h=1}^H \mathbb{E}^{\pi^\star, \hat{P}}\big[\delta_h^\lambda(s_h, a_h)|s_1 = s\big] + \sum_{h=1}^H \mathbb{E}^{\hat{\pi}, P^{\hat{\pi}}}\big[ - \delta_h^\lambda(s_h, a_h)\|s_1 = s\big]. \tag{E.9}$$

Hence, it requires to estimate the range of the regularized Bellman update error $\delta_h^\lambda(s, a)$. Recall the definition in (E.3), we claim that

$$0 \le \delta_h^\lambda(s, a) \le 2\Gamma_h(s, a) \tag{E.10}$$

holds for $\forall(s, a, h) \in \mathcal{S} \times \mathcal{A} \times [H]$. For the LHS of (E.10), first we notice that if $\langle \phi(s, a), \hat{w}_h^\lambda \rangle - \Gamma_h(s, a) \le 0$, the inequality holds trivially as $\hat{Q}_h^\lambda(s, a) = 0$. Next we consider the case where $\langle \phi(s, a), \hat{w}_h^\lambda \rangle - \Gamma_h(s, a) \ge 0$. By the definition of $\hat{Q}_h^\lambda(s, a)$ and the assumption in the lemma, we have

$$\begin{aligned}
\delta_h^\lambda(s, a) &= \mathcal{T}_h^\lambda \hat{V}_{h+1}^\lambda(s, a) - \hat{Q}_h^\lambda(s, a) \\
&= \mathcal{T}_h^\lambda \hat{V}_{h+1}^\lambda(s, a) - \min\big\{\langle \phi(s, a), \hat{w}_h^\lambda \rangle - \Gamma_h(s, a), H - h + 1\big\} \\
&\ge \mathcal{T}_h^\lambda \hat{V}_{h+1}^\lambda(s, a) - \langle \phi(s, a), \hat{w}_h^\lambda \rangle + \Gamma_h(s, a) \\
&\ge 0.
\end{aligned}$$

On the other hand, by the assumption in the lemma, we have

$$\langle \phi(s, a), \hat{w}_h^\lambda \rangle - \Gamma_h(s, a) \le \mathcal{T}_h^\lambda \hat{V}_{h+1}^\lambda(s, a) \le H - h + 1.$$

Hence, we can upper bound $\delta_h^\lambda(s, a)$ as

$$\begin{aligned}
\delta_h^\lambda(s, a) &= \mathcal{T}_h^\lambda \hat{V}_{h+1}^\lambda(s, a) - \hat{Q}_h^\lambda(s, a) \\
&= \mathcal{T}_h^\lambda \hat{V}_{h+1}^\lambda(s, a) - \max\big\{\langle \phi(s, a), \hat{w}_h^\lambda \rangle - \Gamma_h(s, a), 0\big\} \\
&\le \mathcal{T}_h^\lambda \hat{V}_{h+1}^\lambda(s, a) - \langle \phi(s, a), \hat{w}_h^\lambda \rangle + \Gamma_h(s, a) \\
&\le 2\Gamma_h(s, a).
\end{aligned}$$

This concludes the claim. Now it remains to bound the empirical transition kernel $\hat{P}$. Noticing the fact that $\forall h \in [H], (s, a) \in \mathcal{S} \times \mathcal{A}$,

$$\begin{aligned}
\lambda D(\hat{\mu}_{h,i}\|\mu_{h,i}^0) &\le \mathbb{E}_{s' \sim \hat{\mu}_{h,i}}[\hat{V}_{h+1}^\lambda(s')] + \lambda D(\hat{\mu}_{h,i}\|\mu_{h,i}^0) \\
&= \inf_{\mu_{h,i} \in \Delta(\mathcal{S})} \big[\mathbb{E}_{s' \sim \mu_{h,i}}[\hat{V}_{h+1}(s')] + \lambda D(\mu_{h,i}\|\mu_{h,i}^0)\big] \\
&\le \inf_{\mu_{h,i} \in \Delta(\mathcal{S})} \big[\mathbb{E}_{s' \sim \mu_{h,i}}[V_{h+1}^{\star,\lambda}(s')] + \lambda D(\mu_{h,i}\|\mu_{h,i}^0)\big] \\
&\le \mathbb{E}_{s' \sim \mu_{h,i}^0}[V_{h+1}^{\star,\lambda}(s')]
\end{aligned} \tag{E.11}$$

$$\leq \max_{s \in \mathcal{S}} V_{h+1}^{\star,\lambda}(s),$$

where (E.11) comes from the pessimism of value function, i.e $\hat{V}_{h+1}^{\lambda}(s) \leq V_h^{\star,\lambda}(s), \forall h \in [H]$. Hence, the empirical transition kernel $\hat{P}_h(\cdot|s,a)$ is contained in the set $\mathcal{U}^{\lambda}(P^0)$ defined in (5.1). Hence, by (E.9) and (E.10), we have

$$\text{SubOpt}(\hat{\pi}, s, \lambda) \leq \sum_{h=1}^{H} \mathbb{E}^{\pi^{\star}, \hat{P}} \big[ \delta_h^{\lambda}(s_h, a_h)|s_1 = s \big] + \sum_{h=1}^{H} \mathbb{E}^{\hat{\pi}, P^{\hat{\pi}}} \big[ - \delta_h^{\lambda}(s_h, a_h)\|s_1 = s \big]$$

$$\leq 2 \sum_{h=1}^{H} \mathbb{E}^{\pi^{\star}, \hat{P}} \big[ \Gamma_h(s_h, a_h)|s_1 = s \big]$$

$$\leq 2 \sup_{P \in \mathcal{U}^{\lambda}(P^0)} \sum_{h=1}^{H} \mathbb{E}^{\pi^{\star}, P} \big[ \Gamma_h(s_h, a_h)|s_1 = s \big].$$

This concludes the proof. $\qquad\square$

### E.2. Proof of Lemma C.2

*Proof.* For all $h \in [H]$, from the definition of $w_h$,

$$\|\boldsymbol{w}_h^{\lambda}\|_2 = \|\mathbb{E}_{s \sim \boldsymbol{\mu}_h^0}[\hat{V}_{h+1}^{\lambda}(s)]_{\alpha_{h+1}}\|_2 \leq H\sqrt{d},$$

where the inequality follows from the fact that $\hat{V}_{h+1}^{\lambda} \leq H$, for all $h \in [H]$. Meanwhile, by the definition of $\hat{w}_h^{\lambda}$ in Algorithm 3, and the triangle inequality,

$$\|\hat{w}_h^{\lambda}\|_2 = \left\| \boldsymbol{\Lambda}_h^{-1} \sum_{\tau=1}^{K} \boldsymbol{\phi}(s_h^{\tau}, a_h^{\tau})[\hat{V}_{h+1}^{\lambda}(s)]_{\alpha_{h+1}} \right\|_2$$

$$\leq H \sum_{\tau=1}^{K} \|\boldsymbol{\Lambda}_h^{-1} \boldsymbol{\phi}(s_h^{\tau}, a_h^{\tau})\|_2$$

$$= H \sum_{\tau=1}^{K} \sqrt{\boldsymbol{\phi}(s_h^{\tau}, a_h^{\tau})^{\top} \boldsymbol{\Lambda}_h^{-1/2} \boldsymbol{\Lambda}_h^{-1} \boldsymbol{\Lambda}_h^{-1/2} \boldsymbol{\phi}(s_h^{\tau}, a_h^{\tau})}$$

$$\leq \frac{H}{\sqrt{\gamma}} \sum_{\tau=1}^{K} \sqrt{\boldsymbol{\phi}(s_h^{\tau}, a_h^{\tau})^{\top} \boldsymbol{\Lambda}_h^{-1} \boldsymbol{\phi}(s_h^{\tau}, a_h^{\tau})} \qquad (\text{E.12})$$

$$\leq \frac{H\sqrt{K}}{\sqrt{\gamma}} \sqrt{\sum_{\tau=1}^{K} \boldsymbol{\phi}(s_h^{\tau}, a_h^{\tau})^{\top} \boldsymbol{\Lambda}_h^{-1} \boldsymbol{\phi}(s_h^{\tau}, a_h^{\tau})} \qquad (\text{E.13})$$

$$= \frac{H\sqrt{K}}{\sqrt{\gamma}} \sqrt{\text{Tr}(\boldsymbol{\Lambda}_h^{-1}(\boldsymbol{\Lambda}_h - \gamma \mathbf{I}))}$$

$$\leq \frac{H\sqrt{K}}{\sqrt{\gamma}} \sqrt{\text{Tr}(\mathbf{I})}$$

$$= H\sqrt{\frac{Kd}{\gamma}},$$

where (E.12) follows from the fact that $\|\boldsymbol{\Lambda}_h^{-1}\| \leq \gamma^{-1}$, (E.13) follows from the Cauchy-Schwartz inequality. Then we conclude the proof. $\qquad\square$

### E.3. Proof of Lemma C.3

*Proof.* By definition, we have

$$\|\boldsymbol{w}_h^{\lambda}\|_2 = \left\| \mathbb{E}_{s \sim \boldsymbol{\mu}_h^0} \left[ e^{-\frac{\hat{V}_{h+1}^{\lambda}(s)}{\lambda}} \right] \right\|_2 = \int_{\mathcal{S}} \left\| e^{-\frac{\hat{V}_{h+1}^{\lambda}(s)}{\lambda}} \boldsymbol{\mu}_h^0(s) \right\|_2 ds \leq \int_{\mathcal{S}} \|\boldsymbol{\mu}_h^0(s)\|_2 ds \leq \sqrt{d},$$

this concludes the proof of $\boldsymbol{w}_h^\lambda$. For $\hat{\boldsymbol{w}}_h^\lambda$,

$$
\begin{aligned}
\|\hat{\boldsymbol{w}}_h^\lambda\|_2 &= \left\| \boldsymbol{\Lambda}_h^{-1} \sum_{\tau=1}^K \phi(s_h^\tau, a_h^\tau) e^{-\frac{\hat{V}_{h+1}^\lambda(s_{h+1})}{\lambda}} \right\|_2 \\
&\leq \sum_{\tau=1}^K \|\boldsymbol{\Lambda}_h^{-1} \phi(s_h^\tau, a_h^\tau)\|_2 \\
&= \sum_{\tau=1}^K \sqrt{\phi(s_h^\tau, a_h^\tau)^\top \boldsymbol{\Lambda}_h^{-1/2} \boldsymbol{\Lambda}_h^{-1} \boldsymbol{\Lambda}_h^{-1/2} \phi(s_h^\tau, a_h^\tau)} \\
&\leq \frac{1}{\sqrt{\gamma}} \sum_{\tau=1}^K \sqrt{\phi(s_h^\tau, a_h^\tau)^\top \boldsymbol{\Lambda}_h^{-1} \phi(s_h^\tau, a_h^\tau)} && \text{(E.14)} \\
&\leq \frac{\sqrt{K}}{\sqrt{\gamma}} \sqrt{\sum_{\tau=1}^K \phi(s_h^\tau, a_h^\tau)^\top \boldsymbol{\Lambda}_h^{-1} \phi(s_h^\tau, a_h^\tau)} && \text{(E.15)} \\
&= \frac{\sqrt{K}}{\sqrt{\gamma}} \sqrt{\mathrm{Tr}(\boldsymbol{\Lambda}_h^{-1}(\boldsymbol{\Lambda}_h - \gamma \mathbf{I}))} \\
&\leq \frac{\sqrt{K}}{\sqrt{\gamma}} \sqrt{\mathrm{Tr}(\mathbf{I})} = \sqrt{\frac{Kd}{\gamma}},
\end{aligned}
$$

where (E.14) follows from the fact that $\|\boldsymbol{\Lambda}_h^{-1}\| \leq \gamma^{-1}$, (E.15) follows from the Cauchy-Schwartz inequality. Then we conclude the proof. $\qquad\square$

### E.4. Proof of Lemma C.4

*Proof.* Denote $\boldsymbol{A} = \beta^2 \boldsymbol{\Lambda}_h^{-1}$, then we have $\|\boldsymbol{\theta}\|_2 \leq L, \|\boldsymbol{A}\|_2 \leq B^2 \gamma^{-1}$. For any two functions $V_1, V_2 \in \mathcal{V}$ with parameters $(\boldsymbol{\theta}_1, \boldsymbol{A}_1), (\boldsymbol{\theta}_2, \boldsymbol{A}_2)$, since both $\{\cdot\}_{[0, H-h+1]}$ and $\max_a$ are contraction maps,

$$
\mathrm{dist}(V_1, V_2) \tag{E.16}
$$
$$
\leq \sup_{s,a} \left| \phi(s,a)^\top (\boldsymbol{\theta}_1 - \boldsymbol{\theta}_2) - \lambda \left( \log \left( 1 + \sum_{i=1}^d \|\phi_i(s,a)\mathbf{1}_i\|_{\boldsymbol{A}_1} \right) - \log \left( 1 + \sum_{i=1}^d \|\phi_i(s,a)\mathbf{1}_i\|_{\boldsymbol{A}_2} \right) \right) \right|
$$
$$
\leq \sup_{\phi \in \mathbb{R}^d, \|\phi\| \leq 1} \left| \phi^\top (\boldsymbol{\theta}_1 - \boldsymbol{\theta}_2) - \lambda \log \frac{1 + \sum_{i=1}^d \|\phi_i(s,a)\mathbf{1}_i\|_{\boldsymbol{A}_1}}{1 + \sum_{i=1}^d \|\phi_i(s,a)\mathbf{1}_i\|_{\boldsymbol{A}_2}} \right|
$$
$$
\leq \sup_{\phi \in \mathbb{R}^d: \|\phi\| \leq 1} |\phi^\top (\boldsymbol{\theta}_1 - \boldsymbol{\theta}_2)| + \lambda \sup_{\phi \in \mathbb{R}^d: \|\phi\| \leq 1} \left| \log \frac{1 + \sum_{i=1}^d \|\phi_i(s,a)\mathbf{1}_i\|_{\boldsymbol{A}_1}}{1 + \sum_{i=1}^d \|\phi_i(s,a)\mathbf{1}_i\|_{\boldsymbol{A}_2}} \right|, \tag{E.17}
$$

we notice the fact that: for any $x > 0, y > 0$,

$$
\left| \log \frac{1+x}{1+y} \right| = \left| \log \left( \frac{x-y}{1+y} + 1 \right) \right| \leq \log(|x-y| + 1) \leq |x-y|.
$$

Therefore, (E.17) can be bounded as:

$$
\begin{aligned}
\text{(E.17)} &\leq \sup_{\phi \in \mathbb{R}^d: \|\phi\| \leq 1} |\phi^\top (\boldsymbol{\theta}_1 - \boldsymbol{\theta}_2)| + \lambda \sup_{\phi \in \mathbb{R}^d: \|\phi\| \leq 1} \left| \sum_{i=1}^d \|\phi_i(s,a)\mathbf{1}_i\|_{\boldsymbol{A}_1} - \sum_{i=1}^d \|\phi_i(s,a)\mathbf{1}_i\|_{\boldsymbol{A}_2} \right| \\
&= \sup_{\phi \in \mathbb{R}^d: \|\phi\| \leq 1} |\phi^\top (\boldsymbol{\theta}_1 - \boldsymbol{\theta}_2)| + \lambda \sup_{\phi \in \mathbb{R}^d: \|\phi\| \leq 1} \left| \sum_{i=1}^d \sqrt{\phi_i \mathbf{1}_i^\top \boldsymbol{A}_1 \phi_i \mathbf{1}_i} - \sum_{i=1}^d \sqrt{\phi_i \mathbf{1}_i^\top \boldsymbol{A}_2 \phi_i \mathbf{1}_i} \right| \\
&\leq \|\boldsymbol{\theta}_1 - \boldsymbol{\theta}_2\|_2 + \lambda \sup_{\phi \in \mathbb{R}^d: \|\phi\| \leq 1} \sum_{i=1}^d \sqrt{\phi_i \mathbf{1}_i^\top (\boldsymbol{A}_1 - \boldsymbol{A}_2) \phi_i \mathbf{1}_i} \tag{E.18}
\end{aligned}
$$

$$\leq \|\boldsymbol{\theta}_1 - \boldsymbol{\theta}_2\|_2 + \lambda \sqrt{\|\boldsymbol{A}_1 - \boldsymbol{A}_2\|} \sup_{\boldsymbol{\phi} \in \mathbb{R}^d : \|\boldsymbol{\phi}\| \leq 1} \sum_{i=1}^d \|\phi_i \mathbf{1}_i\|$$

$$\leq \|\boldsymbol{\theta}_1 - \boldsymbol{\theta}_2\|_2 + \lambda \sqrt{\|\boldsymbol{A}_1 - \boldsymbol{A}_2\|_F}, \tag{E.19}$$

where the (E.18) follows from the triangular inequality and the fact $|\sqrt{x} - \sqrt{y}| \leq \sqrt{|x - y|}$, and $\|\cdot\|_F$ denotes the Frobenius norm. We next define that $\mathcal{C}_{\boldsymbol{\theta}}$ is an $\epsilon/2$-cover of $\{\boldsymbol{\theta} \in \mathbb{R}^d | \|\boldsymbol{\theta}\|_2 \leq L\}$, and the $\mathcal{C}_{\boldsymbol{A}}$ is an $\epsilon^2/4\lambda^2$-cover of $\{\boldsymbol{A} \in \mathbb{R}^{d \times d} | \|\boldsymbol{A}\|_F \leq d^{1/2} B^2 \gamma^{-1}\}$. By Lemma F.5, we have that:

$$|\mathcal{C}_{\boldsymbol{\theta}}| \leq (1 + 4L/\epsilon)^d, |\mathcal{C}_{\boldsymbol{A}}| \leq (1 + 8\lambda^2 d^{1/2} B^2/\gamma\epsilon^2)^{d^2}.$$

By (E.19), for any $V_1 \in \mathcal{V}$, there exists $\boldsymbol{\theta}_2 \in \mathcal{C}_{\boldsymbol{\theta}}$ and $\boldsymbol{A}_2 \in \mathcal{C}_{\boldsymbol{A}}$ s.t $V_2$ parametrized by $(\boldsymbol{\theta}_2, \boldsymbol{A}_2)$ satisfying $\mathrm{dist}(V_1, V_2) \leq \epsilon$. Therefore, we have the following:

$$\log |\mathcal{N}(\epsilon)| \leq \log |\mathcal{C}_{\boldsymbol{\theta}}| + \log |\mathcal{C}_{\boldsymbol{A}}| \leq d \log(1 + 4L/\epsilon) + d^2 \log(1 + 8\lambda^2 d^{1/2} B^2/\gamma\epsilon^2).$$

Hence we conclude the proof. $\qquad \square$

### E.5. Proof of Lemma C.5

*Proof.* By definition, we have that

$$\|\hat{\boldsymbol{w}}_h^\lambda\|_2$$

$$= \left\| \left[ \max_{\alpha \in [(\hat{V}_{h+1}^\lambda)_{\min}, (\hat{V}_{h+1}^\lambda)_{\max}]} \left\{ \hat{\mathbb{E}}^{\mu_{h,i}^0} [\hat{V}_{h+1}^\lambda(s)]_\alpha + \frac{1}{4\lambda} (\hat{\mathbb{E}}^{\mu_{h,i}^0} [\hat{V}_{h+1}^\lambda(s)]_\alpha)^2 - \frac{1}{4\lambda} \hat{\mathbb{E}}^{\mu_{h,i}^0} [\hat{V}_{h+1}^\lambda(s)]_\alpha^2 \right\} \right]_{i \in [d]} \right\|_2$$

$$\leq \left\| \left[ H + \frac{H^2}{2\lambda} \right]_{i \in [d]} \right\|_2 \tag{E.20}$$

$$= \sqrt{d} \left( H + \frac{H^2}{2\lambda} \right),$$

where (E.20) follows by the fact that $\hat{\mathbb{E}}^{\mu_{h,i}^0} [\hat{V}_{h+1}^\lambda(s)]_\alpha \in [0, H], \hat{\mathbb{E}}^{\mu_{h,i}^0} [\hat{V}_{h+1}^\lambda(s)]_\alpha^2 \in [0, H^2]$. $\qquad \square$

### E.6. Proof of Lemma D.2

*Proof.* We first proof the LHS of the lemma by induction from last stage $H$. From the definition of $V_H^{\pi,\lambda}$ and $\boldsymbol{\theta}_h$, we can learn that

$$V_H^{\pi,\lambda}(s_1) = r_H(s_1, \pi_H(s_1)) = \phi(s_1, \pi(s_1))^\top \boldsymbol{\theta}_h = \frac{\delta}{2d} \left( d + \sum_{i=1}^d \xi_{Hi} \mathbb{E}^\pi a_{Hi} \right).$$

This is the base case. Now suppose the conclusion holds for stage $h+1$, that is to say,

$$V_{h+1}^{\pi,\lambda}(s_1) \leq \frac{\delta}{2d} \sum_{j=h+1}^H (1-\epsilon)^{j-h-1} \left( d + \left( \sum_{i=1}^d \xi_{ji} \mathbb{E}^\pi a_{ji} \right) \right).$$

Recall the regularized robust bellman equation in Proposition 3.2 and the regularized duality of the three divergences, we have

$$Q_h^{\pi,\lambda}(s_1, a) = r_h(s_1, a) + \inf_{\boldsymbol{\mu}_h \in \Delta(\mathcal{S})^{d+2}, P_h = \langle \boldsymbol{\phi}, \boldsymbol{\mu}_h \rangle} \left[ \mathbb{E}_{s' \sim P_h(\cdot|s,a)} [V_{h+1}^{\pi,\lambda}(s')] + \lambda \langle \phi(s,a), \boldsymbol{D}(\boldsymbol{\mu}_h \| \boldsymbol{\mu}_h^0) \rangle \right]$$

$$\leq r_h(s_1, a) + \mathbb{E}_{s' \sim P_h^0(\cdot|s_1,a)} [V_{h+1}^{\pi,\lambda}(s')] \tag{E.21}$$

$$= r_h(s_1, a) + (1 - \epsilon) V_{h+1}^{\pi,\lambda}(s_1). \tag{E.22}$$

Then with regularized robust bellman equation in Proposition 3.2 and the inductive hypothesis, we have

$$V_h^{\pi,\lambda}(s_1) = Q_h^{\pi,\lambda}(s_1, \pi(s_1))$$

$$\leq r_h(s_1, \pi_h(s_1)) + (1 - \epsilon)V_{h+1}^{\pi,\lambda}(s_1) \tag{E.23}$$

$$= \frac{\delta}{2d}\Big(d + \sum_{i=1}^{d}\xi_{hi}\mathbb{E}^{\pi}a_{hi}\Big) + \frac{\delta}{2d}\sum_{j=h+1}^{H}(1-\epsilon)^{j-h}\Big(d + \Big(\sum_{i=1}^{d}\xi_{ji}\mathbb{E}^{\pi}a_{ji}\Big)\Big)$$

$$= \frac{\delta}{2d}\sum_{j=h}^{H}(1-\epsilon)^{j-h}\Big(d + \Big(\sum_{i=1}^{d}\xi_{ji}\mathbb{E}^{\pi}a_{ji}\Big)\Big).$$

Hence, by the induction argument, we conclude the proof of the RHS. Furthermore, for any $h \in [H]$, we can upper bound $V_h^{\pi,\lambda}(s)$ as

$$V_h^{\pi,\lambda}(s) \leq \frac{\delta}{2d}\sum_{j=h}^{H}(1-\epsilon)^{j-h}\Big(d + \Big(\sum_{i=1}^{d}\xi_{ji}\mathbb{E}^{\pi}a_{ji}\Big)\Big) \leq \delta(H-h) \leq \lambda(H-h)/H \leq \lambda, \tag{E.24}$$

where the third inequality holds by the definition of $\delta$. For the left, we prove by discussing the KL, $\chi^2$ and TV cases respectively.

**Case I - TV.** The case for TV holds trivially as by Proposition 4.3, we have

$$Q_h^{\pi,\lambda}(s_1, a) = r_h(s_1, a) + \inf_{\boldsymbol{\mu}_h \in \Delta(\mathcal{S})^{d+2}, P_h = \langle \boldsymbol{\phi}, \boldsymbol{\mu}_h \rangle}\Big[\mathbb{E}_{s' \sim P_h(\cdot|s,a)}\big[V_{h+1}^{\pi,\lambda}(s')\big] + \lambda\langle\boldsymbol{\phi}(s,a), \boldsymbol{D}(\boldsymbol{\mu}_h\|\boldsymbol{\mu}_h^0)\rangle\Big]$$

$$= r_h(s_1, a) + \langle\boldsymbol{\phi}(s_1, a), \mathbb{E}_{s' \sim \boldsymbol{\mu}_h^0}[V_{h+1}^{\pi,\lambda}(s')]_{\min_{s'}(V_{h+1}^{\pi,\lambda}(s'))+\lambda}\rangle \tag{E.25}$$

$$= r_h(s_1, a) + \mathbb{E}_{s' \sim P_h^0(\cdot|s,a)}[V_{h+1}^{\pi,\lambda}(s')]_{\min_{s'}(V_{h+1}^{\pi,\lambda}(s'))+\lambda}$$

$$= r_h(s_1, a) + (1 - \epsilon)V_{h+1}^{\pi,\lambda}(s_1), \tag{E.26}$$

where (E.26) holds by (E.24). Hence, the inequality in (E.23) holds for equality. This concludes the proof for TV-divergence.

**Case II - KL.** We prove by induction. The case holds trivially in last stage $H$. Suppose

$$V_{h+1}^{\pi,\lambda}(s_1) \geq \frac{\delta}{2d}\sum_{j=h+1}^{H}(1-\epsilon)^{j-h-1}\Big(d + \Big(\sum_{i=1}^{d}\xi_{ji}\mathbb{E}^{\pi}a_{ji}\Big)\Big) - (H-h)\lambda\epsilon(e-1).$$

Recall the duality form of Proposition 4.6, the Q-function at stage $h$ can be upper bounded as:

$$Q_h^{\pi,\lambda}(s_1, a) = r_h(s_1, a) + \inf_{\boldsymbol{\mu}_h \in \Delta(\mathcal{S})^{d+2}, P_h = \langle \boldsymbol{\phi}, \boldsymbol{\mu}_h \rangle}\Big[\mathbb{E}_{s' \sim P_h(\cdot|s,a)}\big[V_{h+1}^{\pi,\lambda}(s')\big] + \lambda\langle\boldsymbol{\phi}(s,a), \boldsymbol{D}(\boldsymbol{\mu}_h\|\boldsymbol{\mu}_h^0)\rangle\Big]$$

$$= r_h(s_1, a) + \langle\boldsymbol{\phi}(s_1, a), -\lambda\log\mathbb{E}_{s' \sim \boldsymbol{\mu}_h^0}e^{-V_{h+1}^{\pi,\lambda}(s')/\lambda}\rangle$$

$$= r_h(s_1, a) - \lambda\log\big(\epsilon + (1-\epsilon)e^{-V_{h+1}^{\pi,\lambda}(s_1)/\lambda}\big)$$

$$= r_h(s_1, a) + V_{h+1}^{\pi,\lambda}(s_1) - \lambda\log\big(\epsilon e^{V_{h+1}^{\pi,\lambda}(s_1)/\lambda} + (1-\epsilon)\big)$$

$$\geq r_h(s_1, a) + V_{h+1}^{\pi,\lambda}(s_1) - \lambda\epsilon\big(e^{V_{h+1}^{\pi,\lambda}(s_1)/\lambda} - 1\big) \tag{E.27}$$

$$\geq r_h(s_1, a) + V_{h+1}^{\pi,\lambda}(s_1) - \lambda\epsilon(e-1), \tag{E.28}$$

where (E.27) follows by the fact that $\log(1 + x) \leq x, \forall x > 0$, (E.28) follows by (E.24). Therefore, by the inductive hypothesis, we have

$$V_h^{\pi,\lambda}(s_1) = Q_h^{\pi,\lambda}(s_1, \pi(s_1))$$

$$\geq r_h(s_1, \pi_h(s_1)) + (1-\epsilon)V_{h+1}^{\pi,\lambda}(s_1) - \lambda\epsilon(e-1)$$

$$= \frac{\delta}{2d}\sum_{j=h}^{H}(1-\epsilon)^{j-h}\Big(d + \Big(\sum_{i=1}^{d}\xi_{ji}\mathbb{E}^{\pi}a_{ji}\Big)\Big) - (H-h)\lambda\epsilon(e-1).$$

This finishes the KL setting.

**Case III - $\chi^2$.** Similar to the case in TV, KL, by the duality of $\chi^2$ in Proposition 4.8, we have

$$
\begin{aligned}
Q_h^{\pi,\lambda}(s_1, a) &= r_h(s_1, a) + \inf_{\boldsymbol{\mu}_h \in \Delta(\mathcal{S})^{d+2}, P_h = \langle \boldsymbol{\phi}, \boldsymbol{\mu}_h \rangle} \left[ \mathbb{E}_{s' \sim P_h(\cdot|s,a)} \left[ V_{h+1}^{\pi,\lambda}(s') \right] + \lambda \langle \boldsymbol{\phi}(s,a), \boldsymbol{D}(\boldsymbol{\mu}_h \| \boldsymbol{\mu}_h^0) \rangle \right] \\
&= r_h(s_1, a) + (1 - \epsilon) \sup_{\alpha \in [V_{\min}, V_{\max}]} \left\{ [V_{h+1}^{\pi,\lambda}(s_1)]_\alpha - \frac{\epsilon}{4\lambda} [V_{h+1}^{\pi,\lambda}(s_1)]_\alpha^2 \right\} \\
&\geq r_h(s_1, a) + (1 - \epsilon) \left[ V_{h+1}^{\pi,\lambda}(s_1) - \frac{\epsilon}{4\lambda} [V_{h+1}^{\pi,\lambda}(s_1)]^2 \right] \\
&\geq r_h(s_1, a) + (1 - \epsilon) V_{h+1}^{\pi,\lambda}(s_1) - \frac{\epsilon\lambda(1 - \epsilon)}{4},
\end{aligned}
\tag{E.29}
$$

where (E.29) follows by (E.24). Hence, similar to **Case II**, by induction, we have

$$
V_h^{\pi,\lambda}(s_1) \geq \frac{\delta}{2d} \sum_{j=h}^{H} (1 - \epsilon)^{j-h} \left( d + \left( \sum_{i=1}^{d} \xi_{ji} \mathbb{E}^\pi a_{ji} \right) \right) - (H - h) \frac{\epsilon\lambda(1 - \epsilon)}{4}.
$$

This finishes the $\chi^2$ setting, and we complete the proof. $\qquad\square$

# F. Auxiliary Lemmas

**Lemma F.1** (Lemma D.3 of Liu & Xu (2024a)). For any $h \in [H]$, let $\mathcal{V}_h$ denote a class of functions mapping from $\mathcal{S}$ to $\mathbb{R}$ with the following form:

$$
V_h(x; \boldsymbol{\theta}, \beta, \boldsymbol{\Lambda}_h) = \max_{a \in \mathcal{A}} \left\{ \boldsymbol{\phi}(s, a)^\top \boldsymbol{\theta} - \beta \sum_{i=1}^{d} \| \phi_i(\cdot, \cdot) \mathbf{1}_i \|_{\boldsymbol{\Lambda}_h^{-1}} \right\}_{[0, H-h+1]},
$$

the parameters $(\boldsymbol{\theta}, \beta, \boldsymbol{\Lambda}_h)$ satisfy $\|\boldsymbol{\theta}\|_2 \leq L, \beta \in [0, B], \gamma_{\min}(\boldsymbol{\Lambda}_h) \geq \gamma$. Let $\mathcal{N}_h(\epsilon)$ be the $\epsilon$-covering number of $\mathcal{V}$ with respect to the distance $\text{dist}(V_1, V_2) = \sup_x |V_1(x) - V_2(x)|$. Then

$$
\log \mathcal{N}_h(\epsilon) \leq d \log(1 + 4L/\epsilon) + d^2 \log(1 + 8d^{1/2}B^2/\gamma\epsilon^2).
$$

**Lemma F.2** (Corollary 4.2.11 of Vershynin (2018)). Denote the $\epsilon$-covering number of the closed interval $[a, b]$ for some real number $b > a$ with respect to the distance metric $d(\alpha_1, \alpha_2) = |\alpha_1 - \alpha_2|$ as $\mathcal{N}_\epsilon([a, b])$, then we have $\mathcal{N}_\epsilon([a, b]) \leq 3(b-a)/\epsilon$.

**Lemma F.3** (Lemma B.2 of Jin et al. (2021)). Let $f : \mathcal{S} \to [0, R-1]$ be any fixed function. For any $\delta \in (0, 1)$, we have

$$
P \left( \| \sum_{\tau=1}^{K} \boldsymbol{\phi}(s_h^\tau, a_h^\tau) \eta_h^\tau(f) \|_{\Lambda_h^{-1}}^2 \geq R^2 (2 \log(1/\delta) + d \log(1 + K/\gamma)) \right) \leq \delta,
$$

where $\eta_h^\tau(f) = \mathbb{E}_{s' \sim P_h^0(\cdot|s_h^\tau, a_h^\tau)}[f(s')] - f(s_{h+1}^\tau)$.

**Lemma F.4** (Lemma F.3 of Liu & Xu (2024b)). If $K \geq \mathcal{O}(d^6)$ and the feature map is define as Appendix D.1, then with probability at least $1 - \delta$, we have for any transition P,

$$
\sum_{h=1}^{H} \mathbb{E}^{\pi^\star, P} \left[ \sum_{i=1}^{d} \| \phi_i(s, a) \mathbf{1}_i \|_{\Lambda_h^{-1}} | \mathbf{s}_1 = s_1 \right] \leq \frac{4d^{3/2} H}{\sqrt{K}}.
$$

**Lemma F.5** (Lemma 5.2 of Vershynin (2010)). For any $\epsilon > 0$, the $\epsilon$-covering number of the Euclidean ball in $\mathbb{R}^d$ with radius $R > 0$ is upper bounded by $(1 + 2R/\epsilon)^d$.

