# OpenReview forum: "Robust Offline Reinforcement Learning with Linearly Structured $f$-Divergence Regularization"
_ICML.cc/2025/Conference — ICML 2025 poster_

### Official Review · Reviewer_8c9y · 2025-03-11

**Overall Recommendation:** 3

**Summary:**

This paper introduces a new framework, the $d$-rectangular linear robust regularized Markov decision process ($d$-RRMDP), for offline RL and develops a family of algorithms called robust regularized pessimistic value iteration (R2PVI) to learn robust policies. Upper bounds on the sub-optimality gap and the information theoretic lower bounds for $d$-RRMDPs are also provided. Numerical results are included to support the theoretical guarantees.

**Claims And Evidence:**

The baseline selection is reasonable, but incorporating more comprehensive evaluation criteria would further strengthen the analysis.

**Essential References Not Discussed:**

N/A

**Experimental Designs Or Analyses:**

Is that possible to validate the algorithms in a more complex setup, like mujoco?

**Methods And Evaluation Criteria:**

The baseline selection is reasonable but a more complex evaluation criteria will be benificial.

**Other Comments Or Suggestions:**

N/A

**Other Strengths And Weaknesses:**

Strengths:
This work presents a comprehensive theoretical analysis, establishing both upper and lower bounds for the newly proposed framework and algorithms.

Weaknesses:
The discussion on the practical impact and applicability of the method in real-world scenarios is limited.

**Questions For Authors:**

N/A

**Relation To Broader Scientific Literature:**

1. This paper introduces a new framework for enhancing robustness in offline RL, incorporating a broad class of divergences to improve generalization.
2. It establishes upper bounds on suboptimality and information-theoretic lower bounds for $d$-RRMDPs.

**Theoretical Claims:**

Most claims are well-supported, but the comparison of upper bounds with existing works lacks a thorough analysis of the robustness-efficiency trade-off. Since $\lambda$ controls the degree of regularization and robustness, evaluating only its scale relative to $\beta$ in the suboptimality gap is insufficient. A more comprehensive discussion on how choosing $\lambda$ affects robustness while maintaining efficiency compared to prior methods would strengthen the argument.

---

> ### Author Rebuttal · Authors · 2025-04-01
>
> We thank the reviewers for positive feedback on our work. We hope our response fully addresses your questions
>
> ---
> **Q**: A more comprehensive discussion on how choosing λ affects robustness while maintaining efficiency compared to prior methods would strengthen the argument. (more experiment)
>
>
> **A**: Figure 1b demonstrates how the parameter $\lambda$ controls the robustness of our policy, where robustness is defined as the policy's resistance to performance degradation under environmental disturbances. In general, larger values of $\lambda$ yield more robust optimal policies. However, we caution against direct comparisons of $\lambda$ values across different divergence measures (TV, KL, chi2), as varying $\lambda$ fundamentally alters the optimization objective and thus the resulting optimal policy. For a rigorous theoretical analysis of the relationship between λ and the robustness radius $\lambda$, we direct the reviewer to the discussion in Line 318.
>
> ---
> **Q**:  Clarify of computation efficiency
>
>
> **A**: While our work primarily focuses on the theoretical analysis of the RRMDP framework, we acknowledge that extending this framework to more general experimental settings represents an important complementary research direction.  Such extensions present distinct challenges that differ fundamentally from theoretical analysis, requiring careful consideration of practical implementation issues. Nevertheless, recent research has highlighted the promising potential of extending theoretical frameworks like RRMDP to more complex and realistic environments.
> - The learning of feature mapping. Recent advancements, such as [1], demonstrate the feasibility of generalizing linear MDPs to more flexible representations. This line of work offers valuable insights into how complex scenarios can be modeled through linear MDP settings.
> - The development of specialized benchmarks like [2] provides valuable infrastructure for systematically evaluating different approaches to uncertainty modeling, including both DRMDP and RRMDP frameworks.
> These developments highlight the feasibility of extending theoretical results to practical applications. However, given our current focus on establishing fundamental theoretical guarantees and the associated time constraints, we leave the design of comprehensive experiments in more general environments as an important direction for future research.
>
> ---
>
> We hope that we have addressed all of your questions/concerns. If you have further questions, we would be happy to answer them and if you don’t, would you kindly consider increasing your score?
>
> ---
>
> References
>
> [1] Zhang, Tianjun, Tongzheng Ren, Mengjiao Yang, Joseph Gonzalez, Dale Schuurmans, and Bo Dai. "Making linear mdps practical via contrastive representation learning." In International Conference on Machine Learning, pp. 26447-26466. PMLR, 2022.
>
> [2] Shangding Gu and Laixi Shi and Muning Wen and Ming Jin and Eric Mazumdar and Yuejie Chi and Adam Wierman and Costas Spanos. "Robust Gymnasium: A Unified Modular Benchmark for Robust Reinforcement Learning." arXiv preprint arXiv: 2502.19652.

---

> > ### Comment · Reviewer_8c9y · 2025-04-04
> >
> > I appreciate the authors for providing these details, which address most of my concerns. Thus, I will retain my original positive score.

---

> > > ### Author Response · Authors · 2025-04-07
> > >
> > > We thank the reviewer again for your positive feedback!

---

### Official Review · Reviewer_HBar · 2025-03-12

**Overall Recommendation:** 3

**Summary:**

The authors proposed a framework to solve the d-rectangular linear RRMDP. They extend the previous work under the distributional robust MDP framework by unifying three ways that define the potential MDPs consistent with the offline datasets and provide the theoretical analysis on the proposed method. Some brief experimental results are given to show the effectiveness of the proposed method.

**Claims And Evidence:**

The paper is clear-structured and the theoretical performance of the proposed method is good.

**Essential References Not Discussed:**

n/a

**Experimental Designs Or Analyses:**

Experiments are too simple. could the proposed methods be suitable for  more complicated benchmarks, such as MuJoCo?

**Methods And Evaluation Criteria:**

yes

**Other Comments Or Suggestions:**

n/a

**Other Strengths And Weaknesses:**

n/a

**Questions For Authors:**

see above.

**Relation To Broader Scientific Literature:**

n/a

**Theoretical Claims:**

1. In the introduction, the authors raised two main problems of previous method: Theoretical gaps and computation complexity. Although the authors theoretically prove the previous one, about the latter, only a simple experiment is provided, which is not sufficient to prove this point - computation complexity, is well addressed. Could the authors provide more discussion about this? For instance, compare with [Pessimistic q-learning for offline reinforcement learning: Towards optimal sample complexity].

2.The whole framework is based on the linear MDP, as assumed in 3.1, which may hinder the practical application of the proposed method.

---

> ### Author Rebuttal · Authors · 2025-04-01
>
> We thank the reviewers for positive feedback on our work. We hope our response fully addresses your questions
>
> ---
> **Q**: Difference with [Pessimistic q-learning for offline reinforcement learning: Towards optimal sample complexity]
>
>
> **A**: We claim the difference between our work and [Pessimistic q-learning for offline reinforcement learning: Towards optimal sample complexity].
> 1. Different objectives:
> - Shi et al. primarily address dynamics shift and sample complexity challenges in standard MDPs, focusing on learning a policy under the empirical transition dynamics.
> - Our work studies the Robust RL (RMDP) framework, which optimizes policies under the worst-case transition kernel within a structured uncertainty set. This leads to fundamentally different problem formulations and optimal policies.
> 2. Different Environments and Assumptions:
> - Shi et al. operates in tabular settings with finite state and action spaces.
> - Our work adopts a linear MDP structure, enabling scalable function approximation and extending the analysis to high-dimensional settings.
>
> ---
> **Q**:  Clarify of computation efficiency
>
>
> **A**: We emphasize that our computational efficiency is evaluated relative to the standard Distributionally Robust MDP (DRMDP) framework and its associated algorithms.
>
> Under both TV and KL divergence measures, our approach achieves superior computation efficiency by leveraging closed-form duality solutions, which are more tractable than those in the conventional DRMDP framework. In large state-action spaces, solving the dual problem under the DRMDP framework becomes computationally prohibitive, whereas our method remains scalable. For a detailed comparison of computational complexity, we refer the reader to Figure 2.
>
> ---
>
> We hope that we have addressed all of your questions/concerns. If you have further questions, we would be happy to answer them and if you don’t, would you kindly consider increasing your score?
>
> ---
>
> References
>
> [1] Shi, L., Li, G., Wei, Y., Chen, Y., and Chi, Y. (2022). Pessimistic Q-learning for offline reinforcement learning: Towards optimal sample complexity. In Proceedings of the 39th International Conference on Machine Learning, volume 162, pages 19967–20025. PMLR.

---

### Official Review · Reviewer_k882 · 2025-03-13

**Overall Recommendation:** 3

**Summary:**

This paper studies ways to learn a good policy in offline RL with Linear MDPs such that the policy is robust to changing the model within some f-divergence neighborhood.  More precisely, the authors consider linear MDPs and suppose they have access to an offline data set of trajectories.  Unlike standard offline RL, the authors aim to find policies that have optimal robust-regularized value, where this function is defined as the infimum over a set of feasible models (transition densities) of the expected cumulative reward of a policy plus a regularization parameter multiplied by some divergence between the model and some baseline model.  The authors consider three f-divergences: TV, KL, and chi^2 divergence and introduce an algorithmic framework based on dynamic programming, which essentially produces pessimistic estimates of the regularized Q function at each time step using an elliptic bonus.  The linearity of the model allows for a tractable decomposition of the f-divergence regularized Q-function as represented by a linear function of the features, where the representation is the sum of the reward representing vector and another vector that solves a variational problem representing the regularized value function; again, this decomposition holds due to the linearity of the MDP.  The authors express this latter variational problem as alearning problem and demonstrate htat this vector can be learned, which is how the reward estimates are constructed.  As mentioned above, pessimism is introduced through a standard elliptic bonus.  The authors use this approach to show that their dynamic programming algorithm achieves robustness and, given suffficient feature coverage of the offline dataset, good performance.  The authors show through a lower bound that the feature coverage is necessary information theoretically.  Finally, the authors conclude with a small experimental suite comparing their approach to pessimistic algorithms that do not incorporate robustness and other recent robust offline algorithms.


####

After the rebuttal, I maintain my original (positive) score.  I remain a little bit concerned about the numerical issues in long-horizon, sparse reward settings for the KL regularization setting, where we expect to see expressions like $e^{O(H)}$.

**Claims And Evidence:**

Yes.

**Essential References Not Discussed:**

N/A

**Experimental Designs Or Analyses:**

They seem fine.

**Methods And Evaluation Criteria:**

I think the methods are in general reasonable, given the limited number of naturally occurring linear MDPs.

**Other Comments Or Suggestions:**

What is the [0,H] in the second line of equation 4.4?

**Other Strengths And Weaknesses:**

I think one minor weakness of this work is the notion of robustness.  While I understand it is important for the analysis, it seems to me that requiring robustness to changes on $\mu$ is a fairly weak notion and more applicable notions of robustness might be policies that are robust to imperfectly learning the featurization (as in low-rank MDPs) or policies that are robust to small perturbations of the linearity assumptions themselves.  I also think that the paper could be significantly more clearly written, with notation defined before it is used as opposed to deferred to the appendix.

**Questions For Authors:**

- Are there not potential numerical issues with the expression for the w_h in the KL case considering that solving this problem requires exponenentiating something on the order $\Theta(H / \lambda)$?

- Is there a formulation for general f-divergences beyond those considered in the paper?

**Relation To Broader Scientific Literature:**

See summary.

**Theoretical Claims:**

I checked some of the proofs of the theoretical claims, in particular the proofs of Proposition 4.1 and Theorem 5.1.  The remaining results are believable given standard RL theory, but the reviewing time was insufficient to provide a detailed check.

---

> ### Author Rebuttal · Authors · 2025-04-01
>
> We thank the reviewers for positive feedback on our work. We hope our response fully addresses your questions
>
> ---
> **Q**: “While I understand it is important for the analysis, it seems to me that requiring robustness to changes on $\mu$ is a fairly weak notion and more applicable notions of robustness might be policies that are robust to imperfectly learning the featurization (as in low-rank MDPs) or policies that are robust to small perturbations of the linearity assumptions themselves.”
>
>
> **A**: Thank you for your insightful comment. We fully agree that considering different notions of robustness is crucial in policy learning, as the literature on robust MDPs explores diverse structured uncertainty sets.
> Regarding your reference to “policies that are robust to imperfectly learning the featurization (as in low-rank MDPs) and or policies that are robust to small perturbations of the linearity assumptions themselves”, we interpret this as pertaining to policies derived under an more general uncertainty set, such as (s,a)-uncertainty set. In such cases, the worst-case transition kernel may not admit linear structure, distinguishing it from our focus.
> We further highlight the difference between the difference of d-rectangular uncertainty structure and (s,a)-uncertainty set structure.
> - (s,a)-Uncertainty sets permit correlated perturbations across state-action pairs, leading to more conservative min-max policies (Iyengar, 2005; Wiesemann et al., 2013).
> - d-Rectangular Uncertainty sets assume independent worst-case dynamics for each (s,a) pair, resulting in less conservative policies.
> As noted in our work (Lines 75–90, left column), prior research has established the tractability of regularization-based frameworks under (s,a)-uncertainty sets. Our contribution fills the theoretical gaps for d-rectangular settings, which we believe is of independent interest.
>
> ---
> **Q**: The clearly written of robustness
>
>
> **A**: We sincerely thank the advice for removing the notation back to the main text. We remove the notation to the appendix mainly for satisfying the page limit required by the ICML. We have moved the notation to the main text.
>
> ---
> **Q**: “What is the [0,H] in the second line of equation 4.4?"
>
>
> **A**: The subscript of [0,H] refers to clip the vector into interval [0,H]. We have provided illustrations in the main text and notation.
>
> ---
> **Q**: Numerical issues when solving on O(H/lambda)
>
>
> **A**: We appreciate the reviewer's insightful comment regarding potential numerical instability in the KL divergence case. In our current experimental setting (with limited horizon H and bounded value functions), we did not encounter such numerical issues.
> As for potential numerical issues, we assume that the numerical issues may result from taking log-transformation of $e^{H/\hat{V}}$ (line 250- 258). However, thanks to the homogeneity and linear properties of Robust Regularized Bellman Equation (prop 3.2), the reward can be normalized in [0,1] to adjust to different circumstances, this may avoid numerical problems raised from the estimation of $e^{H/\hat{V}}$.
>
> ---
> **Q**: “Is there a formulation for general f-divergences beyond those considered in the paper?"
>
>
> **A**: For the definition of general f-divergence, we direct the reviewer to the notation introduced in our article (Line 551), which aligns with prior works (e.g., Yang et al. and Panaganti et al.).  While our RRMDP framework and algorithms can be extended to general f-divergence (see appendix A for details), we also highlight that we provide specific theoretical analysis with TV, KL, chi2 for two key reasons:
> - Practical Relevance: the TV, KL, chi2 divergence have already been adopted and commonly used  in empirical RL. (Shi et al.)
> - Theoretical Challenges: Analyzing sample complexity under general f divergence  remains an open problem due to the varying dual formulations induced by different divergences (see appendix A for detailed discussion). A unified framework for general f-divergences is an exciting direction for future work.
> ---
>
> We hope that we have addressed all of your questions/concerns. If you have further questions, we would be happy to answer them and if you don’t, would you kindly consider increasing your score?
>
> ---
>
> References
>
> [1] Iyengar, G. N. Robust dynamic programming. Mathematics of Operations Research, 30(2):257–280, 2005.
>
> [2] Wiesemann, W., Kuhn, D., Rustem, B.: Robust Markov decision processes. Mathematics of Operations Research 38(1), 153–183 (2013)
>
> [3]Yang, W., Wang, H., Kozuno, T., Jordan, S. M., and Zhang, Z. Robust markov decision processes without model estimation. arXiv preprint arXiv:2302.01248, 2023.
>
> [4]Panaganti, K., Wierman, A., and Mazumdar, E. Model-free robust $\phi$-divergence reinforcement learning using both offline and online data. arXiv preprint arXiv:2405.05468, 2024a.
>
> [5]Shi, L. and Chi, Y. Distributionally robust model-based offline reinforcement learning with near-optimal sample complexity. JMLR, 2024.

---

### Decision · Program_Chairs · 2025-05-01

**Decision:**

Accept (poster)

**Comment:**

Robust Offline Reinforcement Learning with Linearly Structured f-Divergence Regularization


Paper-Summary: This paper introduces a new framework called the d-rectangular linear robust regularized Markov decision process (d-RRMDP) to address the limitations of existing robust RL methods under dynamics shifts in offline settings. Unlike prior approaches that rely on unstructured or rectangular uncertainty sets, this framework introduces latent linear structure into both the transition kernels and the regularization terms, thereby improving both computational efficiency and theoretical robustness. The authors propose an algorithmic family called R2PVI (Robust Regularized Pessimistic Value Iteration), which leverages general f-divergence regularization (including TV, KL, and χ² divergences) and linear function approximation to learn robust policies from pre-collected datasets. They provide both instance-dependent and instance-independent upper bounds on the suboptimality gap of R2PVI, and establish an information-theoretic lower bound demonstrating the fundamental hardness of robust offline learning in the d-RRMDP setting.



Review and feedback: We received 3 expert reviews, with scores, 3, 3, 3, and the average score is 3.00.

The reviewers are generally positive about the technical contribution of this paper. The reviewers noted that theoretical contributions are strong and well-structured. The paper derives minimax-optimal bounds for offline RL with robustness regularization under linear MDP assumptions. The introduction of the $\phi$-rectangular uncertainty sets adds flexibility and generality, unifying various prior robust RL approaches. Reviewers appreciated the clarity of the analysis and proofs, especially the treatment of KL-divergence-based uncertainty sets and the duality approach to derive computationally tractable updates.

Reviewers have also made suggestions for improvement, including additional experimental results, computational tractability, and comparison with other algorithms. Please address these comments/suggestions while preparing the final submission.